



# Focal-TSMP: Deep learning for vegetation health prediction and agricultural drought assessment from a regional climate simulation

Mohamad Hakam Shams Eddin[1] and Juergen Gall[1,2]

[1]Institute of Computer Science, University of Bonn, Friedrich-Hirzebruch-Allee 5, 53115 Bonn, Germany
[2]Lamarr Institute for Machine Learning and Artificial Intelligence, 53115 Bonn, Germany.

**Correspondence:** Mohamad Hakam Shams Eddin (shams@iai.uni-bonn.de)

**Abstract.** In this study, we investigate applying deep learning (DL) models on a regional climate simulation produced by the Terrestrial Systems Modelling Platform (TSMP Ground to Atmosphere G2A) for vegetation health modeling and agricultural drought assessment. The TSMP simulation is performed in a free mode and the DL model is used in an intermediate step to synthesize Normalized Difference Vegetation Index (NDVI) and Brightness Temperature (BT) images from the TSMP simulation over Europe. These predicted images are then used to derive different vegetation and drought indices like NDVI anomaly, BT anomaly, Vegetation Condition Index (VCI), Thermal Condition Index (TCI), and Vegetation Health Index (VHI). To ensure reliability and to assess the model applicability with different seasonality and spatial variability, we provide an analysis of model biases and uncertainties across different regions over the Pan-Europe domain. We further provide an analysis about the contribution of the input variables from the TSMP model components to ensure a better understanding of the model prediction. A comprehensive evaluation on the long-term TSMP using reference remote sensing data showed sufficiently good agreements between the model predictions and observations. While model performance varies on the test set between different climate regions, it achieves a mean absolute error (MAE) of $0.027$ and $1.90$ K$^\circ$ with coefficient of determination ($R^2$) scores of $0.88$ and $0.92$ for NDVI and BT, respectively, at $0.11^\circ$ resolution for sub-seasonal predictions. Our study could be used as a complimentary evaluation framework for climate change simulations with TSMP. Moreover, the developed DL model could be integrated with data assimilation and used for down-stream tasks, i.e., modelling the impact of extreme events on vegetation responses with different climate change scenarios. In summary, we demonstrate the feasibility of using DL on a TSMP simulation to synthesize NDVI and BT, which can be used for agricultural drought forecasting. Our implementation is publicly available at the project page (https://hakamshams.github.io/Focal-TSMP).

## 1 Introduction

There is a growing consensus of the need to improve our state of knowledge on extreme events under a changing climate. According to recent studies on historical trends and current projections, different regions of the Earth would be more vulnerable to extreme events such as flash droughts (Christian et al., 2021, 2023; Yuan et al., 2023), meteorological and agricultural droughts (Essa et al., 2023), forest wildfires (Patacca et al., 2023), and water storage deficiency (Pokhrel et al., 2021). The expected increase in concurrence of agricultural droughts would cause crop production losses and vegetation mortality. In



particular, people in regions with fragile adaptation and mitigation strategies will be more effected. Forecasting the vegetation responses and their evolving patterns conditioned on climate scenarios is therefore a requirement to form better mitigation and adaptation strategies.

Nowadays, satellite observations around the world provide a near real-time global monitoring of vegetation and drought conditions. However, in order to prepare for long-term alleviation plans, it is desirable to forecast information about vegetation

health and agricultural drought events in the future. While short-term forecasting, i.e., for a few weeks, on a large-scale is very useful for short-term planning, a more significant contribution could be achieved with a much longer forecasting time in the future (Marj and Meijerink, 2011). In relation to this, there has been a growing line of research over the past in improving and deploying climate modelling that attempt to simulate the underlying processes of the Earth system (Shrestha et al., 2014; Lawrence et al., 2019). These modelling platforms are essential to understand changes in the water cycle and to realize and

forecast climatic extreme events in a model simulation (Miralles et al., 2019). Meanwhile, many application and domain dependent drought indices have been proposed to capture various drought signals for agricultural systems (Meza et al., 2020). Vegetation products derived from satellite land surface reflectances are used in particular as proxies for vegetation health and consequently as agricultural drought indicators (Qin et al., 2021; Vreugdenhil et al., 2022). Based on these recent developments in Earth modelling platforms and remote sensing vegetation indices, it is natural to ask the question of whether the systems of

agricultural drought forecasting could be further enhanced.

In this study, we address the problem of predicting satellite-derived vegetation indices from a free evolving simulation based on the Terrestrial Systems Modelling Platform (TSMP Ground to Atmosphere G2A) (Furusho-Percot et al., 2019). More precisely, we predict from the simulation the Normalized Difference Vegetation Index (NDVI) and Brightness Temperature (BT) as they would have been observed from AVHRR NOAA satellite systems. NDVI is computed from the reflectance in

visible red ($\rho_R$) and near-infrared bands ($\rho_{NIR}$). It is a standard product that is extensively used in applications for vegetation health and crop yield (Tucker, 1979). While BT is a calibrated spectral radiation derived from the thermal band ($\rho_{IR}$) and can be used for temperature-related vegetation stress monitoring (Kogan, 1995a). We assume that a climate simulation (i.e., TSMP simulation) that is close to the true state of the Earth should be able to reproduce vegetation products (i.e., NDVI and BT) regardless of the target satellite platform (in this study AVHRR NOAA). Radiative transfer models (RT) are normally

used to synthesis such spectral band information of specific satellite systems. Physically-based models benefit from the fact that they are built upon rich domain knowledge in physics. However, there are still many challenges related to these models. This includes the constrained parametrization for specific bands under various surface-atmosphere conditions (i.e., different cloud schemes and detailed representation of scattering processes) and satellite parameters (i.e., zenith and the scattering angles) as well as assumptions about albedo/emissivity and the high computational resources to run the model (Geiss et al.,

2021; Scheck, 2021). Besides, there exist climate-vegetation models which directly simulate the vegetation dynamic based on ecological processes and statistical modeling. Nevertheless, they are limited by the complexity of the processes and poor generalization (Chen et al., 2021). Recently, deep learning (DL) models have become popular to build a predictive model for tasks that include complex or intractable cause and effect relations within the Earth system (Bergen et al., 2019; Tuia et al., 2023). In addition, DL can be used to handle biases implicitly, thus simplifying the entire workflow (Schultz et al., 2021). For





instance, DL was recently used in climate modelling for bias correction and down-scaling to project extremes (Blanchard et al., 2022), weather forecasting (Lam et al., 2022; Chen et al., 2023; Bi et al., 2023; Ben-Bouallegue et al., 2023), and generalized multi-task learning (Nguyen et al., 2023; Lessig et al., 2023). In this work, we thus propose a DL approach based on focal modulation networks (Yang et al., 2022) to simultaneously predict NDVI and BT from the model simulation. In this way, we leverage a simulation model for long-term forecasting and DL for mapping the forecast variables to vegetation related indices

that are not part of the simulation model.

As an example of a down-stream application, we apply the predicted NDVI and BT for long-term agricultural drought forecasting, where we derive Vegetation Condition Index (VCI), Thermal Condition Index (TCI), and Vegetation Health Index (VHI) (Yang et al., 2020) as agricultural drought indicators from the predicted NDVI and BT. As part of this, we analyze whether a DL model trained on simulation produced by TSMP can be used for vegetation health forecasting at a continental-

scale by identifying regions and periods of uncertainty in the model prediction. Moreover, we analyze the importance of the input explanatory variables with explainable artificial intelligence. We achieve an overall mean absolute error (MAE) of $0.027$ and $1.90 \, \text{K}^\circ$ with coefficient of determination ($R^2$) scores of $0.88$ and $0.92$ in predicting NDVI and BT, respectively for sub-seasonal predictions at $0.11^\circ$ resolution. Our results indicate that a direct prediction of vegetation products from TSMP with deep learning is an effective way to examine the overall predictive capability of TSMP to forecast agricultural drought

events. The results suggest that a model trained on TSMP to predict vegetation products could be valuable for scenario-based assessments of vegetation response to climate change.

The rest of this article is organized as follows. Section 2 reviews the related literature. Section 3 describes the datasets that are used in the experiments. The methodology is described in Sect. 4. Section 5 and 6 include the experimental results. An analysis about variable importance is given in Sect. 7. Finally, a discussion and conclusions are provided in Sect. 8 and 9,

respectively.

## 2  Related works

### 2.1  Radiative transfer models

Forward operators like a radiative transfer (RT) solver can synthesize satellite images from the output of a numerical weather prediction (NWP) model (Li et al., 2022). These synthetic images can then be used to evaluate the representation capability

of the model or for data assimilation purposes, i.e., to verify the spatial structure of clouds in the atmospheric models. We briefly review some related works for synthetic satellite imagery. Zhang et al. (2015) applied the radiative transfer for TOVS (RTTOV) (Saunders et al., 2018) with input from the Weather Research and Forecasting (WRF) model (Skamarock et al., 2019) to model BT of oxygen and water-vapor absorption bands from geostationary satellites. Scheck et al. (2016) developed a method for fast satellite image synthesis (MFASIS), a fast 1D RT for data assimilation based on a pre-computed look-up

table with the discrete ordinate method (Stamnes et al., 1988). The reflectance at the top atmosphere is approximated by a mathematical function that takes into account the assumed relevant variables of simplified vertical profiles from the numerical weather forecasting COSMO-DE from the German Weather Service (DWD) and satellite parameters. They tested their model



for two visible satellite bands ($\rho_1$: 0.6 μm and $\rho_2$: 0.8 μm) from the spinning enhanced visible and infrared imager (SEVIRI). In Scheck et al. (2018), they extended their work to include more 3D RT effects. Another work for developing a RT model for visible and near infrared radiances was presented in (Wang et al., 2013). Furthermore, Geiss et al. (2021) analyzed the impact of cloud-related representations on visible and infrared image synthesis. They conduct a direct comparison between observed images from SEVIRI ($\rho_1$: 0.6 μm, $\rho_2$: 0.8 μm, and $\rho_3$: 10.8 μm) and their equivalent synthetic images computed based on Scheck et al. (2016) and Saunders et al. (2018) from the icosahedral non-hydrostatic model (ICON-D2, Zängl et al. (2015)). More recently, machine learning (ML) methods have being used to support data assimilation systems (Düben et al., 2021; Valmassoi et al., 2022). Such methods can be applied directly to NWP with minimum design choice of predictor variables and are considered as promising to automate the processes, i.e., ML can be used as an emulator for some physical processes. Chevallier et al. (2000) used as one of the first works neural networks for long-wave RT. Later, in a work by Lakshmanan et al. (2012), a multi-layer perceptron (MLP) was used to generate satellite images in the visible bands for model visualization. To generate training data, they applied the successive order of interaction RT solver (Heidinger et al., 2006) to compute synthetic satellite images for several days from the output of the WRF model. Ahmad et al. (2019) relied on traditional ML to predict 6 independent passive microwave BT spectral differences over snow-covered land. They used assimilated input from Noah-multiparameterization (Yang et al., 2011) and predicted satellite observations derived from the Advanced Microwave Scanning Radiometer for Earth Observing Systems (Kelly, 2009). Shi et al. (2018) performed a short-term assimilation of infrared BT ($\rho_1$: 11.2 μm, $\rho_2$: 12.0 μm, $\rho_3$: 6.7 μm and $\rho_4$: 3.9 μm) derived from radiometer FY-2D satellite data. A grid of atmospheric profiles was generated with the WRF model. This grid with different cloud micro-physical schemes was used as input for the RTTOV to simulate BT. Scheck (2021) proposed to use MLP to emulate the theoretical reflectance calculation from the MFASIS operator, where the DL model was used to replace the look-up table in Scheck et al. (2016). Similar approaches based on MLP were presented in (Stegmann et al., 2022). They were also used to emulate 3D effects on RT (Meyer et al., 2022) and to generate near-infrared satellite images ($\rho_1$: 1.6 μm) in (Baur et al., 2023). Recently, Liang et al. (2023) used ML to assimilate different bands of BT from Advanced Microwave Sounding Unit-A. In their framework, the satellite observed radiance was assimilated using RTTOV and specific MLP models were trained for each band and satellite. Yu et al. (2023) proposed to use ML models as emulators to simulate a subset of atmospheric radiation variables from a model simulation.

In this paper, we investigate the use of DL to predict products of atmospherically corrected observed albedo/emissivity on land (atmospherically corrected bottom of atmosphere) like NDVI and BT simultaneously rather than training the neural network to serve as an emulator for a predefined physical-based RT model. In other words, our training data are derived from real-world satellite observations (empirical operator) without assimilating data or assumptions about radiations. Unlike aforementioned works, we use input data from CLM (surface) and ParFlow (sub-surface) for the neural network to account for a more detailed representation of the reflectance/emissivity on ground and for land-atmosphere coupling. In addition, we built our neural network on Vision Transformers (Dosovitskiy et al., 2021) and Convolutional Neural Networks (CNN) models taking into account the spatial context around each input pixel and operating on the whole scene at once. This was motivated by previous studies that indicate that an effective model of the environment should consider the spatial-correlation within the domain (see Sect. 2.2).





## 2.2 Vegetation health prediction

A plenitude of studies exist about vegetation health prediction and forecasting from Earth observations. Unlike hydro-meteorological
variables that can be predicted or forecast using NWP, vegetation products demand an extended modeling representation of the
surface and sub-surface (Lees et al., 2022). Recently, Salakpi et al. (2022a, b) predicted short-term VCI based on previous veg-
etation conditions and observational anomaly indices in a Bayesian auto-regressive approach. However, the interaction between
vegetation and climate variability exhibit strong non-linear behaviours. In this respect, many studies explored the applicability
of DL for vegetation health monitoring using climate models and remote sensing data (Ferchichi et al., 2022). In Wu et al.
(2020), a MLP was used to model the relation between NDVI and precipitation. Kraft et al. (2019) built a global model for
NDVI dynamics using variables from ERA-Interim (Dee et al., 2011) together with static variables as predictor variables. They
built their models on a recurrent network with long short-term memory (LSTM) (Hochreiter and Schmidhuber, 1997) and MLP.
In a different study, Prodhan et al. (2021) predicted the soil moisture deficit index (SMDI) using MLP, random forests (RF), and
a global land data assimilation system (Rodell et al., 2004). Others aimed to forecast or synthesize vegetation products from
past spectral information (Nay et al., 2018; Yu et al., 2022) or vegetation statistics (Das and Ghosh, 2016; Adede et al., 2019).
Furthermore, Lees et al. (2022) used ERA5 data (Hersbach et al., 2020) and past vegetation conditions to predict short-term
VCI using an LSTM. Vo et al. (2023) proposed to use an LSTM for short-term forecasting of the natural drought index (NDI),
using an ensemble of climate model forecasts and observational data as input. Another approach was presented in (Hammad
and Falchetta, 2022) to predict short-term VHI based on probabilistic random forests (Meinshausen and Ridgeway, 2006) and
past Earth observations. Recently, Requena-Mesa et al. (2021) addressed the problem of optical satellite imagery forecasting as
a guided video prediction task. In their framework, vegetation dynamics approximated by NDVI is modeled at high resolution
using past satellite images as initial conditions and static and reanalysis data as a model guidance. Similar approaches with
this framework were presented in (Robin et al., 2022; Kladny et al., 2022; Diaconu et al., 2022) and on a continental-scale in
(Benson et al., 2023). While these works differ in their methodologies, i.e. in the predicted vegetation products, model archi-
tectures, and spatio-temporal resolutions, they have overall a good performance for short-term forecasting. Nonetheless, only
few studies address long-term vegetation conditions forecasting. Marj and Meijerink (2011) presented an approach based on
MLP and two climate signals to forecast vegetation conditions like NDVI and VCI in the next growing season. In a later study,
Miao et al. (2015) aimed to model the future change of GIMMS NDVI3g (Pinzon and Tucker, 2014) based on an ensemble of
climate scenarios CMIP5 (Taylor et al., 2012) on a decadal-scale from 2020 to 2100. They first used a linear regression to learn
the relation between climate observations and NDVI data and then used the learned relations along with climate scenarios to do
the forecasting. A similar line of research was conducted by Patil et al. (2017). They employed a RF to model NDVI images. In
their work, a RF was trained with historical climate data from the WorldClim dataset (Hijmans et al., 2005) to predict visible
and near-infrared bands observed by Landsat 7. The trained model was then used to forecast land cover response based on a
climate change scenario for the period 2061-2080. More recently in (Chen et al., 2021), an LSTM model was used to predict
NDVI on a global-scale while Wei et al. (2023) proposed to forecast the leaf area index (LAI) based on a climate projection
using a RF model trained to predict LAI from historical data.



Most studies focused only on a single indicator like NDVI excluding BT. The combination of NDVI and BT with their corresponding drought indices provides complementary information on the vegetation state and is beneficial for vegetation monitoring (Yang et al., 2020). In this study, we aim to use DL to predict vegetation products like NDVI, BT, VCI, TCI, and VHI at a continental-scale from a regional climate simulation. We also focus on long-term forecasting without using an initial state, i.e., satellite images from previous time steps. Previous works train and evaluate DL models on biased-corrected reanalysis data. In contrast, we evaluate the approach with real-world observations using a run of the simulation in the past. It is worth to note that this evaluation is more consistent with real-world deployment schemes, since it is questionable how a model that has been trained and evaluated on reanalysis data will perform on a biased climate projection simulations. Thus, we opt for a simulation that mimics a climate projection of the past and train and evaluate the model on it to internally correct biases and predict vegetation products.

## 3 Datasets and data preprocessing

In this section, we describe the datasets used in the experiments. The TSMP simulation is presented in Sect. 3.1, the observational remote sensing data for model training and evaluation are presented in Sect. 3.2, and the preprocessing framework of the data is described in Sect. 3.3

### 3.1 Regional Earth system simulation

For this study, we use the simulation produced by Terrestrial System Modelling Platform version 1.1. (TerrSysMP or TSMP) at the Research Centre Jülich (FZJ) at IBG-3 Institute and originally described in (Shrestha et al., 2014) and (Gasper et al., 2014). The simulation used in this study is introduced in (Furusho-Percot et al., 2019). TSMP is a physics-based integrated simulation representing a near-nature realization of the terrestrial hydrologic and energy cycles that cannot be directly obtained from measurements. Its setup consists of three main interconnected model components:

- The Consortium for Small Scale Modelling (COSMO) version 5.01 is a numerical weather model to simulate the diabatic and adiabatic atmospheric processes (Baldauf et al., 2011).

- The Community Land Model (CLM) version 3.5 to simulate the bio-geophysical processes on the land surface (Oleson et al., 2004, 2008).

- ParFlow version 3.2. is a hydrological model to explicitly simulate the 3D dynamic processes of water in the land surface and underground (Jones and Woodward, 2001; Kollet and Maxwell, 2006; Jefferson and Maxwell, 2015; Maxwell et al., 2015; Kuffour et al., 2020).

ECMWF ERA-Interim data (Dee et al., 2011) were used to define the initial and boundary conditions for the simulation. Based on this setup, a spinup of 10 years was conducted to reach the dynamic equilibrium before the actual run. We selected 29 main variables from COSMO, 8 variables from CLM, and 2 main variables from ParFlow. Additionally, we used 3 static





variables from the analysis (Poshyvailo-Strube et al., 2022). An analysis about the explanatory variables is provided in Sect. 7 and variable descriptions are listed in Tables A1 and A2. The three model components were fully coupled via the OASIS3 coupler (Valcke, 2013) to form a unified soil–vegetation–atmosphere model. This scheme was built without nudging or any type of DA allowing the free-running of the simulated variables. Thus, TSMP is ideal for representing the heterogeneity of the water cycle from the subsurface to the top atmosphere in a free evolution. In addition, the long-term simulation is performed for a historical time period from January 1989 until summer in September 2019 with output variables aggregated on a daily basis and extended over the Europe EURO-CORDEX EUR-11 domain (Giorgi et al., 2009; Gutowski Jr. et al., 2016; Jacob et al., 2020) with various vegetation types and climate conditions. The grid specification for TSMP is a standardized rotated coordinate system ($\phi_{meta} = 39.5°$ N, $\lambda_{meta} = 18°$ E) with a spatial resolution of $\sim 0.11°$ ($\sim 12.5$ km) and $412 \times 424$ grid cells in the rotated latitudinal and longitudinal direction, respectively. These spatio-temporal dimensions and model setup make TSMP suitable for climatological studies at a continental-scale. For recent evaluations of TSMP processes, please see (Furusho-Percot et al., 2022) and (Naz et al., 2023) and for recent studies of applying DL on TSMP simulations, please see (Patakchi Yousefi and Kollet, 2023) for bias correction and (Ma et al., 2021) for drought analysis.

## 3.2 Observational remote sensing data

Satellite-based vegetation health products were obtained from the National Oceanic and Atmospheric Administration (NOAA), Center for Satellite Applications and Research (STAR) (https://www.star.nesdis.noaa.gov/star/index.php). The blended version (Yang et al., 2020) is composed of long-term remote sensing data derived from two systems of satellites: Advanced Very High Resolution Radiometer (AVHRR) from 1981 to 2012 and its successor Visible Infrared Imaging Radiometer Suite (VIIRS) from 2013 onward. The dataset includes two essential products, namely NDVI and BT (Table A3). NDVI is computed from the red ($\rho_R$) and near-infrared ($\rho_{NIR}$) bands:

$$\text{NDVI} = \frac{(\rho_{NIR} - \rho_R)}{(\rho_{NIR} + \rho_R)}. \tag{1}$$

The NDVI is unitless and given in the range [-0.1, 1]. Same NDVI values should not be interpreted similarly for different ecosystems. In other words, the interpretation is highly dependant on the location and ecosystem productivity (Kogan, 1995b). BT is derived from the infrared ($\rho_{IR}$) band and given in Kelvin (K°) within the range [0, 400]. To handle high frequency noise caused by clouds, aerosol, and atmospheric variation along with different random error sources, NDVI and BT were temporally aggregated into smoothed noise reduced weekly products. In addition, post-launch calibration coefficients and solar/sensor zenith angles are applied to account for sensor degradation and orbital drift. The outlier removal is essential to exclude invalid measurements. Additionally, this weekly temporal resolution is enough to capture the phenological phases of vegetation and adequate for satellite data application (Kogan et al., 2011; Yang et al., 2020). Based on NDVI, BT and their long-term climatologies, the upper and lower bounds of the ecosystem can be estimated. Consequently, VCI, TCI, and VHI can be derived pixel-wise (Kogan, 1995a, 1990). The vegetation condition index is given by:

$$\text{VCI} = 100 \frac{(\text{NDVI} - \text{NDVI}_{min})}{(\text{NDVI}_{max} - \text{NDVI}_{min})}, \quad \text{with } \text{VCI} \in [0, 100], \tag{2}$$





where NDVI is the weekly noise reduced NDVI, and $\text{NDVI}_{min}$ and $\text{NDVI}_{max}$ the multi-year weekly absolute minimum and

maximum NDVI values, respectively. The thermal condition index is given by:

$$\text{TCI} = 100 \frac{(\text{BT}_{max} - \text{BT})}{(\text{BT}_{max} - \text{BT}_{min})}, \quad \text{with} \quad \text{TCI} \in [0, 100], \tag{3}$$

where BT is the weekly noise reduced BT, and $\text{BT}_{min}$ and $\text{BT}_{max}$ the multi-year weekly absolute minimum and maximum
BT values, respectively. The vegetation health index is given by:

$$\text{VHI} = (\alpha)\text{VCI} + (1 - \alpha)\text{TCI}, \quad \text{with} \quad \text{VHI} \in [0, 100], \tag{4}$$

where $\alpha$ is a weighting coefficient. While VCI is a proxy for the moisture condition and its lower values reflect a water-related
stress, TCI is a proxy for the thermal condition and its lower values indicate a temperature and wetness-related stress. The
composite index VHI is a linear combination of the former two indices to approximate the vegetation health. VHI fluctuates
annually between 0 (unfavourable condition) to 100 (favourable condition). The values of these indices above 100 and below 0
are clipped. Moreover, the dataset is provided globally with $\sim 0.05°$ ($\sim 4$ km) spatial resolution mapped into the Plate Carrée

projection. NOAA VP have been broadly used for research and real-world applications. For a summary on the validation and
studies that use this dataset for agricultural droughts monitoring, we refer to Yang et al. (2020).

### 3.3 Preprocessing

In this section we describe the data preprocessing that is needed prior to apply DL. Overall the TSMP has 30 years of data
(1989-2019). We reserved the years 1989-2009 (AVHRR era) and 2013-2016 (VIIRS era) for training, 2010-2011 (AVHRR

era) and 2017 (VIIRS era) for validation, and 2012 (AVHRR era), 2018-2019 (VIIRS era) for testing. For TSMP, we excluded
the lateral boundary relaxation zone by removing invalid grid points from the boundaries. This results in a final grid with
$397 \times 409$ grid cells in the latitudinal and longitudinal direction, respectively. In order to connect local-related characteristics
to climate conditions, we computed 3 additional static variables from the static variables described in Table A2. We computed
slope (Horn, 1981) and roughness (Wilson et al., 2007) from orography and distance to water from the land/sea mask. Due to

the fact that the remote sensing data were obtained from two different satellite systems, the data derived from VIIRS have to
be first adjusted to insure continuity and consistency with the data derived from AVHRR. Yang et al. (2018, 2021b) showed
that the discrepancy between sensors are mainly due to the differences in spectral response ranges and calibration parameters.
This has a larger effect on NDVI/VCI than on BT/TCI (Kogan et al., 2015). Considering this issue, we followed the same
re-compositing approach described in Yang et al. (2021b) to generate cross-sensor vegetation products for the time period from

2013 to 2019. In fact NDVI/BT from different sensors can be decomposed into climatologies and VCI/TCI. The climatology
provides information about the Ecosystem and it is sensor-specific. While VCI/TCI for the same ecosystem location are cross-
sensor. Thus, using climatology from AVHRR and VCI from VIIRS, Eq. (2) can be reformulated to re-compose NDVI for
VIIRS as following:

$$\text{NDVI}'_{(AVHRR)} = \left( \frac{\text{VCI}_{(VIIRS)}}{100} \right) (\text{NDVI}_{(max,AVHRR)} - \text{NDVI}_{(min,AVHRR)}) + \text{NDVI}_{(min,AVHRR)}, \tag{5}$$





where $\text{NDVI}'_{(AVHRR)}$ is the converted weekly noise reduced NDVI from VIIRS to AVHRR, $\text{VCI}_{(VIRRS)}$ is the Vegetation Condition Index derived from VIIRS, $\text{NDVI}_{(min,AVHRR)}$ and $\text{NDVI}_{(max,AVHRR)}$ are the multi-year weekly absolute minimum and maximum NDVI values (climatology) derived from AVHRR, respectively. Similarly from Eq. (3) we have:

$$\text{BT}'_{(AVHRR)} = \text{BT}_{(max,AVHRR)} - \left(\frac{\text{TCI}_{(VIIRS)}}{100}\right)\left(\text{BT}_{(max,AVHRR)} - \text{BT}_{(min,AVHRR)}\right), \tag{6}$$

where $\text{BT}'_{(AVHRR)}$ is the converted weekly noise reduced BT from VIIRS to AVHRR, $\text{TCI}_{(VIRRS)}$ is the Thermal Condition
Index derived from VIIRS, $\text{BT}_{(min,AVHRR)}$ and $\text{BT}_{(max,AVHRR)}$ are the multi-year weekly absolute minimum and maximum BT values (climatology) derived from AVHRR, respectively. Please note that $\text{VCI}_{(VIIRS)}$ and $\text{TCI}_{(VIIRS)}$ were based on a pseudo long-term VIIRS climatology (for more details on this, please see Yang et al. (2018)). In addition, the TSMP simulation and target remote sensing data have to be spatially aligned in the same domain. After the continuity at NDVI and BT level has been realized, we mapped these two products into the TSMP rotated coordinate system over the EURO-CORDEX EUR-11
domain. For the mapping, we up-scaled the data from $0.05°$ to $0.11°$ resolution based on a first-order conservative mapping (Jones, 1999) using the package from Zhuang et al. (2020). For calculating the spatial mean, we excluded invalid, water, and coastal lines pixels. Afterwards, we computed VCI, TCI and VHI based on Eq. (2)-(4). We note that the weighted coefficient $\alpha$ in Eq. (4) can be empirically calibrated as a spatially variant factor (Zeng et al., 2022, 2023). Following previous works, we set $\alpha$ to its standard value $0.5$ in all experiments as in Yang et al. (2020). Furthermore, masks over desert and very cold
areas were extracted from the quality assurance (QA) metadata provided with the data. Eventually, the preprocessed data are aggregated into data cubes on a weekly basis and stored as netCDF files. This observed remote sensing dataset can serve as a reference to train and evaluate the DL model performance. Overall, this includes $1263$, $156$, and $139$ samples (weeks) for training, validation, and testing, respectively. To avoid overfitting or the domination of few input variables, we normalized the input of TSMP by subtracting the mean and dividing by the standard deviation corresponding to each input variable. These
statistics were computed only from the years that are used for training. The invalid values of pixels were replaced with zeros values as input to the DL model.

## 4 Methodology

*Problem formulation.* Given $\text{TSMP} \in \mathbb{R}^{V \times T \times W \times H}$ as a climate change simulation, where $V$ is the number of output variables from the COSMO, CLM, and ParFlow models and the static forcing variables, $T$ is the temporal dimension and $W$ and $H$ are
the spatial extensions, our objective is to construct a mapping function $f$ to predict $\text{NDVI} \in \mathbb{R}^{I \times W \times H}$ and $\text{BT} \in \mathbb{R}^{I \times W \times H}$ on a weekly basis:

$$f : (\text{TSMP}; \theta) \rightarrow (\text{NDVI}, \text{BT}), \tag{7}$$

where $I$ is the number of weeks and $\theta$ are the weights of the model. To accomplish this, we propose to approximate this function using a DL model based on a U-Net (Ronneberger et al., 2015) with focal modulations (Yang et al., 2022) as building blocks.
The input for DL is a data cube representing a specific week $i$ of TSMP data and the output are NDVI and BT corresponding to



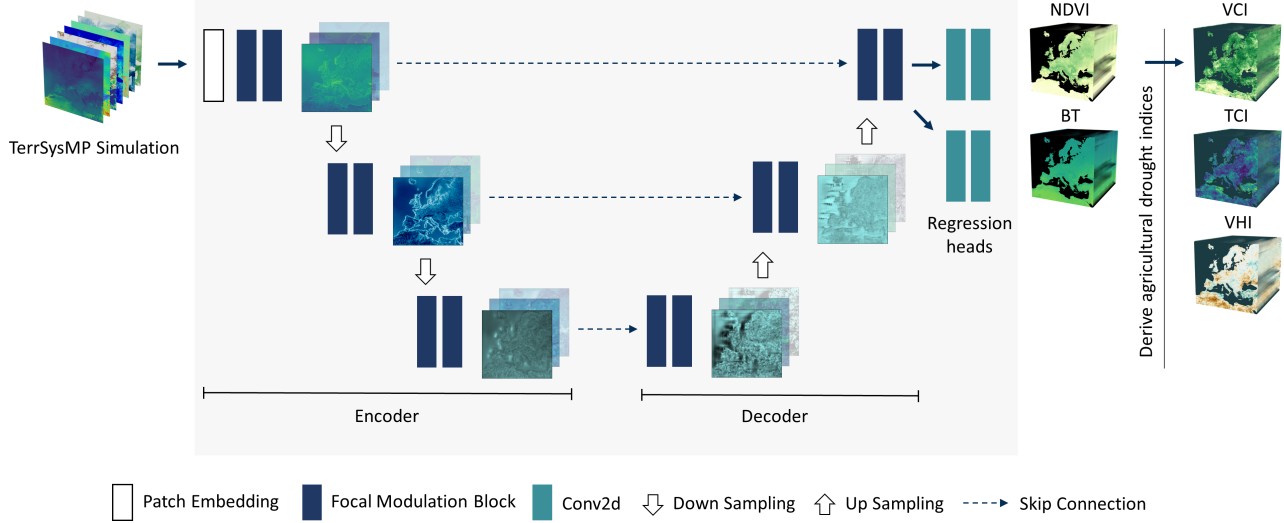

**Figure 1.** An overview of the proposed model to predict NDVI and BT from a TSMP climate simulation. The model follows the U-Net shape with encoder and decoder layers. We use focal modulation as the basic building block for the model. The input TSMP simulation is first encoded into a latent representation via encoder layers. In a subsequent step, the decoder constructs new features to be given as input to two separated regression heads that output NDVI and BT simultaneously. The predicted NDVI and BT can then be used to derive different agricultural drought indices such as VCI, TCI, and VHI.

the same week $i$. We denote the weekly averaged input data cube produced by TSMP as $\mathbf{X}^i \in \mathbb{R}^{V \times W \times H}$. Where we obtain $\mathbf{X}^i$ by taking the mean of the days corresponding to the week $i$. For simplicity, we will drop the notation $i$ in the following sections. First, the network architecture is introduced in Sect. 4.1 and the focal modulation is then described in Sect. 4.2. Finally, we discuss the loss functions in Sect. 4.3.

## 4.1 Model architectures

The model design follows the U-Net shape with encoder and decoder layers connected via skip connections and followed by two regression heads. Figure. 1 provides an overview of the model architecture. The model consists of the following main parts:

*Patch embedding*. The patch embedding is implemented as a single 1D convolution, where one patch is equivalent to one pixel. The role of this embedding is to project the input $\mathbf{X}$ from $V$ dimension into a channel dimension that matches the channel dimension $C_{(en,1)}$ of the first encoder block. In contrast to related works with transformers, we do not reduce the spatial resolution at this step. This is important to mitigate blurring effects for regression tasks. An analysis of the impact of the patch size for embedding is provided in Appendix E.

*Encoder*. The encoder consists of 3 encoding layers. Each layer has 2 consecutive focal modulation blocks that have the same number of channel dimension. We use focal modulation to capture local to global dependencies in the domain (Sect. 4.2).



We apply down-sampling on the output of the first two encoder layers to reduce the spatial resolution by a factor of 2 and double the number of channels. The down-sampling is implemented as a 2D convolution with $2 \times 2$ kernel size and stride of 2. We set $C_{(en,1)} = 96$ as the number of channels of the first encoder layer. Consequently, the encoder has the dimensionality $\{C_{(en,1)} = 96, C_{(en,2)} = 192, C_{(en,3)} = 384\}$, where $C_{(en,2)}$ is the dimensionality for the second encoder layer and $C_{(en,3)}$ is the dimensionality for the third encoder layer. The encoder allows the network to extract low to high level features in a hierarchical way. Note that focal modulation allows an additional hierarchical feature extraction at each level (Sect. 4.2).

*Skip connections.* These connections copy outputs from each encoder layer into its corresponding decoder layer. The purpose of this is to enhance the gradient flow in the network and preventing vanishing gradient issues.

*Decoder.* The decoder has a similar design to the encoder. It consists of 3 decoder layers with 2 consecutive focal modulation blocks for each decoder layer. The input for the first decoder layer is the output of the last encoder layer copied via a skip connection. While the input for the second and third decoder layers is a concatenation of the output from the previous decoder layer with the output of the corresponding encoder layer. The outputs of the first and second decoder layers are up-sampled to double the image size and reduce the dimensionality by a factor of 2. The up-sampling is implemented as a bilinear interpolation followed by a 2D convolution with $1 \times 1$ kernel size and stride of 1. The decoder layers has the dimensionality $\{C_{(de,1)} = C_{(en,3)} = 384, C_{(de,2)} = C_{(en,2)} + C_{(de,1)} = 384, C_{(de,3)} = C_{(en,1)} + C_{(de,2)} = 288\}$, where $C_{(de,1)}$, $C_{(de,2)}$, and $C_{(de,3)}$ are the dimensionality for the first, second, and third decoder layers, respectively. The purpose of the decoder is to gradually construct the input for the regression heads from the encoded features.

*Regression heads.* The ouput of the last decoder layer is then given as input to two separated regression heads to predict NDVI and BT. Each head has two 2D convolutions with $3 \times 3$ kernel size and stride of 1 with a LeakyReLU activation in between. The regression head reduces the dimensionality from $C_{(de,3)} = 288$ to 128, and then 1.

## 4.2 Focal Modulations

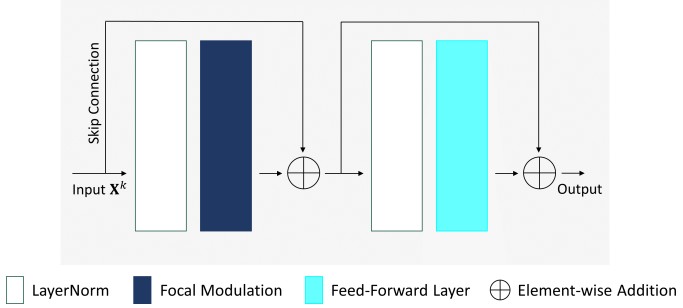

**Figure 2.** An illustration of the focal modulation block. It follows the typical transformer block with a focal modulation instead of self-attention. $\mathbf{X}^k$ represents the input to the $k$-th block

The recent applications of Vision Transformers (ViT) have covered many tasks in the field of computer vision. The network design of ViT along with the multi-head self-attention mechanism (Vaswani et al., 2017) allow ViT to stand as the state-of-the-



art backbone in recent DL models. In contrast to CNNs, ViTs with self-attention modules can handle long-range interactions

across tokens (pixels) more efficiently In a nutshell, the self-attention module aims to transfer pixels representation of a given image into a new feature representation based on a weighted aggregation of interactions between every individual pixel and its surrounding. This mechanism allows the model to focus on more relevant regions of the input images. Despite this powerful transforming process, the computational requirement of a standard ViT has limited its applications. More recently, the Focal Modulation Network (Yang et al., 2022) has been introduced to substitute the self-attention mechanism with a lightweight

focal module. In contrast to self-attention, focal modulation starts with contextual aggregation and ends with interactions. Based on this recently introduced mechanism, DL models were developed for medical image segmentation (Naderi et al., 2022; Rasoulian et al., 2023), change detection for remote sensing data (Fazry et al., 2023), and video action recognition (Wasim et al., 2023). We build our model on focal modulation networks and extend their applications in Geoscience. We first describe how the block is implemented and then describe the main focal modulation module denoted as FocalModulation.

Fig. 2 illustrates the architecture of the focal modulation block used in both the encoder and decoder layers. The design follows a typical transformer block. Let $\mathbf{X}^k \in \mathbb{R}^{N \times C^k \times W^k \times H^k}$ be the input at the $k$-th block, where $N$ is the batch size (number of input tensors), $C^k$ is the number of input channel, and $W^k$ and $H^k$ are the spatial resolution. First, the input is normalized across N via a layer normalization (Ba et al., 2016) denoted as LayerNorm. Using the indices $n \in \{1, \ldots, N\}$, $c^k \in \{1, \ldots, C^k\}$, $w^k \in \{1, \ldots, W^k\}$, and $h^k \in \{1, \ldots, H^k\}$, the LayerNorm can be written as:

$$\text{LayerNorm}(\mathbf{X}^k; (\gamma_l^k, \beta_l^k)) = \left( \frac{\mathbf{X}_{n(c^k,w^k,h^k)}^k - \mu_n^k}{\sigma_n^k} \right) \cdot \gamma_{l\,(c^k,w^k,h^k)}^k + \beta_{l\,(c^k,w^k,h^k)}^k, \tag{8}$$

$$\mu_n^k = \frac{1}{C^k W^k H^k} \sum_{c^k=1}^{C^k} \sum_{w^k=1}^{W^k} \sum_{h^k=1}^{H^k} \mathbf{X}_{n(c^k,w^k,h^k)}^k, \tag{9}$$

$$\sigma_n^k = \sqrt{\frac{1}{C^k W^k H^k} \sum_{c^k=1}^{C^k} \sum_{w^k=1}^{W^k} \sum_{h^k=1}^{H^k} (\mathbf{X}_{n(c^k,w^k,h^k)}^k - \mu_n^k)^2}, \tag{10}$$

where $\mathbf{X}_{n(c^k,w^k,h^k)}^k$ is the input tensor of order $n$ in the batch, $\mu_l^k$ and $\sigma_l^k$ are the computed mean and standard deviation of the corresponding input $\mathbf{X}_{n(c^k,w^k,h^k)}^k$, and $\gamma_{l\,(c^k,w^k,h^k)}^k \in \mathbb{R}^{C^k \times W^k \times H^k}$ and $\beta_{l\,(c^k,w^k,h^k)}^k \in \mathbb{R}^{C^k \times W^k \times H^k}$ are per-element

learnable parameters. These learnable parameters are shared across input tensors. The output of LayerNorm is then passed into the function FocalModulation. After that, the output of the first part is normalized by a second LayerNorm and passed into a feed-forward layer. The feed-forward layer consists of a one linear layer that maps the dimentionality to $r_{mlp} \times C^k$ followed by a GELU activation (Hendrycks and Gimpel, 2016) and a second linear layer to bring the dimensionality back to $C^k$, where $r_{mlp}$ is the MLP ratio parameter. We set $r_{mlp}$ to $4$ for the encoder and decrease it to $2$ for the decoder to reduce

model parameterization. The output of each block can be formulated as follows:

$$\text{FocalModulationBlock}(\mathbf{X}^k) \triangleq \gamma_2^k(\text{Feed-ForwardLayer}(\text{LayerNorm}(\gamma_1^k\text{FocalModulation}(\text{LayerNorm}(\mathbf{X}^k)) + \mathbf{X}^k)))$$
$$+ (\gamma_1^k\text{FocalModulation}(\text{LayerNorm}(\mathbf{X}^k)) + \mathbf{X}^k)), \tag{11}$$

where $\boldsymbol{\gamma}_1^k \in \mathbb{R}^{C^k}$ and $\boldsymbol{\gamma}_2^k \in \mathbb{R}^{C^k}$ are learnable scaling parameters.





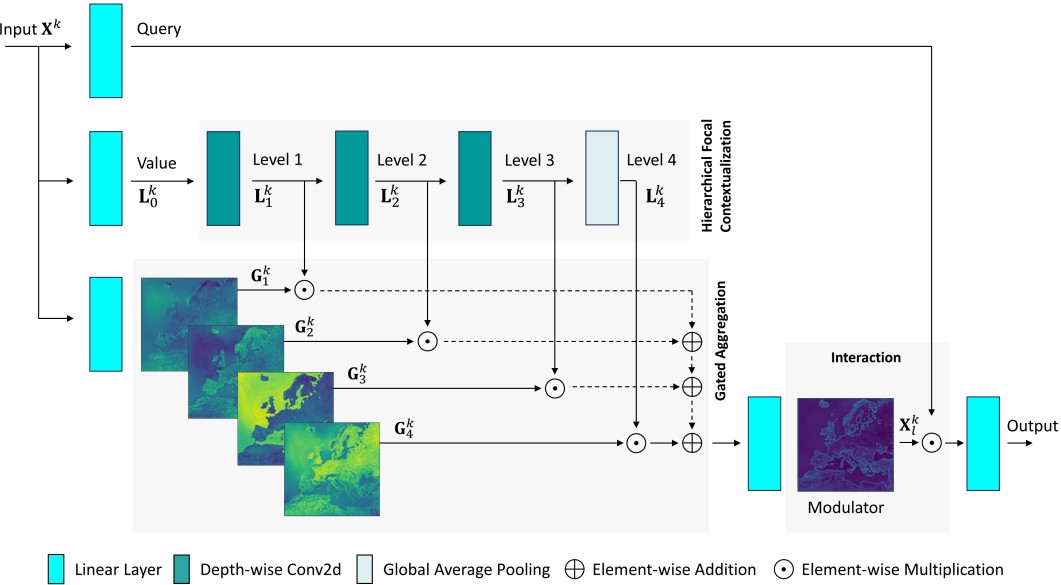

**Figure 3.** An illustration of the function FocalModulation at $k$-th block. It consists of 3 main parts: focal contextualization, gated aggregation, and interaction. First, the query, value and gates are obtained by projecting $\mathbf{X}^k$ with linear layers. Then, a stack of depth-wise 2D convolutions followed by a global pooling is used on the value to derive contextual features around pixels. Gates are used to adaptive aggregate contextual features into a modulator. Finally, the interaction between queried pixels and the modulator is performed and projected by a final linear layer to compute the output. The shown are examples of learned gates along with the pixel-wise magnitude of corresponding modulator at the first block encoder.

The heart of each focal modulation block is the FocalModulation. As seen in Fig. 3, it consists of three main steps: hierarchical contextualization, gated aggregation, and interactions.

*Hierarchical contextualization*. The objective of this part is to encode local to global range dependencies for every pixel. It is based on focal transformer (Yang et al., 2021a) and aims to extract features at 4 different levels. Let $\mathbf{X}^k$ be the input for FocalModulation and $L = 4$ be the number of levels. First, $\mathbf{X}^k$ is projected by a linear layer into a new representation

$\mathbf{L}_0^k = \mathrm{Linear}(\mathbf{X}^k) \in \mathbb{R}^{N \times C^k \times W^k \times H^k}$. Afterwards, the contexts are obtained in a recursive manner using a sequence of 3 depth-wise 2D convolutions (DWConv2D) with GeLU activation and with increased receptive fields. In DWConv2D, each output channel corresponds to a convolution on one input channel. We denote $r_l$ as the kernel size at level $l$ and start with $r_1 = 3$. Thereby, the kernel sizes at the focal levels have the values $r_1=3$, $r_2=5$, $r_3=7$. To obtain a global feature representation, a global average pooling (GAP) followed by a GeLU activation is applied at level $l = 4$. Using the index $l \in \{1, \ldots, L\}$, the

hierarchical contextualization can be formulated as follows:

$$\mathbf{L}_l^k \triangleq \begin{cases} \mathrm{GeLU}(\mathrm{DWConv2D}(\mathbf{L}_{l-1}^k)), & \text{if } 1 \leq l < L, \\ \mathrm{GeLU}(\mathrm{GAP}(\mathbf{L}_{l-1}^k)), & \text{otherwise.} \end{cases} \tag{12}$$





*Gated aggregation*. The gated aggregation adaptively summarizes the extracted hierarchical contexts $\mathbf{L}_l^k$ into a modulator. First, $\mathbf{X}^k$ is projected by a linear layer into 4 gates $\mathbf{G}^k = \text{Linear}(\mathbf{X}^k) \in \mathbb{R}^{N \times L \times W^k \times H^k}$. As can be seen from the example in Fig. 3, the third gate focuses on the water area while other gates focus on different segmented regions. This allows each pixel to

adaptively aggregate features from different semantic regions conditioned on its context. Pixels in a less dynamic environment may depend on more distant pixels while pixels in a more dynamic environment may depend more on the local context. The aggregation is performed over different focal levels and followed by a linear layer:

$$\mathbf{X}_L^k \triangleq \text{Linear}(\sum_{l=1}^{L} \mathbf{G}_l^k \odot \mathbf{L}_l^k), \tag{13}$$

where $\mathbf{X}_L^k \in \mathbb{R}^{N \times C^k \times W^k \times H^k}$ is the contextual aggregated features for each pixel called the modulator, $\mathbf{G}_l^k$ is the gate corre-

sponding to level $l$, and $\odot$ is the Hadamard operator (element-wise multiplication).

*Interaction*. Finally, the interactions between the queried pixels and the modulator is given with the following formula:

$$\text{FocalModulation}(\mathbf{X}^k) \triangleq \mathbf{X}_L^k \odot \text{Linear}(\mathbf{X}^k) \in \mathbb{R}^{N \times C^k \times W^k \times H^k}. \tag{14}$$

### 4.3 Loss function

For training we use the Mean Absolute Error (MAE) as a loss function, since it is less sensitive to outliers:

$$\mathcal{L}_{MAE} = \frac{1}{NWH} \sum_{n=1}^{N} \sum_{w=1}^{W} \sum_{h=1}^{H} |Y_{(n,w,h)} - \hat{Y}_{(n,w,h)}|, \tag{15}$$

where $N$ is the batch size, and $Y_{(n,w,h)}$ and $\hat{Y}_{(n,w,h)}$ are the predicted and observed images, respectively.

In addition, to increase local variability and balance the blurring effects from Eq. (15), we use a perceptual loss (Ledig et al., 2017; Johnson et al., 2016) based on a pre-trained VGG-19 network (Simonyan and Zisserman, 2014) on ImageNet (Deng et al., 2009). This additional loss constrains the generated images to have a similar structure and spatial variability to

the target observed images by comparing multi-level features extracted by a VGG classifier network from both the predicted and observed images:

$$\mathcal{L}_{VGG} = 8\mathcal{L}_{VGG}^1 + \sum_{j=2}^{J} \mathcal{L}_{VGG}^j, \tag{16}$$

$$\mathcal{L}_{VGG}^j = \frac{1}{NC^j W^j H^j} \sum_{n=1}^{N} \sum_{c=1}^{C^j} \sum_{w=1}^{W^j} \sum_{h=1}^{H^j} |\phi^j(Y_{(n,j,c,w,h,)}) - \phi^j(\hat{Y}_{(n,j,c,w,h,)})|, \tag{17}$$

where $J$ is the number of levels from which the VGG features are extracted, $W^j$ and $H^j$ are the spatial extensions of the

respective level within the VGG classifier, $C^j$ is the number of channel dimension of the respective level, and $\phi^j(Y_{(n,j,c,w,h,)})$ and $\phi^j(\hat{Y}_{(n,j,c,w,h,)})$ are the extracted features at level $j$ from the predicted and observed images, respectively. In contrast to classification problems where high level features play more important role, we multiply the low level features by a weighting factor of 8 to preserve the local features and give them more importance since these are more relevant to our regression task.



The VGG network was originally trained with RGB images and giving NDVI and BT as input is not directly possible. To solve
this issue, we replicate NDVI and BT along the channel dimension and feed each of them separately to the VGG network.
The impact of using this perceptual loss is evaluated in Appendix D. The entire loss function to be minimized is thus given as
follows:

$$\mathcal{L} = \mathcal{L}_{MAE}^{NDVI} + 0.1\mathcal{L}_{VGG}^{NDVI} + \mathcal{L}_{MAE}^{BT} + 0.1\mathcal{L}_{VGG}^{BT}, \tag{18}$$

where $\mathcal{L}_{MAE}^{NDVI}$ and $\mathcal{L}_{VGG}^{NDVI}$ are the MAE and VGG losses on NDVI and $\mathcal{L}_{MAE}^{BT}$ and $\mathcal{L}_{VGG}^{BT}$ are the MAE and VGG losses on
BT, respectively. The weighting factor $0.1$ is set to balance the losses. The model is trained with a stochastic gradient descent.
More technical details regarding the training are provided in Sect. 5.

## 5 Experimental results and analysis

*Performance Metrics.* To measure the model performance, we use the Mean Absolute Error (MAE), Root Mean Square Error
(RMSE), coefficient of determination ($R^2$), Pearson Correlation Coefficient ($R_p$), and Spearman Correlation Coefficient ($R_s$).
In addition, we compute the Bias as $(\text{predicted} - \text{observed} = Y_{(w,h)} - \hat{Y}_{(w,h)})$. We compute the metrics for each sample and
then average the values to obtain the last metrics. MAE is computed from Eq. (15). While RMSE can be calculated as follows:

$$\text{RMSE}(Y_{(w,h)}, \hat{Y}_{(w,h)}) = \sqrt{\frac{1}{WH}\sum_{w=1}^{W}\sum_{h=1}^{H}\left(Y_{(w,h)} - \hat{Y}_{(w,h)}\right)^2}. \tag{19}$$

$R^2$ measures the variation of the perdition from the regression fitted line and it is calculated as follows:

$$R^2(Y_{(w,h)}, \hat{Y}_{(w,h)}) = 1 - \frac{\sum_{w=1}^{W}\sum_{h=1}^{H}\left(Y_{(w,h)} - \hat{Y}_{(w,h)}\right)^2}{\sum_{w=1}^{W}\sum_{h=1}^{H}\left(Y_{(w,h)} - \bar{\hat{Y}}_{(w,h)}\right)^2}, \tag{20}$$

where $\bar{\hat{Y}}_{(w,h)}$ is the overall mean observed value. The highest value for $R^2$ is 1 which represents a perfect fit. Please note that
$R^2$ measures the variability in $\hat{Y}_{(w,h)}$ predicted by the model thus it is by definition inversely proportional to the variance and
noise in the observations and should be interpreted carefully.

Pearson correlation ($R_p$) is a parametric correlation that measures the linear correlation between the predicted and observed
values:

$$R_p(Y_{(w,h)}, \hat{Y}_{(w,h)}) = \frac{\sum_{w=1}^{W}\sum_{h=1}^{H}(Y_{(w,h)} - \bar{Y}_{(w,h)})(\hat{Y}_{(w,h)} - \bar{\hat{Y}}_{(w,h)})}{\sqrt{\sum_{w=1}^{W}\sum_{h=1}^{H}(Y_{(w,h)} - \bar{Y}_{(w,h)})^2}\sqrt{\sum_{w=1}^{W}\sum_{h=1}^{H}(\hat{Y}_{(w,h)} - \bar{\hat{Y}}_{(w,h)})^2}}, \tag{21}$$

where $\bar{Y}_{(w,h)}$ is the mean predicted value. The best value for $R_p$ is 1 which represents a perfect positive correlation.

Spearman correlation ($R_s$) is a non-parametric measure of relationship between predicted and observed values that can be
calculated as follows:

$$R_s(Y_{(w,h)}, \hat{Y}_{(w,h)}) = R_p(R(Y_{(w,h)}), R(\hat{Y}_{(w,h)})), \tag{22}$$





where $R(Y_{(w,h)})$ and $R(\hat{Y}_{(w,h)})$ are ranks obtained from the predicted and observed values, respectively. A perfect positive correlation occurs when $R_s$ is 1.

*Comparison with state-of-the-art algorithms.* We study the performance of recently developed vision transformers on our task. We achieve this by sharing the overall model architecture and implementing the main building block inside the encoder and decoder according to different algorithms. The implemented models are as follows:

**U-Net**. We implemented this model as a variation model with 2D CNN instead of Focal Modulation blocks, where we follow the same model of Focal Modulation design but replace the main building blocks with residual Convolutional blocks. It serves as a baseline of typical U-Net models.

**Swin Transformer V1** (Liu et al., 2021) performs self-attention in shifted windows to reduce the computational complexity compared to the original ViT. Transformers based on this model have been commonly applied for variety of tasks in remote
sensing and computer vision.

**Swin Transformer V2** (Liu et al., 2022) is an improved model of Swin V1. The attention mechanism is replaced with a scaled cosine attention to measure pixel feature similarities. Swin V2 utilizes post normalization layers inside the main block thus making the optimization of large models more stable. In addition, it proposes to replace the positional encoding inside the windows with a log-spaced continuous one to ease downstream tasks with pre-trained models.

**Wave-MLP** (Tang et al., 2022) is a MLP-Mixer-based transformer model. The basic block is built on a stack of MLP. Wave-MLP represents each pixel as a wave function with amplitude features representing pixel contents and phase to measure the relations with other pixels.

A part of these models, we report the results for two climatology baselines. The climatology is based on multi-year mean values computed for each pixel and week separately. The first is a climatology (1981-1988) which represents a climatology
computed before the beginning of the simulation. The second is a climatology (1989-2016) which represents a strong climatology baseline computed over the training years during the simulation. Please note that the later climatology is unrealistic, since it computes statistic in the futures. However, it represents a function that models the annual cycles and it can be used to check if the models only converges to the mean values of the predicted variables. We also note that to the best of our knowledge, currently there is no complete physically-based model that approximates the function in Eq. (7) to be compared with.

*Implementation Details.* We re-implemented all aforementioned DL models in our framework and trained them with a fixed random seed, this insures reproducibility and fair comparison. All models have almost the same capacity with $\sim 12$ million parameters. The encoders for transformer models were pre-trained on ImageNet-1K (Deng et al., 2009) while the weights in the decoders and regression heads were initialized randomly from $\mathcal{N}(\mathbf{0}, \mathbf{I=0.02})$. To increase generalization and robustness of the models, we use 3 augmentation techniques. This includes flipping and rotating of the input with a probability of $0.5$ and
randomly perturb the input variables by adding noise $\epsilon \sim \mathcal{N}(\mathbf{0}, \mathbf{I=0.02})$ with a probability of $0.5$. In addition, to generate the input corresponding to week $i$ during training, we randomly average two days corresponding to the week $i$. All models were trained with the $\mathcal{L}$ loss Eq. (18) using the Pytorch framework (Paszke et al., 2019) with a learning rate $0.0003$ and a scheduler to decay the learning rate by a factor of $0.9$ every 16 epochs. AdamW optimizer (Loshchilov and Hutter, 2019) was used for the gradient descent with ($\beta_1 = 0.9$, $\beta_2 = 0.999$) and a weight decay $0.05$. We use dropout probability of $0.2$ and a stochastic





**Table 1.** Comparing the performance of different DL models. The metrics are shown for the validation and test sets.

| | NDVI | | | | | BT (K°) | | | | |
| --- | --- | --- | --- | --- | --- | --- | --- | --- | --- | --- |
| | \multicolumn{10}{c}{Validation - Years (2010, 2011, 2017) - 156 weeks} |
| Algorithm | MAE($\downarrow$) | RMSE($\downarrow$) | $R^2$($\uparrow$) | $R_p$($\uparrow$) | $R_s$($\uparrow$) | MAE($\downarrow$) | RMSE($\downarrow$) | $R^2$($\uparrow$) | $R_p$($\uparrow$) | $R_s$($\uparrow$) |
| Climatology 1981-1988 | 0.0550 | 0.0680 | 0.5763 | 0.8939 | 0.8669 | 2.9130 | 3.7302 | 0.8454 | 0.9466 | 0.9408 |
| Climatology 1989-2016 | 0.0326 | 0.0416 | 0.8372 | 0.9353 | 0.9113 | 2.3017 | 3.0020 | 0.8963 | 0.9601 | 0.9539 |
| 2D CNN | 0.0278 | 0.0365 | 0.8746 | 0.9405 | 0.9171 | 1.9484 | 2.6067 | 0.9252 | 0.9671 | 0.9606 |
| Wave-MLP | 0.0274 | 0.0361 | 0.8765 | 0.9403 | 0.9166 | 1.9755 | 2.6395 | 0.9208 | 0.9662 | 0.9596 |
| Swin Transformer V1 | 0.0276 | 0.0364 | 0.8743 | 0.9396 | 0.9136 | 1.9599 | 2.6369 | 0.9221 | 0.9658 | 0.9589 |
| Swin Transformer V2 | 0.0274 | 0.0366 | 0.8727 | 0.9413 | 0.9173 | 1.9755 | 2.6282 | 0.9235 | 0.9663 | 0.9597 |
| **Focal Modulation** | **0.0270** | **0.0359** | **0.8781** | **0.9433** | **0.9184** | **1.8981** | **2.5433** | **0.9266** | **0.9679** | **0.9613** |
| | \multicolumn{10}{c}{Test - Years (2012, 2018, 2019) - 139 weeks} |
| | NDVI | | | | | BT (K°) | | | | |
| Algorithm | MAE($\downarrow$) | RMSE($\downarrow$) | $R^2$($\uparrow$) | $R_p$($\uparrow$) | $R_s$($\uparrow$) | MAE($\downarrow$) | RMSE($\downarrow$) | $R^2$($\uparrow$) | $R_p$($\uparrow$) | $R_s$($\uparrow$) |
| Climatology 1981-1988 | 0.0567 | 0.0697 | 0.5529 | 0.8933 | 0.8704 | 2.8806 | 3.6864 | 0.8447 | 0.9485 | 0.9470 |
| Climatology 1989-2016 | 0.0314 | 0.0400 | 0.8507 | 0.9433 | 0.9254 | 2.2024 | 2.8880 | 0.9036 | 0.9623 | 0.9606 |
| 2D CNN | 0.0278 | 0.0363 | 0.8754 | 0.9434 | 0.9231 | 1.9782 | 2.6363 | 0.9187 | 0.9650 | 0.9616 |
| Wave-MLP | 0.0267 | **0.0351** | **0.8812** | 0.9444 | 0.9241 | 1.9576 | 2.6425 | 0.9162 | 0.9646 | 0.9620 |
| Swin Transformer V1 | 0.0273 | 0.0359 | 0.8762 | 0.9431 | 0.9223 | 1.9525 | 2.6265 | 0.9183 | 0.9642 | 0.9620 |
| Swin Transformer V2 | **0.0266** | 0.0355 | 0.8795 | **0.9452** | **0.9272** | 1.9048 | 2.5782 | 0.9213 | 0.9651 | 0.9629 |
| **Focal Modulation** | 0.0268 | 0.0353 | 0.8795 | **0.9452** | 0.9243 | **1.8730** | **2.5277** | **0.9227** | **0.9672** | **0.9642** |

depth rate of 0.3. We train with a batch size of $N = 2$ for 100 epochs. For Swin Transformers, we set the window size to 8 and use the following number of heads $\{3, 6, 12\}$ for the encoder and the same order for the decoder. The down sampling in the encoder followed the original implementation in Swin Transformer. Wave-MLP was trained with the dimensionality $\{C_{(en,1)} = 64, C_{(en,2)} = 128, C_{(en,3)} = 320\}$ and $r_{mlp} = 4$ for both the encoder and the decoder. Wave-MLP and Swin V2 use a dropout probability of 0.1 and a stochastic depth rate of 0.2. In addition, we follow the official implementation of Wave-

MLP and uses GroupNorm (Wu and He, 2018) with a group of 1 instead of LayerNorm. Finnaly, all models were trained on individual NVIDIA RTX A6000 GPUs with 48 GB. For the Focal Modulation model, the training took 37 hours on a single NVIDIA RTX A6000 GPU with 4 cores. The estimated inference time to generate one sample for NDVI and BT containing $397 \times 409 \times 2$ grid points is $0.25 \pm 0.01$ seconds on one NVIDIA GeForce RTX 3090 GPU and $16 \pm 0.06$ seconds on one AMD Ryzen 9 3900X 12-Core CPU.



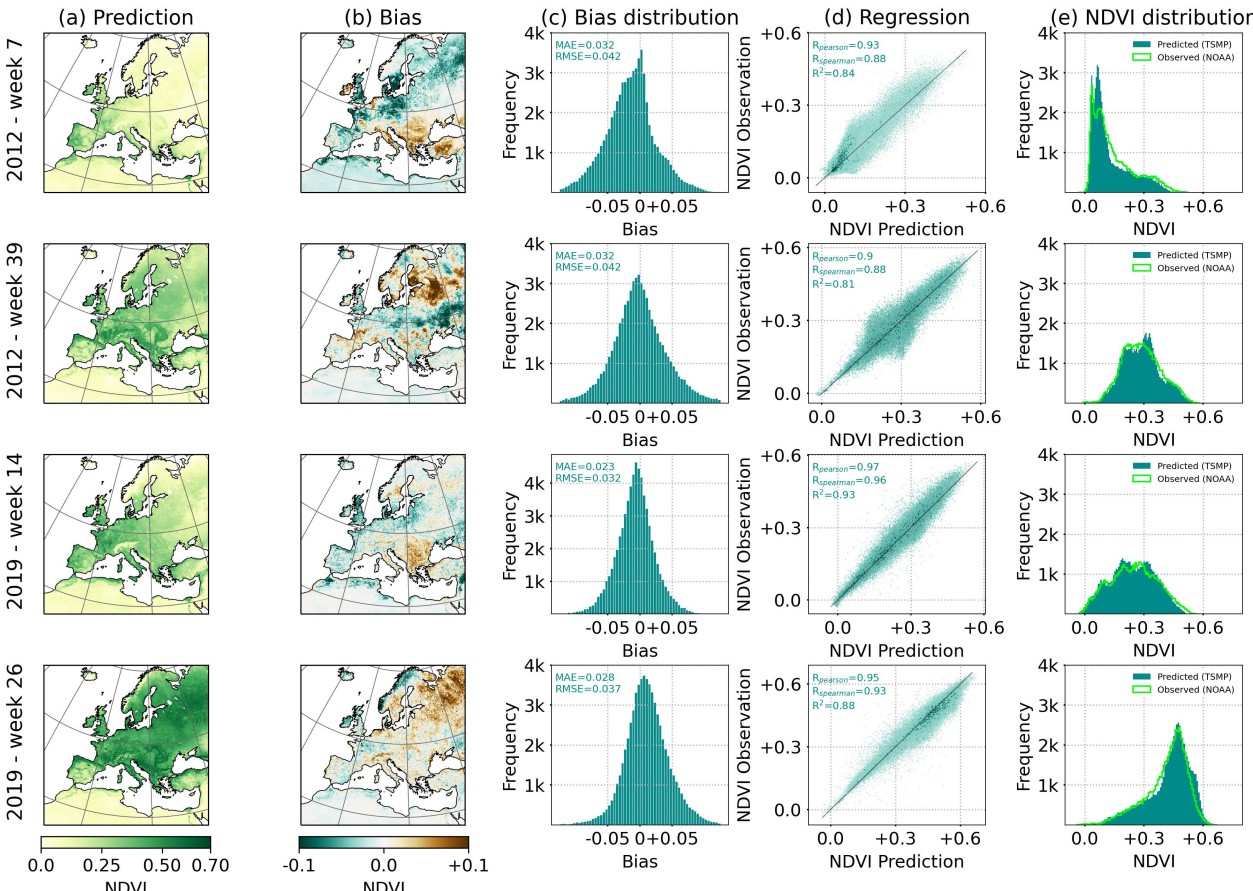

**Figure 4.** Examples predictions for weekly NDVI from the test set. (a) Predicted NDVI. (b) Bias computed as prediction minus observed. (c) Distribution of biases. (d) Regression results as predicted versus observed. (e) Distribution of NDVI values for NOAA observation and model prediction. The metrics are computed over all pixels with vegetation cover.

The quantitative results of the models are shown in Table 1. Pixels without a vegetation cover (i.e., pixels over desert) were excluded from the results. Including these pixels, will overestimate the model performance since they have small variations throughout the years. For the masking, we use NOAA quality assurance (QA) metadata. As can be seen from Table 1, all DL models outperform the first climatology (1981-1988) baseline with a huge margin. This is because the climatology was calculated before the simulation run. This climatology can not capture the dynamic after 3 decades. The second climatology

baseline is stronger. It uses information from multiple years within the simulation run. All DL models still achieve better results indicating that the models have learned the seasonal dynamic beyond climatology. In addition, these non-ML baselines can not be used to derive drought indices (Sect. 6) since they only predict the average values. Furthermore, comparing the correlation and results of BT with NDVI, we can observe that all models achieve higher correlation metrics ($R^2$, $R_p$, and $R_s$) on BT than NDVI. This can be explained by the fact that NDVI is a composition of two bands while BT is only derived from the infrared



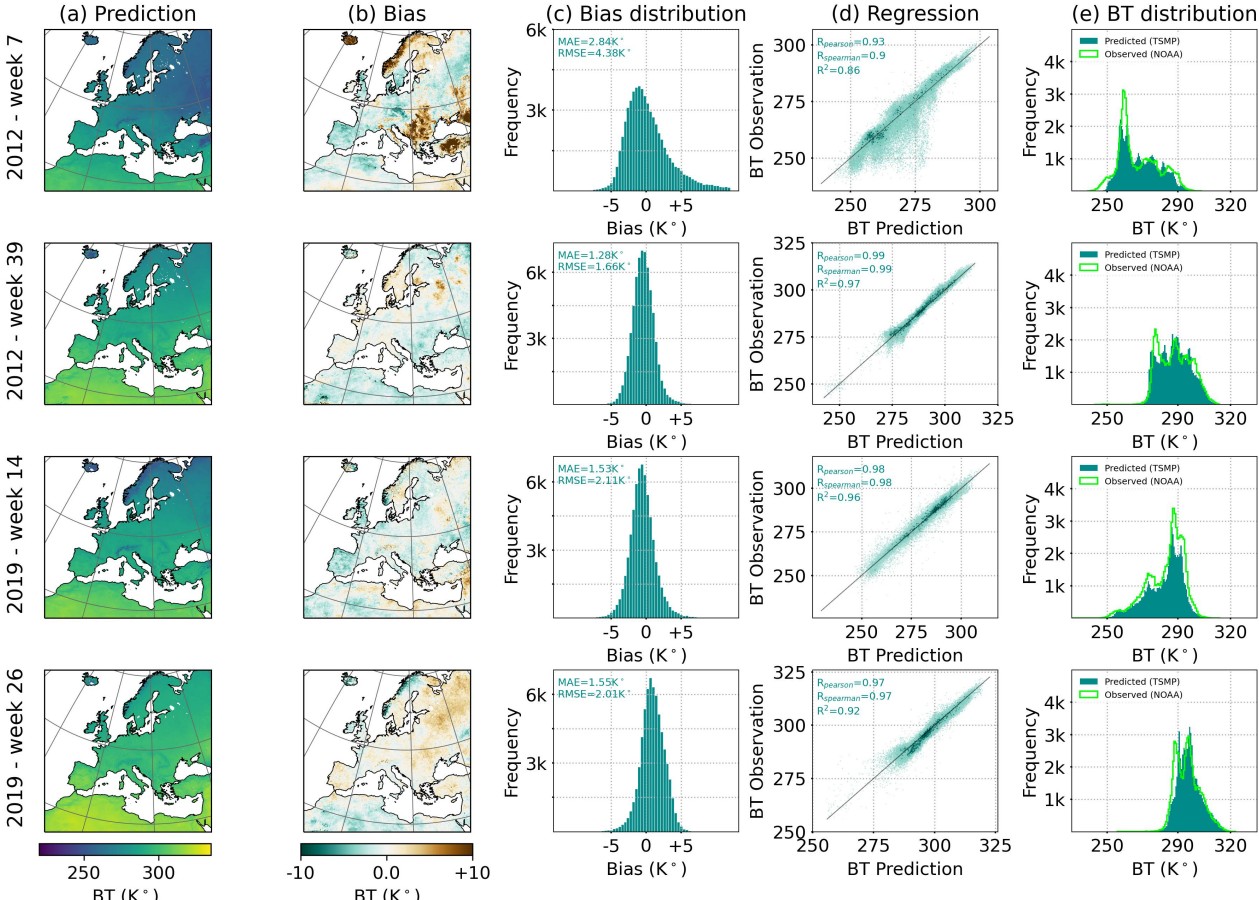

**Figure 5.** Examples predictions for weekly BT from the test set. (a) Predicted BT. (b) Bias computed as prediction minus observed. (c) Distribution of biases. (d) Regression results as predicted versus observed. (e) Distribution of BT values for NOAA observation and model prediction. The metrics are computed over all pixels with vegetation cover.

band, thus it is harder for the models to estimate NDVI than BT. This is also the case for radiative transfer models where reflectance in visible lights is effected more by scattering effects (Geiss et al., 2021). In general, all DL models provide close results and are considered suitable for the task. Focal Modulation clearly outperformed other DL models on the validation set for both NDVI and BT predictions. For the test set on NDVI, it comes slightly after Swin V2 and Wave-MLP transformers. However, Focal Modulation can generalize better for BT thus providing a balanced prediction between NDVI and BT and 480 consequently it is capable to generate an overall better prediction.

Qualitative results for the model prediction with Focal Modulation are shown in Figs. 4 and 5. We take weeks from different seasons through the years and remove pixels over desert for the calculations of bias distribution and regression line. Positive bias values mean that the model overestimates NDVI (BT) while negative ones indicate that the model underestimates NDVI



**Figure 6.** An analysis of uncertainty and model generalization for different times of the year. The analysis was performed on the validation and test sets as one set. (a) NDVI mean bias. (b) NDVI mean Pearson Correlation. (c) BT mean bias. (d) BT mean Pearson Correlation.

(BT). As it shown in Figs. 4 and 5, the biases vary across the weeks and locations. For week 7 in 2012, the biases for both NDVI

and BT are relatively high. Week 26 in 2019 exhibits similar high biases in both NDVI and BT over high latitudes regions. The respective distribution of biases is also shown Figs. 4 and 5. Overall, the results show that the dynamics over the years are well captured. The biases for both NDVI and BT are closely centered around zero with a shift for the center of bias distribution from zeros. This shift is however in the same direction for both NDVI and BT. We can also observe that the model fits the regression



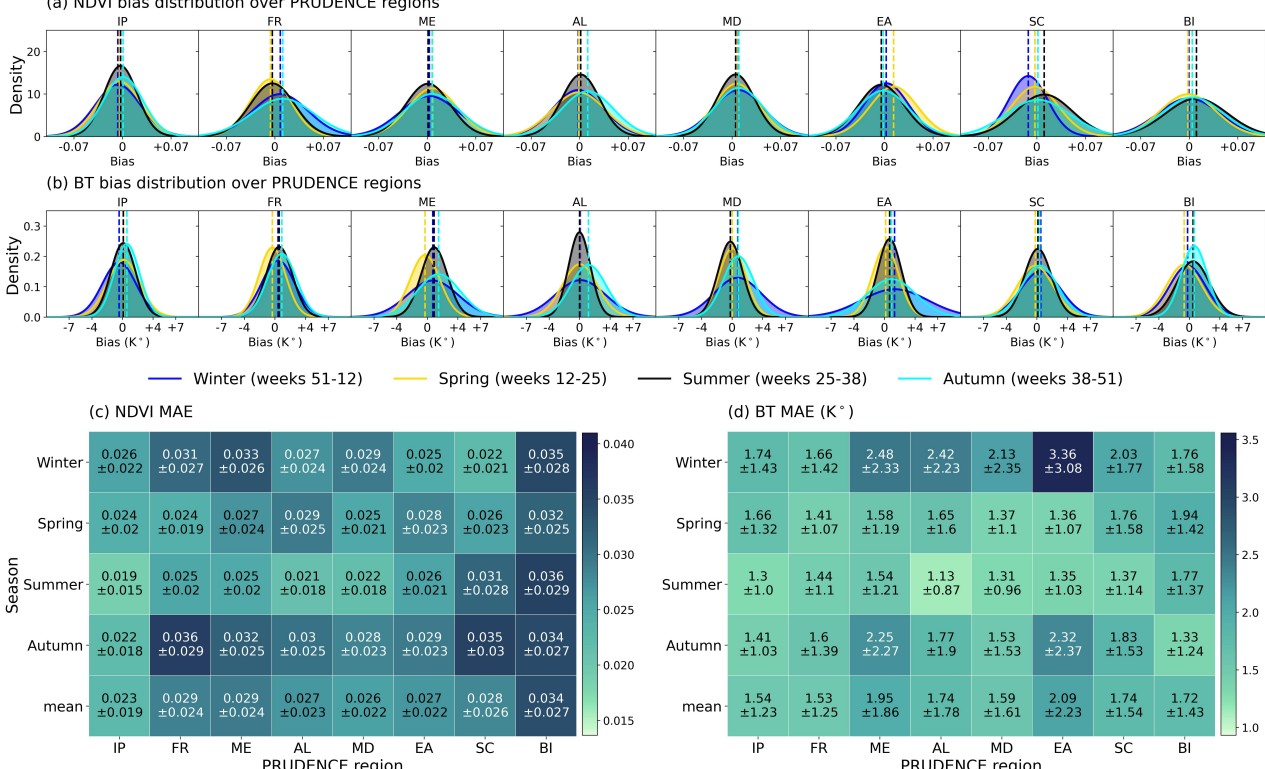

**Figure 7.** An analysis of uncertainty and model generalization for different time of the year over each PRUDENCE region. The analysis was performed on the validation and test sets as one set. (a) NDVI bias distribution. (b) BT bias distribution. Shown are the probability density functions. (c) NDVI MAE. (d) BT MAE.

lines better for weeks 14, 26, and 39 than for week 7 in winter 2012. The comparison between the distributions of predicted
and observed NDVI/BT confirms also the observation that the model captured the dynamic throughout the years.

While this provides examples of the performance for individual samples, in Fig. 6 we provide an additional experiment where we analyze biases of model predictions within different seasons of the year and over PRUDENCE regions (see Appendix Fig. C1 for the definition of PRUDENCE regions). This allows us to assess the model weakness and strengths with different seasonality and spatial variability. The mean biases were computed pixel-wise from both the validation and test years time-
series where we computed the biases for each pixels from the weeks that belong to a specific season and averaged the results to obtain the last metric. In addition, we computed Pearson Correlation $R_p$ pixel-wise in a similar way. As seen in Fig. 6, there are clusters of positive/negative biases that vary with seasons over specific regions. For instance, for NDVI prediction, the eastern part of British Isles exhibits positive biases for all seasons while Iceland and north Africa show constant negative biases. For BT, Southeastern Europe has persistent positive biases with larger errors during winter. In comparison to other seasons, the
winter season has relatively poor predictions especially in the high latitudes regions. One possible explanation for these errors





is the lack of enough and adequate training data in Scandinavian regions during winter. It was also shown by Yang et al. (2020) that high latitude regions are less reliable to derive vegetation products due to snow cover and its effects on the albedo and larger sensor zenith angles. Eisfelder et al. (2023) showed that this reliability of AVHRR time-series retrieval varies with years and across different seasons. Furthermore, this is consistent with previous studies on ParFlow-CLM models that showed that

hydrological modeling performs worse in northeastern Europe due to errors in snow dynamics and regional forces (Naz et al., 2023; Furusho-Percot et al., 2019). While these errors are common limitations of simulations and remote sensing data, it should be noted that the prediction of a DL model has its own uncertainty. Therefore, more efforts are needed to recognize the sources of uncertainty in model prediction.

Pixels over desert, i.e., north Africa, show less variability in NDVI where only little seasonality is shown as in Fig. 4. Thus,

such regions are easier to predict with relatively small biases. However any fluctuation in NDVI prediction over these pixels will lead to lower correlation compared to other regions since the time-series primarily represent noises around the mean NDVI value.

In Fig. 7, we visualize the computations over each PRUDENCE region separately. For Figs. 7a and 7b, we fit a normal distribution over the normalized histogram of biases for each season and over all PRUDENCE regions. For instance, positive

shifts of the estimated means are shown in NDVI for both FR and AL regions during autumn. The same pattern is shown fro SC and BI during summer. As can also be seen in Fig. 7b, a positive shift for BT is shown for all regions during autumn. Furthermore, the shape of the distribution gives an overview of the prediction homogeneity within the region i.e., the prediction is highly uncertain over EA during winter and consequently has a relatively high standard deviation. The mean values in Figs. 7c and 7d represent the expected MAE for all seasons combined. Fig. 7c indicates that in general the model predictions for NDVI

are less certain during autumn in comparison to other periods and over BI within the PRUDENCE regions. For BT, it can be seen from Fig. 7d that the prediction is less certain during winter and over ME and EA regions.

## 6 Agricultural drought assessment

In this section, we assess the model capability to predict different agricultural droughts indices on a high temporal resolution (weekly basis). More specifically, we use the predicted NDVI and BT along with their multi-year climatology to derive NDVI

anomaly, BT anomaly, VCI, TCI, and VHI drought indices. NDVI and BT anomalies were computed by subtracting the mean value of the respective pixel and week from the predictions (observations). VCI, TCI, and VHI were computed from Eq. (2)-(4). Figs. 8 and 9 compare the predicted agricultural drought indices VCI, TCI and VHI by the focal modulation model with the observed ones from NOAA remote sensing data for the years 2010-2012 (Fig. 8) and 2017-2019 (Fig. 9). We spatially average the values inside each PRUDENCE region and plot their respective time-series on a weekly basis. Generally, values

below 40 are identified as abnormally dry conditions (Kogan et al., 2015; Yang et al., 2020). Overall, the prediction resembles the seasonal wetness and dryness at the regional scale. The agreements between predictions and observations vary across regions and time with satisfactory $R_p$ values ranging from 0.50 to 0.77, 0.38 to 0.70, and 0.50 to 0.75 for VCI, TCI, and VHI, respectively. MAE values fluctuate in the range 9.99-6.81, 13.88-10.24, and 5.80-2.69 for VCI, TCI, and VHI, respectively.





**Figure 8.** Comparison of spatially averaged weekly agricultural drought indices between the model prediction and NOAA observation over each PRUDENCE region. Drought indices were computed from the long-term climatology (1989-2016) pixel-wise and on a weekly basis. All results are obtained with the Focal Modulation Network. The ensemble model is the result of all DL models described in Sect. 5 and Table 1. NDVI and BT anomalies are provided in Appendix Fig. F1.

While there is a satisfactory agreement with observations, there are some obvious discrepancies, i.e., in TCI over the Iberian
Peninsula (IP) during summer 2018. More interestingly, we show the bounded results of an ensemble of DL models. This ensemble is the results of all DL models from Table 1. As can be seen, all DL models which are based on different algorithms



**Figure 9.** Comparison of spatially averaged weekly agricultural drought indices between the model prediction and NOAA observations over each PRUDENCE region. Drought indices were computed based on the long-term climatology (1989-2016) pixel-wise and on a weekly basis. All results are obtained with the Focal Modulation Network. The ensemble model is the result of all DL models described in Sect. 5 and Table 1. NDVI and BT anomalies are provided in Appendix Fig. F2.

yield close predictions with small standard deviations. This supports that errors in model prediction are probably to be more attributed to biases in the TSMP model and remote sensing reference data. Finally, as observed from the plots, the thermal surface condition represented by TCI contributes more to the agricultural drought events over Europe than the deficiency in





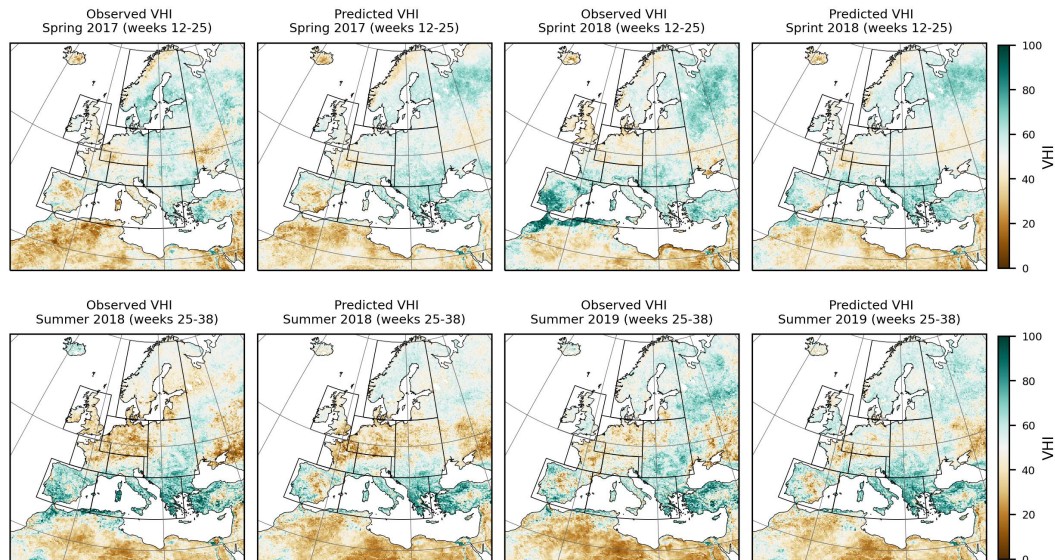

**Figure 10.** Comparison between the seasonal predicted Vegetation Health Index (VHI) and NOAA observations over Pan-Europe domain.

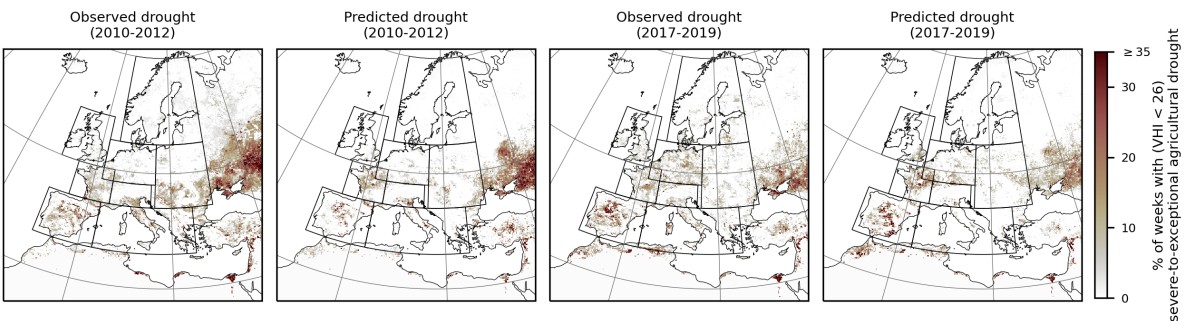

**Figure 11.** Comparison between the predicted drought frequency and NOAA observations over Pan-Europe domain. Frequency represents the percent of weeks with Severe-to-Exceptional drought events (VHI < 26).

vegetation moisture condition approximated as VCI. This is in agreement with (Zeng et al., 2023), which showed that drought effecting vegetation is more likely to be associated with high abnormal temperatures in Europe. This is critical for studies that rely on NDVI as the solely vegetation product to identify drought events over Europe (Sect. 2.2). In the Appendix, we show the time-series for NDVI and BT anomalies in Figs. F1 and F2. We also show vegetation health maps for different seasons from the validation and test years. These predicted maps are depicted in Fig. 10. As shown, i.e., the model predicts an increasing

of agricultural droughts in the summer of 2018 in the Mid Europe and France regions. Xoplaki et al. (2023) associated this extremely dry summer with compound extreme events.



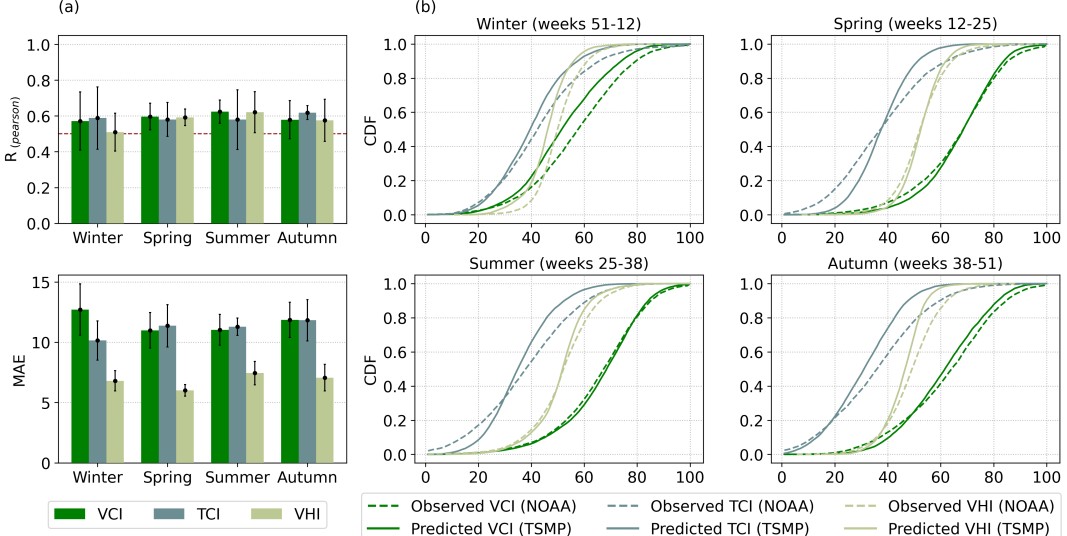

**Figure 12.** An evaluation of seasonally predicted agricultural drought indices with ground truth NOAA observations at the resolution $0.88°$. (a) Bottom is mean absolute errors (MAE) and top is Pearson Correlations ($R_p$) for different seasons. (b) Comparison of the cumulative distribution functions between prediction and observations.

Furthermore, in Fig. 11, we provide an analysis about the frequency of extreme drought for the two periods 2010-2012 and 2017-2019. Frequency represents the percent of weeks with Severe-to-Exceptional drought events where VHI < 26 (Kogan et al., 2020). While Figs. 8 and 9 provide overviews of the averaged values over the regions, the analysis in Fig. 11 provides a
spatial comparison between the model prediction and observations. The major hotspots for the highest extremes are found outside the Prudence regions (North of the Black Sea, Northwest Africa, Egypt and Northwest of the Middle East). In Comparison to the Prudence regions, the Iberian Peninsula and France exhibit more extreme droughts. The model predicts more extreme droughts in those regions and agrees with observations. For the period 2010-2012, the model predicts less extreme droughts in the Mediterranean and Eastern Europe. While for the 2017-2019 period, the model underestimates the frequency of extremes
in the Mid Europe region.

Moreover, Fig. 12 evaluates the model capability to capture seasonal dynamic in drought indices. As seen in Fig.12a, the mean $R_p$ values are greater than 0.5 and around 0.6 for all seasons. MAE values show the highest error in VCI for the winter season. One notable observation is that the error bars have relatively large values indicating a variation in prediction accuracy across the years within the same seasons. This is can be attributed to the seasonality shift in the long-term trends. Klimavičius
et al. (2023) showed that meteorological forces like air temperature have strong impact on growing seasons and phenological trends of NDVI (VCI). The cumulative distribution functions (CDF) in Fig. 12b expresses the main difference in CDF for VCI during winter. While the model prediction overestimates TCI over the seasons.



# 7 Variable importance

**Table 2.** Impact of TSMP model components on the model performance. The metrics are shown for the validation and test sets. All models were trained with Focal Modulation Network.

| | \multicolumn{5}{c}{Validation - Years (2010, 2011, 2017) - 156 weeks} | | | | | | | | | |

| | NDVI | | | | | BT (K°) | | | | |
|---|---|---|---|---|---|---|---|---|---|---|
| Model | MAE(↓) | RMSE(↓) | $R^2$(↑) | $R_p$(↑) | $R_s$(↑) | MAE(↓) | RMSE(↓) | $R^2$(↑) | $R_p$(↑) | $R_s$(↑) |
| COSMO | 0.0281 | 0.0372 | 0.8696 | 0.9403 | 0.9160 | 1.9975 | 2.6389 | 0.9227 | 0.9667 | 0.9615 |
| CLM | 0.0289 | 0.0382 | 0.8586 | 0.9369 | 0.9115 | 2.0187 | 2.7080 | 0.9160 | 0.9653 | 0.9600 |
| ParFlow | 0.0303 | 0.0396 | 0.8500 | 0.9314 | 0.9042 | 2.2029 | 2.9254 | 0.9052 | 0.9617 | 0.9545 |
| **COSMO + CLM + ParFlow** | **0.0270** | **0.0359** | **0.8781** | **0.9433** | **0.9184** | **1.8981** | **2.5433** | **0.9266** | **0.9679** | **0.9613** |

Test - Years (2012, 2018, 2019) - 139 weeks

| | NDVI | | | | | BT (K°) | | | | |
|---|---|---|---|---|---|---|---|---|---|---|
| Model | MAE(↓) | RMSE(↓) | $R^2$(↑) | $R_p$(↑) | $R_s$(↑) | MAE(↓) | RMSE(↓) | $R^2$(↑) | $R_p$(↑) | $R_s$(↑) |
| COSMO | 0.0285 | 0.0372 | 0.8619 | 0.9437 | 0.9238 | 2.0847 | 2.7549 | 0.9060 | 0.9633 | 0.9612 |
| CLM | 0.0269 | 0.0355 | 0.8782 | 0.9443 | 0.9238 | 1.9362 | 2.6303 | 0.9185 | 0.9650 | 0.9637 |
| ParFlow | 0.0291 | 0.0379 | 0.8648 | 0.9396 | 0.9175 | 2.2663 | 2.9481 | 0.8962 | 0.9635 | 0.9604 |
| **COSMO + CLM + ParFlow** | **0.0268** | **0.0353** | **0.8795** | **0.9452** | **0.9243** | **1.8730** | **2.5277** | **0.9227** | **0.9672** | **0.9642** |

To analyze the impact of each TSMP model components on the model prediction, we present in Table 2 the prediction results obtained with COSMO, CLM, and ParFlow. For this experiment, we train 3 models based on focal modulation with the dimensionality $\{C_{(en,1)} = 64, C_{(en,2)} = 128, C_{(en,3)} = 256\}$. As seen in Table 2, compared to CLM and ParFlow, COSMO achieves the best results for the validation set while CLM outperforms both for the test set. COSMO has important variables related to water contents and clouds along with other variables related to the atmospheric effects on the reflected signal on the ground. CLM has complementary variables related to heat fluxes and evapotranspiration. ParFlow can approximate the hydrology and serve as a proxy for the soil conditions. The results show that all model components are useful and the best result is obtained when all these models are used.

While Table 2 provides an overview on the importance of model components, apriori choice of proper input variables from each of these model components to predict NDVI and BT requires substantive efforts and assumptions. Especially, when the underlying physical process to construct albedo/emissivity from TSMP and tracing the atmospheric effects with satellite and solar geometry is very complex. Channel attention (Woo et al., 2018; Hu et al., 2018) was commonly used in the field of computer vision and remote sensing to enhance feature representations inside DL models. A channel attention module aims to calibrate the input variables/channels by learning an input-dependent scale for each channel. Thus, it can model the inter-



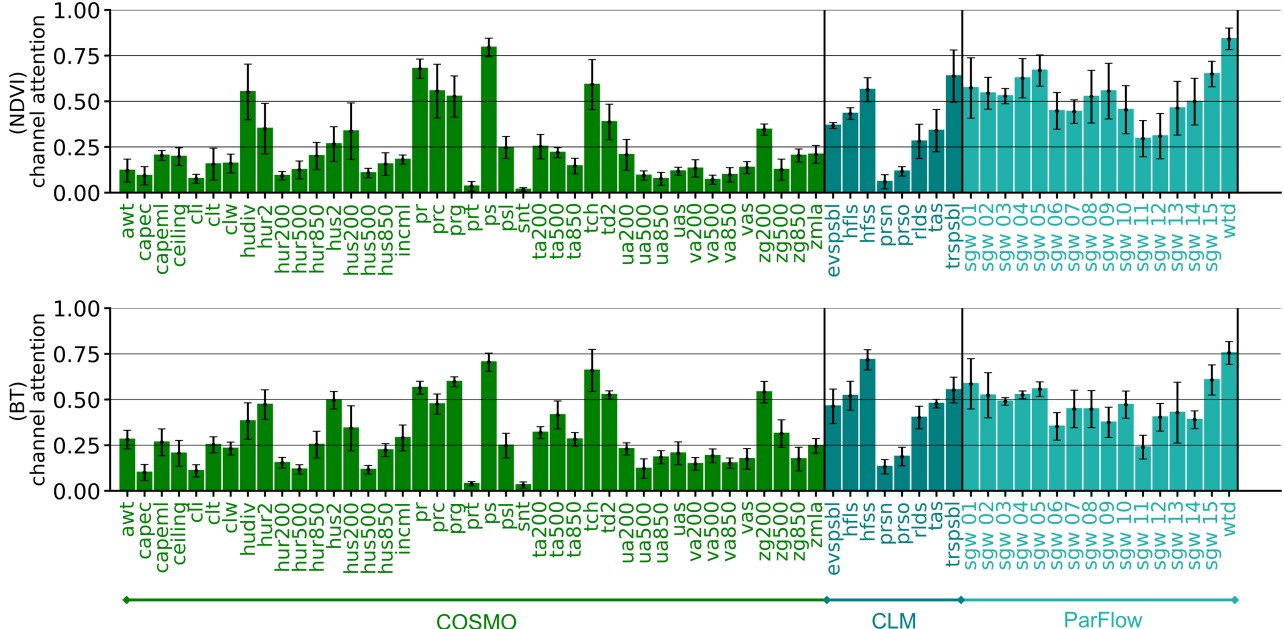

**Figure 13.** Channel attention for TSMP input variables. The activations are shown for both NDVI (top) and BT (bottom) with respect to all weeks in the validation and test sets.

correlation across variables adaptively. In this work, we propose to use channel attention to determine the relative importance of TSMP input variables. Implementation details about the module is provided in Sect. B and Fig. B1. We used channel

attention directly before the patch embedding for the U-Net model. To disentangle the correlation between NDVI and BT, we trained two separated models. One to predict NDVI and another one to predict BT. Note that we only used channel attention for this experiment. Fig. 13 provides example attentions induced for each input variable from COSMO, CLM, and ParFlow with respect to all weeks in the test and validation sets. The attention value is the mean value and it represents the variable importance to predict NDVI (BT). Error bars show how the attention changes across the weeks and input samples. We observe

that the distributions of attention values for NDVI and BT is close. This indicates that the importance of highly relevant input variables are probably shared for both NDVI and BT. In addition, the standard deviations (error bars) suggest that the choice of prior explanatory variables is not trivial since the relative importance can change with time and input samples.

  Overall, not all variables are relevant for the model. For COSMO, atmosphere water divergence (hudiv), humidity-related variables (hus, hur), precipitation variables (pr, prc, prg), surface air pressure (ps), drag coefficient of heat (tch) and geo-

potential height (zg200) receive the highest attention from the DL model. For CLM, all variables are considered important with snowfall flux (prsn) and precipitation on ground (prso) being less important. Regarding ParFlow variables, it can be seen that the model considers most underground water-related variables as relatively important. This is intuitive since water and the amount of underground water storage are important factors for the vegetation growth. The availability of groundwater supply can reduce vulnerability to agricultural drought (Meza et al., 2020; Ma et al., 2021). Some previous studies showed that



precipitation and temperature are strong predictors of NDVI (Miao et al., 2015; Wu et al., 2020; Gao et al., 2023). In addition, the climatology of long-term NDVI is highly correlated with precipitation and the biome classification (Yang et al., 2021b). The relatively high value for zg200 in BT prediction can be explained as the decrease in zg200 increases the likelihood of heatwave occurrence (Miralles et al., 2019). The attention values for COSMO can be interpreted as Nagol et al. (2009) showed that scattering and absorption in the atmosphere affect the visible and near infrared radiance considerably. Shi et al. (2018) and Geiss et al. (2021) analyzed the influence of clouds related parametrization on visible and infrared satellite images and found that the accuracy is closely related to the cloud representation. A further study about the impact of surface and air pressure and water and ice clouds on visible and near-infrared bands can be found in Baur et al. (2023).

## 8   Discussion

The evaluation confirms that a model that was trained directly on observations and based on DL has the capacity to predict NDVI and BT from a TSMP climate simulation. However, there are some limitations.

First our results are primarily related to remote sensing based agricultural drought events. In fact, some studies highlighted inconsistencies in the long-term drought trends from climate simulations (Sheffield et al., 2012; Kew et al., 2021; Vicente-Serrano et al., 2022). Meanwhile others showed positive trends in terrestrial vegetation from remote sensing products (Zhu et al., 2016; Kogan et al., 2020; Eisfelder et al., 2023). This is usually explained as assessments are highly dependent on drought definition (Satoh et al., 2021; Reyniers et al., 2023) and extreme event attribution (Van Oldenborgh et al., 2021), i.e., the drought indicator that was chosen in the methodology and the variations in modelling platforms. In addition, as mentioned in (Pirret et al., 2020; Reyniers et al., 2023) and (Pokhrel et al., 2021), prescribed vegetation assumptions exist in climate simulations which limit the modeling of atmospheric carbon effects or soil moisture deficiency on vegetation. If we add to this the complex spatio-temporal response of vegetation to climate variability (Jin et al., 2023), i.e., regional responses to climate have different dynamics and are more complicated than those at a global scale, we can conclude that predicting the vegetation state in response to drought under climate conditions still poses a major challenge. Please see Chapter 11.6 in IPCC AR6 (Seneviratne et al., 2021) for a summary of the complexity related to current drought characterization. Thus, we propose to use DL to improve the drought analysis by predicting satellite-derived vegetation indices under future climate change that can be combined with meteorological indices which are often used in studies for drought assessment to provide more comprehensive assessments. Moreover, our approach to assess agricultural drought could be extended to incorporate additional remote sensing-derived indices since no drought index is suitable for all regions. This is also reflected in the literature such as in (Qin et al., 2021). The various set of developed drought indicators implies that a more reliable drought assessment should rely on multiple indicators.

Second, the TSMP simulation was performed in a free mode and had no modelling of anthropogenic-related influences. Given that agricultural systems and human activities which are interlinked with drought events could change and follow adaptation strategies (Van Loon et al., 2016), this certainly contributes to the error budget of the model. Developing realistic land





use and water management scenarios within a probabilistic TSMP could reduce these errors. Furthermore, the uncertainty in TSMP is highly linked to potential errors in the driving forces and spin-up initialization.

Third, long-term remote sensing products such as vegetation health products are derived from different sensors and their
quality varies across regions. Data availability is subject to cloud cover and geographic location, i.e., high latitude regions are subject to occlusion and snow effects on albedo. In addition, the NOAA vegetation products depend on temporal compositing to handle high frequency and atmosphere transmittance (Yang et al., 2020). The absence of a generalized physical-based model to enhance accuracy over various surfaces and for all conditions generates difficulties for satellite products (Kogan, 1995b). Nagol et al. (2009) assessed the uncertainty of NDVI in this regards. Indeed, these causes add some uncertainties to
the model training and evaluation. Using more recent atmospheric correction methods such as in (Moravec et al., 2021) could also enhance the results. Furthermore, zenith/solar angles were not used in our study. However, these information with other correction parameters would also be valuable to enhance the accuracy (see Sect. 2.1). Yang et al. (2020) suggested that the roles of different vegetation health products to recognize drought events should be weighted in a location-aware methodology. This is also supported by studies in (Zeng et al., 2022, 2023). In this respect, Yang et al. (2021b) showed that vegetation
products over regions with extreme little seasonality, i.e., desert and high mountains have higher errors. This can be seen from Eq. (2)-(4), where small differences between maximum and minimum values could lead to higher deviation in the vegetation indices.

Fourth, our model is trained to predict vegetation products as they would be observed from the AVHRR platform. It could be appealing to predict target variables from different platforms or following different atmospheric corrections. This could be
done by training multiple DL models or by providing a satellite- or model-related guidance as input to the model. In the field of computer vision, guidance is normally used to drive the DL model to generate a target output that belongs to a specific category, i.e., a specific class in the case of image generation (Ho and Salimans, 2021). This could be addressed as a future research direction.

Fifth, as seen in Sect. 5 and 6, errors/biases produced by DL models are unpreventable. Since a tuning of trained DL model
parameters is not trivial, some experts argued that an online evaluation and continual learning are required otherwise a heuristic tuning of parameterization within the physical-based framework still has an advantage over data-driven in this regard (Rasp et al., 2018).

The results imply that by improving the modeling of NDVI and BT in TSMP, it could be possible to reduce biases in the model simulation. DL models gain more transparency when they are combined with explanations. Identifying the importance
of input variables can also be used as a starting point for related tasks, i.e., detecting anomalous events in the simulation. Nonetheless, the correlations shown in Section 7 must not be interpreted as a causal reasoning. One main reason is that data in Earth science are subject to complicated interactions and inherently inter-dependent. There may be hidden confounding variables that influence the explanatory variables as well as the evolution of the climate and vegetation variability. It is also worth noting that the learned variable importance by machine learning models is dependent on how the variables are represented
in the training data (Betancourt et al., 2022). Furthermore, since TSMP was run in a free mode simulation some variables inherited larger biases than others. This may drive the model to rely less on such variables even if they are considered important





in scientific literature. The same thing applies to highly correlated variables where changing the model architecture may alter dependencies as well (Betancourt et al., 2022).

## 9 Conclusions

In this paper, we presented a new deep learning based approach for vegetation health prediction from a regional climate simulation. The developed model enabled the prediction of variables which are not part of the input simulation. In particular, we developed a vision transformer model with focal modulation to predict NDVI and BT images from a long-term TSMP-G2A simulation at $0.11°$ resolution and on a weekly basis. We further validated the approach with NOAA remote sensing satellite observations and identified regions of uncertainty in the model predictions. We extended the commonly used explanatory

variables by using a plenty of TSMP variables and analyzed their relative importance for the task with channel attention as an explainable AI method. Our work can be extended to predict other vegetation products from different satellite platforms depending on requirements.

   Apart of this, agricultural drought assessment was performed based on vegetation health products, namely VCI, TCI and VHI, which were derived from the predicted NDVI and BT and long-term climatology. In this regard, the applicability of the

model was spatially and temporally analyzed at a continental scale. Overall, the results showed that the model performance has a sufficiently good agreement with real-world satellite observations. The main application of this study is to predict the future trends in the vegetation dynamic based on climate scenarios. Providing this information, the model can help to recognize regions that are expected to be more vulnerable to agricultural drought risks. The predicted satellite-based indices can be also combined with different meteorological drought indices to provide more comprehensive drought assessments under future

climate change.

   We believe that our study could be also useful to integrate deep learning with data assimilation, i.e., to simulate remote sensing products from down-scaled simulations and to be used as a supportive evaluation framework to further investigate the predictive capability of the simulation to reproduce drought events and consequently to improve the TSMP model development.

*Code availability.* The source code and the pretrained models to reproduce the results are published at https://zenodo.org/records/10015049
(Shams Eddin and Gall, 2023a). The source code is also available on GitHub at https://github.com/HakamShams/Focal_TSMP.

*Data availability.* The pre-processed data used in this study are available at https://doi.org/10.5281/zenodo.10008815 (Shams Eddin and Gall, 2023b). The original TSMP data are stored at Jülich Research Centre at https://www.re3data.org/repository/r3d100012923 (Furusho-Percot et al., 2019a), as well as at PANGAEA at https://doi.pangaea.de/10.1594/PANGAEA.901823 (Furusho-Percot et al., 2019b). The raw vegetation health products can be downloaded from the National Oceanic and Atmospheric Administration (NOAA), Center for Satellite

Applications and Research (STAR) at https://www.star.nesdis.noaa.gov/star/index.php (Yang et al., 2020).



# Appendix A: Datasets

**Table A1.** Technical details on the output variables in the TSMP EUR-11 simulation. For more information on the data, we refer to Furusho-Percot et al. (2019).

| Model | Variable name | Long name | Unit | Level |
|---|---|---|---|---|
| COSMO | awt | Atmosphere total water content | kg m$^{-2}$ | 1 |
| | capec | Specific convectively available potential energy | j kg$^{-1}$ | 1 |
| | capeml | Cape of mean surface layer parcel | j kg$^{-1}$ | 1 |
| | ceiling | Cloud ceiling height (above mean sea level) | m | 1 |
| | cli | Vertical integrated cloud ice | kg m$^{-2}$ | 1 |
| | clt | Total cloud fraction | 1 | 1 |
| | clw | Vertical integrated cloud water | kg m$^{-2}$ | 1 |
| | hudiv | Atmosphere water divergence | kg m$^{-2}$ | 1 |
| | hur2 | 2m relative humidity | % | 1 |
| | hur(200, 500, 850) | Relative humidity (at 200, 500 and 850 hpa) | % | 3 |
| | hus2 | 2m specific humidity | 1 | 1 |
| | hur(200, 500, 850) | Relative humidity (at 200, 500 and 850 hpa) | 1 | 3 |
| | incml | Convective inhibition of mean surface layer parcel | j kg$^{-1}$ | 1 |
| | pr | Precipitation | kg m$^{-2}$ | 1 |
| | prc | Convective precipitation | kg m$^{-2}$ | 1 |
| | prg | Large scale precipitation | kg m$^{-2}$ | 1 |
| | prt | Total rain water content vertically integrated | kg m$^{-2}$ | 1 |
| | ps | Surface air pressure | pa | 1 |
| | psl | Sea level pressure | pa | 1 |
| | snt | Total snow content vertically integrated | kg m$^{-2}$ | 1 |
| | ta(200, 500, 850) | Air temperature (at 200, 500 and 850 hpa) | K° | 3 |
| | tch | Drag coefficient of heat | 1 | 1 |
| | td2 | 2m dew point temperature | K° | 1 |
| | ua(200, 500, 850) | Eastward wind (at 200, 500 and 850 hpa) | m s$^{-1}$ | 3 |
| | uas | Eastward near-surface wind velocity | m s$^{-1}$ | 1 |
| | va(200, 500, 850) | Northward wind (at 200, 500 and 850 hpa) | m s$^{-1}$ | 3 |
| | vas | Northward near-surface wind velocity | m s$^{-1}$ | 1 |
| | zg(200, 500, 850) | Geopotential height (at 200, 500 and 850 hpa) | m | 3 |
| | zmla | Height of boundary layer | m | 1 |
| CLM | evspsbl | Evapotranspiration | mm s$^{-1}$ | 1 |
| | hfls | Surface upward sensible heat flux | w m$^{-2}$ | 1 |
| | hfss | Surface upward sensible heat flux | w m$^{-2}$ | 1 |
| | prsn | Snowfall flux | kg m$^{-2}$ s$^{-2}$ | 1 |
| | prso | Precipitation on ground | kg m$^{-2}$ s$^{-2}$ | 1 |
| | rlds | Incoming shortwave radiation | w m$^{-2}$ | 1 |
| | tas | Near-surface air temperature | K° | 1 |
| | trspsbl | Transpiration | w m$^{-2}$ | 1 |
| ParFlow | sgw | Groundwater saturation | 1 | 15 |
| | wtd | Water table depth | m | 1 |



**Table A2.** Technical details on the static variables from CLM in the TSMP EUR-11 simulation and the computed static variables.

| Model | Variable name | Long name | Unit | Level |
|---|---|---|---|---|
| CLM | orog | Surface height or digital elevation model (DEM) | m | 1 |
| | sftlf | Land-sea fraction | % | 1 |
| | zbot | Atmospheric reference height (from COSMO to CLM) | m | 1 |
| Computed from Land-sea fraction | - | Distance to water | km | 1 |
| Computed from Orography | - | Roughness | 1 | 1 |
| Computed from Orography | - | Slope | ° | 1 |

**Table A3.** Technical details on the spectral channel characteristics for Advanced Very High Resolution Radiometer (AVHRR) and Visible Infrared Imaging Radiometer Suite (VIIRS).

| Satellite system | Spectral band | Spectral range (μm) |
|---|---|---|
| AVHRR | $\rho_R$ | 0.58 - 0.68 |
| | $\rho_{NIR}$ | 0.725 - 1.1 |
| | $\rho_{IR}$ | 10.3 - 11.3 |
| VIIRS | $\rho_R$ | 0.600 - 0.680 |
| | $\rho_{NIR}$ | 0.846 - 0.885 |
| | $\rho_{IR}$ | 10.500 - 12.400 |

## Appendix B: Channel Attention

Channel attention aims to condense the input channels into a lower dimensionality and then construct channel scales with a sigmoid activation function (Sigmoid$(x) = \frac{1}{1+e^{-x}} \in [0,1]$). In this manner, the neural network learns to calibrate the input channels with the learned scaling depending on the input channels. Given $\mathbf{X} \in \mathbb{R}^{V \times W \times H}$ as input TSMP simulation, where $V$ is the number of output variables from COSMO, CLM, and ParFlow models, and $W$ and $H$ are the spatial extensions, the channel attention is computed as follows:

$$\text{ChannelAttention}(\mathbf{X}) \triangleq \text{Sigmoid}(\text{MLP}(\text{GAP}(\mathbf{X})) + \text{MLP}(\text{GSD}(\mathbf{X}))) \in \mathbb{R}^{V \times 1 \times 1}, \tag{B1}$$

where Sigmoid is the sigmoid function, MLP consists of two linear layers with a ReLU activation in between. The first decreases the dimension to $\frac{V}{r_{att}}$ and the subsequent layer maps it back to $V$. GAP is global average pooling, and GSD is the global standard deviation. For the experiments in Section 7, we trained four separated models with $r_{att} = 3$, and $r_{att} = 5$ with the dimensionality $\{C_{(en,1)} = 64, C_{(en,2)} = 128, C_{(en,3)} = 256\}$, and average the results.



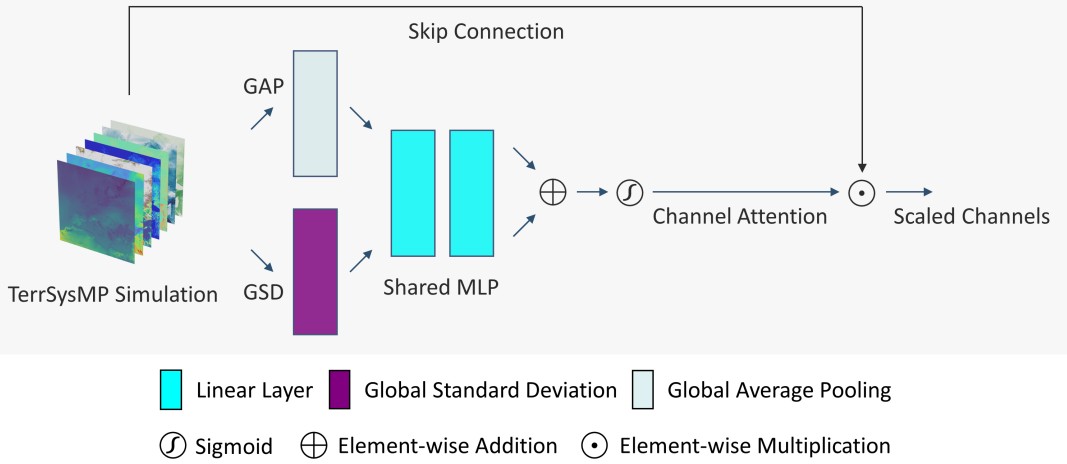

**Figure B1.** Illustration channel attention implementation. The output of channel attention is multiplied with the input TSMP to scale the channels from COSMO, CLM, and ParFlow according to their activation values.

## Appendix C: PRUDENCE scientific regions

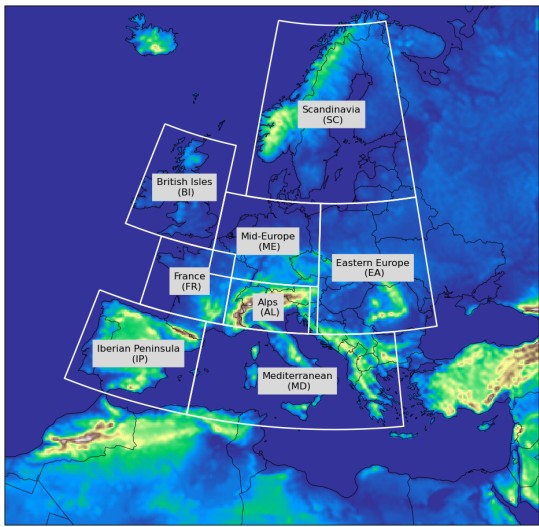

**Figure C1.** Orography over the EURO-CORDEX domain. The white boundaries with the labeled names inside define the PRUDENCE regions. The time series for validating and testing agricultural drought indices were computed over these regions.





## Appendix D: Ablation Study

In Table D1, we provide an additional analysis about the impact of the perceptual VGG loss described in Eq. (16). When adding a perceptual loss for training, we observe a consistent improvement for all metrics while residuals are slightly bigger for the test set. As shown in Figs. D1 and D2, adding the loss $\mathcal{L}_{VGG}$ reduces the blurring effect and increases variability.

**Table D1.** Ablation study on the perceptual VGG loss described in Eq. (16). The metrics are shown for the validation and test sets as one set. The used model is a U-Net based on focal modulation.

| | Loss function | NDVI | | | | | BT (K°) | | | | |
|---|---|---|---|---|---|---|---|---|---|---|---|
| | | MAE(↓) | RMSE(↓) | $R^2$(↑) | $R_p$(↑) | $R_s$(↑) | MAE(↓) | RMSE(↓) | $R^2$(↑) | $R_p$(↑) | $R_s$(↑) |
| Val | $\mathcal{L}_{MAE}$ | 0.0274 | 0.0364 | 0.8744 | 0.9400 | 0.9139 | 1.9562 | 2.5945 | 0.9255 | 0.9664 | 0.9597 |
| | $\mathcal{L}_{MAE} + \mathcal{L}_{VGG}$ | **0.0270** | **0.0359** | **0.8781** | **0.9433** | **0.9184** | **1.8981** | **2.5433** | **0.9266** | **0.9679** | **0.9613** |
| Test | $\mathcal{L}_{MAE}$ | **0.0266** | **0.0350** | **0.8819** | 0.9443 | 0.9219 | 1.9642 | 2.6329 | 0.9181 | 0.9639 | 0.9610 |
| | $\mathcal{L}_{MAE} + \mathcal{L}_{VGG}$ | 0.0268 | 0.0353 | 0.8795 | **0.9452** | **0.9243** | **1.8730** | **2.5277** | **0.9227** | **0.9672** | **0.9642** |

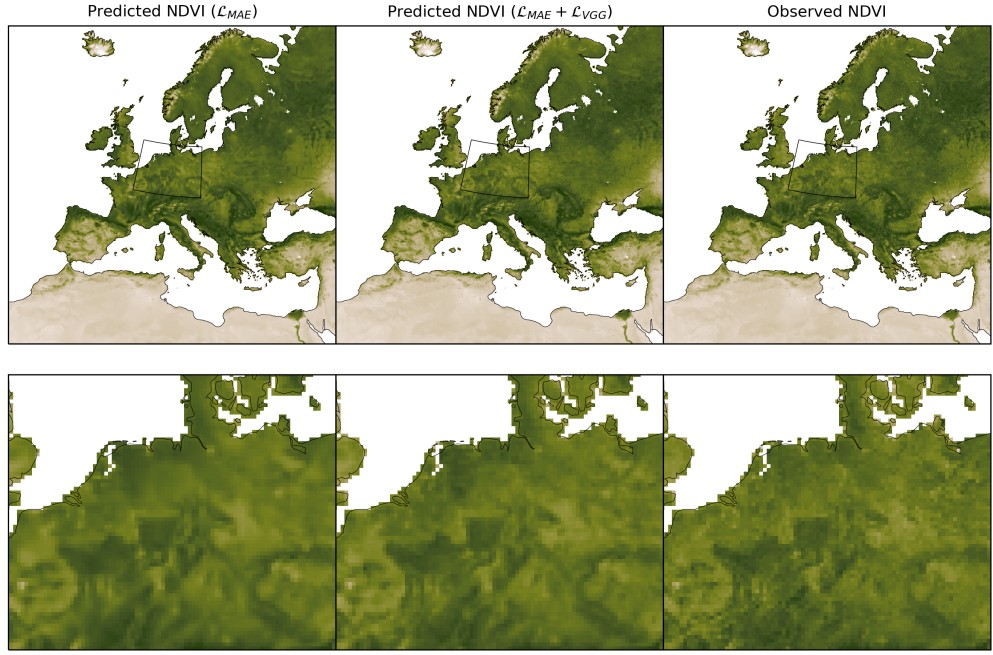

**Figure D1.** Impact of the perceptual VGG loss on NDVI predictions and image sharpness. The shown example is for the week 30 in the year 2018. Best seen in digital formats with colors.



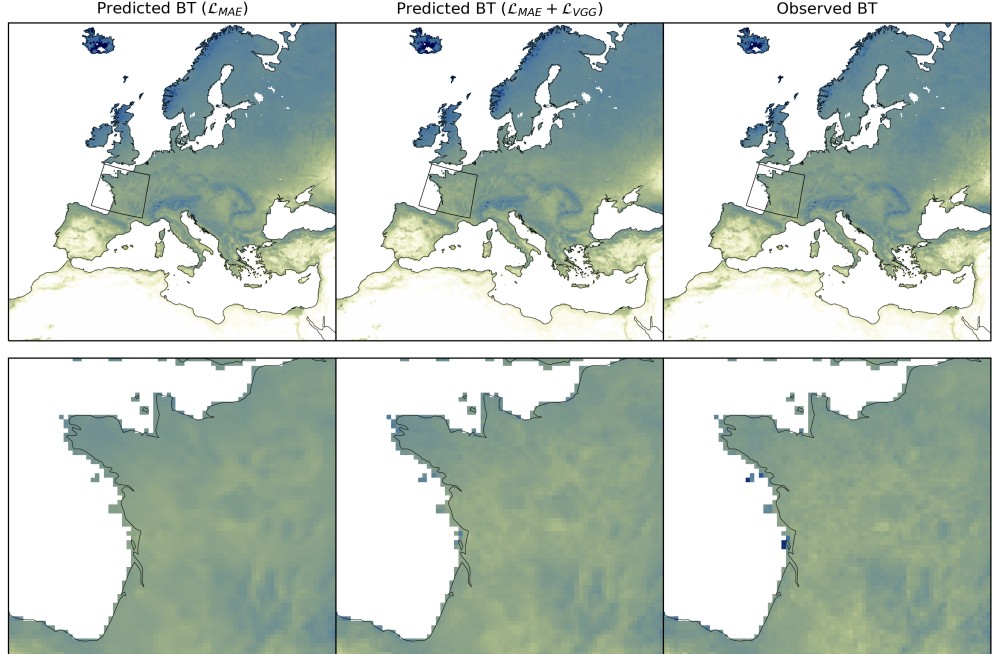

**Figure D2.** Impact of the perceptual VGG loss on BT predictions and image sharpness. The shown example is for the week 30 in the year 2018. Best seen in digital formats with colors.

## Appendix E: Patch embedding

Patch embedding with a patch size $> 1$ is commonly used in vision transformer architectures. The main aim of this embedding
is to increase the channel dimension and reduce the computational demands of the self-attention modules. This can be done by merging and embedding neighborhood pixels/tokens thus reducing the spatial or temporal resolution. In Table E1, we show that decreasing the spatial dimension of the raw input for the encoder has negative effects on our image-to-image regression task in both quantitative and qualitative terms. This can be understood as the information was lost and the model had a hardness to output the original resolution without relying on additional conditions (i.e., skip connection from the original resolution, static
features or time step). Note that for all experiments we keep using the down- and up-sampling with a factor of 2 in both encoder and decoder while we only change the patch size before the first encoder layer. To match the original spatial resolution, we used an additional bilinear up-sampling after the last decoder layer.



**Table E1.** Impact of patch size for patch embedding before the first encoder layer. The metrics are shown for the validation and test sets. The used model is a U-Net based on focal modulation model.

| | Patch size | NDVI | | | | | BT (K°) | | | | |
|---|---|---|---|---|---|---|---|---|---|---|---|
| | | MAE($\downarrow$) | RMSE($\downarrow$) | $R^2$($\uparrow$) | $R_p$($\uparrow$) | $R_s$($\uparrow$) | MAE($\downarrow$) | RMSE($\downarrow$) | $R^2$($\uparrow$) | $R_p$($\uparrow$) | $R_s$($\uparrow$) |
| Val | $1 \times 1$ | **0.0270** | **0.0359** | **0.8781** | **0.9433** | **0.9184** | **1.8981** | **2.5433** | **0.9266** | **0.9679** | **0.9613** |
| | $2 \times 2$ | 0.0280 | 0.0369 | 0.8707 | 0.9374 | 0.9116 | 1.9372 | 2.6108 | 0.9243 | 0.9664 | 0.9604 |
| | $4 \times 4$ | 0.0291 | 0.0383 | 0.8625 | 0.9345 | 0.9075 | 2.0033 | 2.6957 | 0.9184 | 0.9633 | 0.9570 |
| Test | $1 \times 1$ | **0.0268** | **0.0353** | **0.8795** | **0.9452** | **0.9243** | **1.8730** | **2.5277** | **0.9227** | **0.9672** | **0.9642** |
| | $2 \times 2$ | 0.0271 | 0.0355 | 0.8786 | 0.9422 | 0.9185 | 1.9638 | 2.6669 | 0.9157 | 0.9645 | 0.9618 |
| | $4 \times 4$ | 0.0286 | 0.0375 | 0.8644 | 0.9363 | 0.9141 | 2.1741 | 2.9132 | 0.8977 | 0.9594 | 0.9580 |

## Appendix F: Supplementary Results

*Author contributions.* This research was coordinated and supervised by JG. M.H S.E developed the software, performed the experiments, developed the method, and wrote the initial manuscript. JG reviewed and edited the manuscript. The authors read and approved the final manuscript.

*Competing interests.* The authors declared that they have no competing interests.

*Acknowledgements.* We thank Jülich Research Centre for providing the TSMP dataset to the community and Dr. Klaus Georgen for the technical discussion related to the TSMP simulation. We would also like to thank Dr. Leonhard Scheck for the helpful discussions on radiative transfer models and Dr. Petra Friederichs for the thoughtful discussions on detection and attribution of weather and climate extremes.

*Financial support.* This work was funded by the Deutsche Forschungsgemeinschaft (DFG, German Research Foundation) – SFB 1502/1–2022 - Project-number: 450058266 within the Collaborative Research Centre (CRC) for the project Regional Climate Change: Disentangling the Role of Land Use and Water Management (DETECT).



**Figure F1.** Supplementary results to Fig. 8. Comparison of spatially averaged weekly NDVI anomalies between the model prediction and NOAA observation over each PRUDENCE region. Anomaly was computed by subtracting the mean values from predictions (observations). The mean values were computed from the long-term climatology (1989-2016) pixel-wise and on a weekly basis. All results are obtained with a DL model based on Focal Modulation Network. The ensemble model is the result of all DL models described in Sect. 5 and Table 1.





**Figure F2.** Supplementary results to Fig. 8. Comparison of spatially averaged weekly BT anomalies between the model prediction and NOAA observation over each PRUDENCE region. Anomaly was computed by subtracting the mean values from predictions (observations). The mean values were computed from the long-term climatology (1989-2016) pixel-wise and on a weekly basis. All results are obtained with a DL model based on Focal Modulation Network. The ensemble model is the result of all DL models described in Sect. 5 and Table 1.



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
