# Peer review of "Focal-TSMP: Deep learning for vegetation health prediction and agricultural drought assessment from a regional climate simulation"

_EGUsphere, 2023_

## Referee Comment (RC1)

Review for "Focal-TSMP: Deep learning for vegetation health prediction and agricultural drought assessment from a reginal climate simulation" By Eddin et al.

In this study, the authors proposed deep learning models to predict NDVI and brightness temperature (BT) with simulations of a coupled earth system model as inputs. As a downstream example, the predicted NDVI and BT are used to evaluate long term agricultural drought indices (VCI, TCI and VHI) and the importance of input explanatory variables are also explored with explainable artificial intelligence. The deep learning models, based on vision transformers and CNN, takes advantages of tangling complex relations as well as biases compared to traditional radiative transfer models (RT). The results indicate an overall MAE of 0.027 and 1.90 K with R2 scores of 0.88 and 0.92 in predicting NDVI and BT and the authors claims that this framework is an effective way to examine the overall predictive capability of the earth system model to predict agricultural drought events. It is novel to use DL models to predict drought related variables which are not directly simulated by earth system models. However, the manuscript is not well written and organized, and needs major revisions before being published at GMD. My major comments are as follows:

1. The introduction lacks rationales on conducting the study. I suggest integrating sections 2.1 and 2.2 into the introduction. In the introduction, I would like to emphasize that the authors need to summarize what previous studies did, not describe what they did one by one (first paragraph in section 2.1 and first paragraph in section 2.2, which are not very readable and not necessary), but summarize and provide rationales and research gaps that this study will fill.
2. Lines 190-192: what is the rationale on variable selection?
3. Section 4.1: the authors need to provide rationale and advantages about the proposed architecture at the beginning of this section compared to state of art architectures (information in lines 322 to 333 with adjustments should be provided at the beginning of section 4.1). I also suggest using jargons as less as possible in the methodology section.
4. I suggest that lines 403-464 should be in the methodology section as baseline approaches and evaluation metrics. I also suggest the italics section in section 5 to be separated section as 5.1, 5.2 and so on, which will be much clear.
5. It is not clear what the authors would like to emphasize in lines 438-444.
6. It is not clear in line 451. Based on the methodology (lines 285-289), both the input variables and outputs are in weekly time scale, why here claims average two days?
7. The results are not well organized and too many sections. I suggest organizing all the results into one section as subsections for each topic.
8. Lines 467-471: it is confusing about different climatology. Based on the dataset information in section 3.1, the model simulation is from 1989 to 2019. What does climatology from 1981 to 1988 come from? What is the main point for analyzing different climatology (climate change?)? Table 1 is also confusing for mixing climatology different periods and  validation/test different years.
9. Based on Table 1, the metric differences for different DL models are trivial (mostly around percentage scale), although the DL models have very different building blocks. The proposed DL model is also not consistent better than others particularly for the test dataset. My doubt is whether the trivial differences are caused by stochasticity not due to DL model itself. The authors claimed fixed random seed for reproductivity, but what if the seed is not fixed and what does the results look like for running the model several times?  Will the results be consistent with any conclusion got from the table?

10. I suggest combining section 8 and 9 as one section (conclusions and limitations). The current discussion section appears only limitations not discussion.

Minor comments

1. At the beginning of abstract, I suggest adding one or two sentences about research problem the authors would like to solve.
2. AVHRR in line 44 firstly appear without full name and also check for other short names.
3. Line 50: change synthesis to be 'synthesize'.
4. Line 195: What is DA represented? If the term only shows once in the manuscript, it is not necessary to use the short name.
5. Line 200: thema and lambda represents?
6. Line 253: Is it equation (5) is for NDVI for AVHRR instead of VIIRS as stated in line 253?
7. Figure 1: It is better to use TSMP instead of TerrSysMP since TSMP is used all over the manuscript.
8. Line 325: it should have a period between 'efficiently In'.
9. Line 368-369: it is not clear why the third gate focusses on the water area? How do you know that?
10. Add references on line 430.
11. Line 448-450: the sign $N(0, I = 0.02)$ is not clear. It is better to use normal distribution with zero mean and standard deviation.
12. Line 461: reported how long it takes for the focal model, what about other DL models?
13. Line 471-472: What is these non-ML baselines? It is not clear for this statement.
14. Line 516: change 'fro' to 'for'.
15. Figure 10: change 'sprint' to 'spring' in the left two plots in the first row.

---

## Author Comment (AC2)

**Answers to Anonymous Referee #1**

We thank the anonymous referee for reviewing the manuscript and for the valuable comments. We will revise the manuscript according to the suggestions. Below are the comments and our detailed responses.

**Major Comment**

**Comment RC1.1** *The introduction lacks rationales on conducting the study. I suggest integrating sections 2.1 and 2.2 into the introduction. In the introduction, I would like to emphasize that the authors need to summarize what previous studies did, not describe what they did one by one (first paragraph in section 2.1 and first paragraph in section 2.2, which are not very readable and not necessary), but summarize and provide rationales and research gaps that this study will fill.*

**Answer to RC1.1** We will revise the introduction and integrate it with the related works section.

**Comment RC1.2** *Lines 190-192: what is the rationale on variable selection?*

**Answer to RC1.2** Only variables within the period applicable for analysis were selected. Some variables were not selected due to the reason that for some variables in the model, the full groundwater states equilibrium starts after 1995. In addition, we did not select the variables that do not cover the whole period of the simulation.

**Comment RC1.3** *Section 4.1: the authors need to provide rationale and advantages about the proposed architecture at the beginning of this section compared to state of art architectures (information in lines 322 to 333 with adjustments should be provided at the beginning of section 4.1). I also suggest using jargons as less as possible in the methodology section.*

**Answer to RC1.3** We will revise section 4.1 accordingly and simplify some terms when possible.

**Comment RC1.4** *I suggest that lines 403-464 should be in the methodology section as baseline approaches and evaluation metrics. I also suggest the italics section in section 5 to be separated section as 5.1, 5.2 and so on, which will be much clear.*

**Answer to RC1.4** We will revise the manuscript accordingly.

**Comment RC1.5** *It is not clear what the authors would like to emphasize in lines 438-444.*

**Answer to RC1.5** The main application of the study is to estimate the vegetation condition using a DL-based model for periods where no satellite images are available. As a baselines, we compare the results of DL models to two NDVI/BT climatologies from remote sensing observations (the climatology is computed pixel-wise and on a weekly basis). The climatology has the limitation that the inter-annual variability in NDVI/BT is neglected as average cycles are used. The first climatology (1981-1988) is used to show the limitation of using prescribed satellite phenology for future projection and we argue a DL-based model maybe a better replacement (for

example future simulations often prescribe vegetation condition in a satellite phenology-mode neglecting the inter-annual variability [1]). This is still not a fair comparison because the DL-models were trained on periods succeed this climatology. The second climatology (1989-2016) is used to show that the DL-models still outperform the climatology and generalized beyond the mean annual cycles of NDVI/BT. We could compute the second climatology (1989-2016) because we used a climate simulation in an overlapped period with historical remote sensing observations. This comparison is not possible for future climate projections since no satellite observations exist yet. Please note that we excluded the validation/testing (2010-2012) years for the computation of this second climatology. The last part is that we could not compare to a specific physically-based model since such a model has to construct the albedo and emission for visible, near-infrared, and infrared bands under all conditions and for all surface types and then trace the atmospheric correction (NOAA AVHRR depends on temporal compositing for the atmospheric correction) to the top of the atmosphere regardless of the solar and satellite geometry. We will rewrite lines 438-444.

**Comment RC1.6** *It is not clear in line 451. Based on the methodology (lines 285-289), both the input variables and outputs are in weekly time scale, why here claims average two days?*

**Answer to RC1.6** We use this as an augmentation technique only during training. For validation and test, we average all days. We will make this clearer in the revised manuscript.

**Comment RC1.7** *The results are not well organized and too many sections. I suggest organizing all the results into one section as subsections for each topic*

**Answer to RC1.7** We will revise the manuscript accordingly.

**Comment RC1.8** *Lines 467-471: it is confusing about different climatology. Based on the dataset information in section 3.1, the model simulation is from 1989 to 2019. What does climatology from 1981 to 1988 come from? What is the main point for analyzing different climatology (climate change?)? Table 1 is also confusing for mixing climatology different periods and validation/test different years.*

**Answer to RC1.8** Please see the answer to RC1.5. We agree that the comparison to climatology in tables may appear confusing. We will make it clearer in the suggested new section "baseline approaches" and change the name of climatology (1981-1988) to climatology I and climatology (1989-2016) to climatology II. For validation/test, we took the last 3 years of AVHRR (2010-2012) and the last 3 years of converted VIIRS (2017-2019).

**Comment RC1.9** *Based on Table 1, the metric differences for different DL models are trivial (mostly around percentage scale), although the DL models have very different building blocks. The proposed DL model is also not consistent better than others particularly for the test dataset. My doubt is whether the trivial differences are caused by stochasticity not due to DL model itself. The authors claimed fixed random seed for reproductivity, but what if the seed is not fixed and what does the results look like for running the model several times? Will the results be consistent with any conclusion got from the table?*

**Answer to RC1.9** Table A reports the mean and standard deviation for training with 3 different random seeds and we will replace Table 1. In general, there are some differences for the results on the test set and we will adjust our comments accordingly in the revised manuscript. We would

also like to note that all of the described models are our implementation and we found Focal Modulation among them to work overall the best. We also replaced the name 2D CNN with U-Net to be consistent through out the manuscript.

Table A: Comparing the performance of different DL models on the validation and test sets.

| Validation - Years (2010, 2011, 2017) - 156 weeks | | | | | |
|---|---|---|---|---|---|
| NDVI | | | | | |
| Algorithm | MAE($\downarrow$) | RMSE($\downarrow$) | $R^2$($\uparrow$) | $R_p$($\uparrow$) | $R_s$($\uparrow$) |
| Climatology 1981-1988 | 0.0550 | 0.0680 | 0.5763 | 0.8939 | 0.8669 |
| Climatology 1989-2016 | 0.0326 | 0.0416 | 0.8372 | 0.9353 | 0.9113 |
| U-Net | 0.0277 ±0.0001 | 0.0365 ±0.0002 | 0.8743 ±0.0008 | 0.9406 ±0.0005 | 0.9172 ±0.0005 |
| Wave-MLP | _0.0272_ ±0.0003 | **0.0358** ±0.0003 | _0.8784_ ±0.0018 | _0.9422_ ±0.0018 | _0.9183_ ±0.0021 |
| Swin Transformer V1 | 0.0273 ±0.0003 | 0.0362 ±0.0003 | 0.8759 ±0.0022 | 0.9411 ±0.0013 | 0.9161 ±0.0023 |
| Swin Transformer V2 | 0.0277 ±0.0003 | 0.0369 ±0.0003 | 0.8703 ±0.0021 | 0.9415 ±0.0010 | 0.9167 ±0.0008 |
| **Focal Modulation** | **0.0269** ±0.0001 | **0.0358** ±0.0002 | **0.8790** ±0.0017 | **0.9432** ±0.0001 | **0.9194** ±0.0009 |
| BT (K) | | | | | |
| Algorithm | MAE($\downarrow$) | RMSE($\downarrow$) | $R^2$($\uparrow$) | $R_p$($\uparrow$) | $R_s$($\uparrow$) |
| Climatology 1981-1988 | 2.9130 | 3.7302 | 0.8454 | 0.9466 | 0.9408 |
| Climatology 1989-2016 | 2.3017 | 3.0020 | 0.8963 | 0.9601 | 0.9539 |
| U-Net | 1.9377 ±0.0093 | 2.6067 ±0.0057 | 0.9243 ±0.0014 | 0.9667 ±0.0004 | _0.9603_ ±0.0007 |
| Wave-MLP | _1.9200_ ±0.0491 | _2.5834_ ±0.0486 | _0.9248_ ±0.0035 | _0.9668_ ±0.0006 | _0.9603_ ±0.0007 |
| Swin Transformer V1 | 1.9642 ±0.0246 | 2.6341 ±0.0303 | 0.9221 ±0.0012 | 0.9661 ±0.0005 | 0.9590 ±0.0006 |
| Swin Transformer V2 | 1.9741 ±0.0191 | 2.6420 ±0.0258 | 0.9225 ±0.0013 | 0.9659 ±0.0011 | 0.9590 ±0.0014 |
| **Focal Modulation** | **1.9010** ±0.0071 | **2.5364** ±0.0073 | **0.9280** ±0.0012 | **0.9679** ±0.0001 | **0.9614** ±0.0007 |
| Test - Years (2012, 2018, 2019) - 139 weeks | | | | | |
| NDVI | | | | | |
| Algorithm | MAE($\downarrow$) | RMSE($\downarrow$) | $R^2$($\uparrow$) | $R_p$($\uparrow$) | $R_s$($\uparrow$) |
| Climatology 1981-1988 | 0.0567 | 0.0697 | 0.5529 | 0.8933 | 0.8704 |
| Climatology 1989-2016 | 0.0314 | 0.0400 | 0.8507 | 0.9433 | 0.9254 |
| U-Net | 0.0274 ±0.0004 | 0.0359 ±0.0005 | 0.8772 ±0.0006 | 0.9435 ±0.0006 | 0.9237 ±0.0009 |
| Wave-MLP | **0.0261** ±0.0006 | **0.0343** ±0.0008 | **0.8861** ±0.0043 | **0.9467** ±0.0024 | _0.9252_ ±0.0011 |
| Swin Transformer V1 | 0.0269 ±0.0003 | 0.0355 ±0.0004 | 0.8795 ±0.0029 | 0.9442 ±0.0010 | 0.9239 ±0.0014 |
| Swin Transformer V2 | 0.0270 ±0.0005 | 0.0359 ±0.0005 | 0.8766 ±0.0038 | 0.9447 ±0.0012 | 0.9251 ±0.0020 |
| **Focal Modulation** | _0.0266_ ±0.0003 | _0.0350_ ±0.0004 | _0.8808_ ±0.0014 | _0.9454_ ±0.0009 | **0.9253** ±0.0016 |
| BT (K) | | | | | |
| Algorithm | MAE($\downarrow$) | RMSE($\downarrow$) | $R^2$($\uparrow$) | $R_p$($\uparrow$) | $R_s$($\uparrow$) |
| Climatology 1981-1988 | 2.8806 | 3.6864 | 0.8447 | 0.9485 | 0.9470 |
| Climatology 1989-2016 | 2.2024 | 2.8880 | 0.9036 | 0.9623 | 0.9606 |
| U-Net | 1.9920 ±0.0148 | 2.6652 ±0.0262 | 0.9164 ±0.0021 | 0.9644 ±0.0009 | 0.9616 ±0.0005 |
| Wave-MLP | _1.9376_ ±0.0184 | _2.6221_ ±0.0177 | 0.9172 ±0.0005 | 0.9647 ±0.0005 | 0.9619 ±0.0008 |
| Swin Transformer V1 | 1.9563 ±0.0329 | 2.6381 ±0.0397 | 0.9169 ±0.0038 | _0.9649_ ±0.0009 | _0.9627_ ±0.0008 |
| Swin Transformer V2 | 1.9516 ±0.0639 | 2.6277 ±0.0874 | _0.9183_ ±0.0060 | 0.9641 ±0.0025 | 0.9619 ±0.0020 |
| **Focal Modulation** | **1.9179** ±0.0458 | **2.5745** ±0.0470 | **0.9204** ±0.0030 | **0.9664** ±0.0007 | **0.9636** ±0.0006 |

**Comment RC1.10** *I suggest combining section 8 and 9 as one section (conclusions and limitations). The current discussion section appears only limitations not discussion.*

**Answer to RC1.10** We will combine the sections.

**Minor Comments**

**Comment RC1.1** *At the beginning of abstract, I suggest adding one or two sentences about research problem the authors would like to solve.*

**Answer to RC1.1** We will revise the abstract accordingly.

**Comment RC1.2** *AVHRR in line 44 firstly appear without full name and also check for other short names.*

**Answer to RC1.2** We will check the abbreviations and adjust the manuscript accordingly.

**Comment RC1.3** *Line 50: change synthesis to be 'synthesize'.*

**Answer to RC1.3** We will correct it.

**Comment RC1.4** *Line 195: What is DA represented? If the term only shows once in the manuscript, it is not necessary to use the short name.*

**Answer to RC1.4** DA refers to data assimilation. We will replace DA by the full name.

**Comment RC1.5** *Line 200: theta and lambda represents?*

**Answer to RC1.5** Theta and Lambda refer to the coordinates of the rotated pole. We will rewrite the terms as follows: "... The grid specification for TSMP is a standardized rotated coordinate system ($\phi_{(rotated\ pole)} = 39.5°$ N, $\lambda_{(rotated\ pole)} = 18°$ E) with a spatial resolution ..."

**Comment RC1.6** *Line 253: Is it equation (5) is for NDVI for AVHRR instead of VIIRS as stated in line 253?*

**Answer to RC1.6** Thank you for this notice. Equation (5) computes NDVI for the AVHRR system. Lines 252-253 should be "... re-composite NDVI for AVHRR ...". We will correct it in the revised manuscript.

**Comment RC1.7** *Figure 1: It is better to use TSMP instead of TerrSysMP since TSMP is used all over the manuscript.*

**Answer to RC1.7** We will change the figure accordingly.

**Comment RC1.8** *Line 325: it should have a period between 'efficiently In'.*

**Answer to RC1.8** We will correct it.

**Comment RC1.9** *Line 368-369: it is not clear why the third gate focuses on the water area? How do you know that?*

**Answer to RC1.9** The bright colors for specific regions in Fig. 3 represent higher values, which correspond to higher attentions of the model to that regions. For $G_3^k$, the attention values are high at areas with water. We will revise lines 368-369 to make it clear.

**Comment RC1.10** *Add references on line 430.*

**Answer to RC1.10** We will add the following references [2–5].

**Comment RC1.11** *Line 448-450: the sign N(0, I =0.02) is not clear. It is better to use normal distribution with zero mean and standard deviation.*

**Answer to RC1.11** We will replace the terms as suggested.

**Comment RC1.12** *Line 461: reported how long it takes for the focal model, what about other DL models?*

**Answer to RC1.12** In Table B, we report the inference time to generate one sample for NDVI and BT containing 397×409×2 grid points. We will include the table.

Table B: Inference time in seconds for different DL models.

| Algorithm | GPU[1] | CPU[2] |
|---|---|---|
| U-Net | 0.09 ±0.02 | 5 ±0.2 |
| Wave-MLP | 0.28 ±0.00 | 10 ±0.3 |
| Swin Transformer V1 | 0.18 ±0.00 | 11 ±0.2 |
| Swin Transformer V2 | 0.19 ±0.00 | 11 ±0.2 |
| Focal Modulation | 0.24 ±0.01 | 12 ±0.1 |

[1]NVIDIA GeForce RTX 3090 GPU
[2]AMD Ryzen 9 3900X 12-Core CPU

**Comment RC1.13** *Line 471-472: What is these non-ML baselines? It is not clear for this statement.*

**Answer to RC1.13** The non-ML baselines represent the climatology. We will rewrite the sentience to make it clearer in the revised manuscript.

**Comment RC1.14** *Line 516: change 'fro' to 'for'.*

**Answer to RC1.14** We will correct it.

**Comment RC1.15** *Figure 10: change 'sprint' to 'spring' in the left two plots in the first row.*

**Answer to RC1.15** We will correct it.

**References**

[1] D. M. Lawrence, R. A. Fisher, C. D. Koven, K. W. Oleson, S. C. Swenson, G. Bonan, N. Collier, B. Ghimire, L. van Kampenhout, D. Kennedy, E. Kluzek, P. J. Lawrence, F. Li, H. Li, D. Lombardozzi, W. J. Riley, W. J. Sacks, M. Shi, M. Vertenstein, W. R. Wieder, C. Xu, A. A. Ali, A. M. Badger, G. Bisht, M. van den Broeke, M. A. Brunke, S. P. Burns, J. Buzan, M. Clark, A. Craig, K. Dahlin, B. Drewniak, J. B. Fisher, M. Flanner, A. M. Fox, P. Gentine, F. Hoffman, G. Keppel-Aleks, R. Knox, S. Kumar, J. Lenaerts, L. R. Leung, W. H. Lipscomb, Y. Lu, A. Pandey, J. D. Pelletier, J. Perket, J. T. Randerson, D. M. Ricciuto, B. M. Sanderson, A. Slater, Z. M. Subin, J. Tang, R. Q. Thomas, M. Val Martin, and X. Zeng, "The community land model version 5: Description of new features, benchmarking, and impact of forcing uncertainty," *Journal of Advances in Modeling Earth Systems*, vol. 11, no. 12, pp. 4245–4287, 2019. [Online]. Available: https://agupubs.onlinelibrary.wiley.com/doi/abs/10.1029/2018MS001583

[2] L. Wang, R. Li, C. Duan, C. Zhang, X. Meng, and S. Fang, "A novel transformer based semantic segmentation scheme for fine-resolution remote sensing images," *IEEE Geoscience and Remote Sensing Letters*, vol. 19, pp. 1–5, 2022.

[3] L. Gao, H. Liu, M. Yang, L. Chen, Y. Wan, Z. Xiao, and Y. Qian, "Stransfuse: Fusing swin transformer and convolutional neural network for remote sensing image semantic segmentation," *IEEE Journal of Selected Topics in Applied Earth Observations and Remote Sensing*, vol. 14, pp. 10 990–11 003, 2021.

[4] L. Wang, R. Li, C. Zhang, S. Fang, C. Duan, X. Meng, and P. M. Atkinson, "Unetformer: A unet-like transformer for efficient semantic segmentation of remote sensing urban scene imagery," *ISPRS Journal of Photogrammetry and Remote Sensing*, vol. 190, pp. 196–214, 2022. [Online]. Available: https://www.sciencedirect.com/science/article/pii/S0924271622001654

[5] A. A. Aleissaee, A. Kumar, R. M. Anwer, S. Khan, H. Cholakkal, G.-S. Xia, and F. S. Khan, "Transformers in remote sensing: A survey," *Remote Sensing*, vol. 15, no. 7, 2023. [Online]. Available: https://www.mdpi.com/2072-4292/15/7/1860

---

## Author Comment (AC3)

**Answers to Anonymous Referee #2**

We thank the anonymous referee for reviewing the manuscript and for the valuable comments. We will revise the manuscript according to the suggestions. Below are the comments and our detailed responses.

**Major Comment**

**Comment RC2.1** *The abstract lacks some context. Not many readers will know TSMP, so it needs to be pointed out more clearly (and earlier) what the applications of this model system are and, then, why it is important to generate pseudo observations of NDVI and BT from simulation output, when real Earth observations are available for these quantities. There are some hints at the end of the abstract (climate simulations), but this remains vague and doesn't help to understand why this study was performed. A side aspect of this is that this lack of claritiy makes it difficult to evaluate the stated model errors. Are MAEs of 0.027 for NDVI and 1.9 K for BT good or better than SOTA? What would be the reference here? Retrieval errors?*

**Answer to RC2.1** We will revise the abstract accordingly. Please note that we do not aim to generate pseudo observations of NDVI/BT when satellite measurements are available but rather to predict NDVI/BT for periods where no satellite observations exist (future periods). However, we have to train and evaluate the model on a historical period. The real application is then to apply DL to a climate projection and predict NDVI/BT in the future. We agree that the stated model errors are hard to be interpreted without a reference. However, this is only mentioned to give an estimation about the accuracy that the model could achieve. The errors for satellite products highly depend on the spatial/temporal resolution and sensor being used. To the best of our knowledge, there are also no baselines models to compare with.

**Comment RC2.2** *Section 1 takes a couple of short-cuts and doesn't always provide good explanations to motivate this study. Please see detailed comments below. Related to this, in section 2 it is unclear why, for example, radiative transfer models are discussed here, and some of the content of this section would better belong elsewhere.*

**Answer to RC2.2** As suggested, we will revise section 1 and section 2.

**Comment RC2.3** *Sections 3-6 are largely OK, except for minor comments listed below.*

**Answer to RC2.3** We will revise sections 3-6 accordingly.

**Comment RC2.4** *Section 7 variable importance: I like this analysis very much. However, I think one could point out that channel importance does not "explain" everything. First: if two variables are correlated, the network may decide to focus on one of them and the other one would seem unimportant, while it could provide almost the same information if it were alone. Second: seemingly "unimportant" variables may play an important role to get the final few percent accuracy out of the models. This could of course be tested by training a model on only the N most relevant variables and compare the results.*

**Answer to RC2.4** We are aware that conclusions from such an analysis should be taken carefully as discussed in lines 654-663. We will move this discussion to Section 7.

**Comment RC2.5** *Section 8: this discussion comes as a surprise as it goes much deeper into remote sensing and modelling issues than any of the other parts of the paper. As indicated above, there is lack of information in the Introduction and related work sections. I therefore suggest to re-arrange some of the text and use some of the material of the discussion in these earlier sections. The discussion could then be shortened and focus more on the applicability and prospects of the new method.*

**Answer to RC2.5** As suggested, we will move parts of section 8 to earlier sections.

**Minor Comments**

**Comment RC2.1** *Abstract l.3: why "intermediate step"? The image synthesis is the main product of the DL model, not an intermediate step in the modelling itself. The derivation of various indices is post-processing. Suggest to remove "in an intermediate step".*

**Answer to RC2.1** We will remove it.

**Comment RC2.2** *l.7: suggest rewording "... to assess the model's applicability to different seasons and regions..."*

**Answer to RC2.2** We will revise it accordingly.

**Comment RC2.3** *l.12 the unit of temperature is K, not K°.*

**Answer to RC2.3** Thank you for this notice. We will check the manuscript and remove the symbol °.

**Comment RC2.4** *Introduction l.20: Suggest to remove the first sentence (motherhood statement) and integrate "under a changing climate" in the following sentence, which provides a more concise and precise start of the text. Not all droughts are extreme, and while extreme events are a good motivation, this study does not focus on extreme events, but rather tries to provide information to assess droughts or the risk of droughts.*

**Answer to RC2.4** We will revise the introduction accordingly.

**Comment RC2.5** *l.32 delete "in the future"*

**Answer to RC2.5** We agree as it looks redundant.

**Comment RC2.6** *l.33 ff. The link made here between climate models and the water cycle is a bit too direct. It is a known weakness of general circulation models (aka "climate models") that convection and rainfall are not well captured over many world regions. This is why there is a need for more specialized hydrology models, which are frequently used for regional instead of global simulations. Also, the introduction of drought indices comes somewhat unmotivated. The rationale behind these is usually to convert information from some instrument (or model) into a meaningful*

quantity that can be used to assess the state of some ecosystem or the climate system. Why focus on agricultural indices here? If this is intended, then this should be stated in the first motivation sentences for this study.

**Answer to RC2.6** We will revise the introduction accordingly as this appears confusing. The study is primarily related to remote sensing based agricultural drought events.

**Comment RC2.7** *l.50-55 The "discussion" about retrievals is good and can be used for the motivation of this study, but it is missing an explicit reference to retrieval errors. The problems with current retrievals (or perhaps even fundamental problems = theoretiscal limitations of physics-based retrievals?) should be described more precisely and with some more detail.*

**Answer to RC2.7** We will revise the introduction to make it clearer. Please note that the study is not mainly related to retrieval errors.

**Comment RC2.8** *l.66 I would avoid the word downstream-application here (even though it is technically correct) and rather formulate "To showcase the value (or potential) or our approach, we calculate (or derive) ..."*

**Answer to RC2.8** We will revised it according to the suggestion.

**Comment RC2.9** *l.74 This sentence is confusing in the context. Before, you give the impression that you rely on the model (implicitly assuming TSMP is perfect), whereas you now state that you can use the derived products to "examine the predicitve capability" of the model. As stated in the major comment above, the Introduction needs to be rewritten with a clearer explanation what this study is based on, how it is motivated (what doesn't work well at present?) and what are its primary objectives. Certainly, the aspect of model errors and their impact - or the potential of the method to quantify them - are one very relevant aspect that should at least shine through in the Introduction.*

**Answer to RC2.9** Thank you for this comment. We will revise the introduction accordingly.

**Comment RC2.10** *Section 2.1: the review of radiative transfer models is OK, but the reader doesn't understand why there is half a page or more on cloud retrievals when this paper is about vegetation indices. It would be helpful to add an introductional sentence or two explaining why section 2 is structured in the way it is and what content is expected. The discussion in 2.1 is perhaps a little too detailed.*

**Answer to RC2.10** We will merge the first two sections and shorten the discussion.

**Comment RC2.11** *l.118: The paragraph introducing the work of this study does not seem to connect to the general radiative transfer discussion above. The connection appears to be only methodologically (use of AI).*

**Answer to RC2.11** Section 2 will be revised and integrated with the introduction.

**Comment RC2.12** *l.132 grammar "the interaction ... exhibits ... behavior".*

**Answer to RC2.12** We will correct it.

**Comment RC2.13** *l.162 awkward phrasing "a single indicator like NDVI excluding BT" - do you mean "... either NDVI or BT"? Or simply cut after "indicator".*

**Answer to RC2.13** We mean when only relying on NDVI. We will rewrite the sentence.

**Comment RC2.14** *l.162 ff. After reading section 2, it becomes clearer what this paper aims to do. Some of the text here should be moved to the Introduction, and section 2 should no longer explain what is done in this study, but concentrate on discussing what has been made available so far.*

**Answer to RC2.14** We will merge parts of section 2 with the introduction.

**Comment RC2.15** *l.178 delete "at" before "IBG-3 institute".*

**Answer to RC2.15** We will delete it.

**Comment RC2.16** *l.180 "near nature realization" - what do you mean by this? Every model is an abstraction of some sort, and many models aim to produce realistic results. However, this expression is not scientific.*

**Answer to RC2.16** We will remove it.

**Comment RC2.17** *l.182 ff. please harmonize grammar in the bullet list - some bullets have verbs others don't.*

**Answer to RC2.17** We will revise it.

**Comment RC2.18** *l.190 "a dynamic equilibrium" - this is not unambiguous and depends on the choice of start and end date, for example.*

**Answer to RC2.18** The dynamic equilibrium with the atmosphere (1979-1989) was obtained to initialize the subsurface and surface hydrologic and energy variables [1]. We will make this clearer in the revised manuscript.

**Comment RC2.19** *l.195 comment, related to l.180: a free running model without DA will always be further away from "nature" than a model with DA.*

**Answer to RC2.19** This is true. The simulation did not use any re-initialization or nudging. We will remove DA.

**Comment RC2.20** *l.198 extending.*

**Answer to RC2.20** It will be corrected.

**Comment RC2.21** *l.199 why mention "with various vegetation types and climate conditions"? If you refer to Europe as a region, then this is kind of obvious and doesn't add information to*

*this sentence describing the model set-up. If this refers instead to a property of the model or model output, then it doesn't belong here, but in a section where you describe the data and data distributions.*

**Answer to RC2.21** We wanted to emphasize that the extension of the region we study (Europe) includes different vegetation types and climate conditions. We agree this seems obvious and we will delete it.

**Comment RC2.22** *l.201 and \*the\* model set-up.*

**Answer to RC2.22** It will be corrected.

**Comment RC2.23** *l.204 I think you could add an extra sentence to say that DL has already been applied to TSMP simulations, instead of just referring to the papers.*

**Answer to RC2.23** We will revise it accordingly.

**Comment RC2.24** *l.221 what are upper and lower bounds of an ecosystem? Do you mean bounds of NDVI and BT for a specific ecosystem class? Also, the sentence "Consequently, ..." doesn't fit well. Better to write "Hence, ..." or "Thus, ..." or "Therefore, ..."*

**Answer to RC2.24** The upper and lower bounds are the min and max values for NDVI and BT for a specific pixel. We will make this clearer and rephrase it.

**Comment RC2.25** *l.230 Is alpha a fixed coefficient or does it vary with ecosystem class or other parameters?*

**Answer to RC2.25** In our work, we used a standard value $\alpha = 0.5$ [2]. Please note that this does not affect NDVI and BT predictions. As mentioned in lines 267-269 and 637-639, it is better to calibrate $\alpha$ with respect to the location (see [3, 4]). However, this is beyond the scope of our work to improve the weighting.

**Comment RC2.26** *l.234 delete "Moreover,".*

**Answer to RC2.26** It will be deleted.

**Comment RC2.27** *l.248-254 grammar (OK, but clearly non-native English).*

**Answer to RC2.27** We will proofread the section.

**Comment RC2.28** *l.271 Please describe the data cube dimensions. Is there one datacube with (time, lat, lon) for each variable? Remove "observed".*

**Answer to RC2.28** We stored a datacube for each week (variable, lat, lon). We will clarify this. "observed" will be removed.

**Comment RC2.29** *l.275 zero*

**Answer to RC2.29** It will be corrected.

**Comment RC2.30** *l.282 the theta doesn't belong in eq 7, which describes the mapping objective. It only comes in when you in fact use a model, i.e. when you describe the U-net.*

**Answer to RC2.30** We will reorder the text and put eq 7 after we mention the model.

**Comment RC2.31** *l.300 number of channel\*s\*.*

**Answer to RC2.31** It will be corrected.

**Comment RC2.32** *l.325 period missing. And: pixel representations.*

**Answer to RC2.32** It will be corrected.

**Comment RC2.33** *l.328 Please be more specific: you refer to the quadratic scaling, which primarily limits the attention span, but not "applications" per se.*

**Answer to RC2.33** You are right. We refer to the quadratic computation complexity of the self-attention. We will make this clearer.

**Comment RC2.34** *l.337 input channel\*s\*.*

**Answer to RC2.34** It will be corrected.

**Comment RC2.35** *l.347 remove "a" before "one".*

**Answer to RC2.35** It will be corrected.

**Comment RC2.36** *l.350 reduce the number of model parameters.*

**Answer to RC2.36** We will revise it as suggested.

**Comment RC2.37** *eq.11 I suggest to replace the somewhat clumsy expressions FocalModulation-Block etc. by shorter variable names which then need to be defined in the text, of course. This would improve readability of the equation.*

**Answer to RC2.37** We will simplify the terms.

**Comment RC2.38** *l.379 "less" compared to what? I assume you mean MSE loss.*

**Answer to RC2.38** Correct. We mean here less than MSE. We will make this explicit in the revised manuscript.

**Comment RC2.39** *l.392 play \*a\* more important role.*

**Answer to RC2.39** It will be corrected.

**Comment RC2.40** *l.425 I think it would be easier to describe the U-net baseline model by simply stating what it consists of instead of "reverse engineering" it by abstracting the focal attention*

*blocks away.*

**Answer to RC2.40** We will revise the description.

**Comment RC2.41** *l.428 ff Please provide a few more details on the competitor models, such as number of layers, size of attention matrix etc. This could also be summarized in an Appendix, which should then be referenced here.*

**Answer to RC2.41** All baseline models share the number of layers but differ in the type of layers. The U-Net model does not use an attention mechanism, but follows the original U-Net design using 2D convolutions. Swin Transformer uses self-attention inside local windows. The implementation details about the models are mentioned in section 5 "Implementation details".

**Comment RC2.42** *l.438 Do you mean "Apart from"?*

**Answer to RC2.42** Yes. We will correct it.

**Comment RC2.43** *l.442 Why is the second climatology computing the future? 2016 is in the past. Also, grammar: future should be singular.*

**Answer to RC2.43** The main application of the study is to estimate the vegetation condition using a DL-based model for future periods where no satellite images are available. We compare the results of the DL models to two NDVI/BT climatologies from remote sensing observations. The first climatology (1981-1988) is used to show the limitation of using prescribed satellite phenology for future projection and we argue that a DL-based model maybe a better replacement (for example future simulations often prescribe vegetation condition in a satellite phenology-mode neglecting the inter-annual variability [5]). This is still not a fair comparison because the DL-models were trained on periods succeeding this climatology. The second climatology (1989-2016) is used to show that the DL-models still outperform the climatology. We could compute the second climatology (1989-2016) because we used a climate simulation in an overlapping period with historical remote sensing observations. This comparison is not possible for future climate projections since no satellite observations exist yet. We will rewrite lines 438-444 to make it clear.

**Comment RC2.44** *l.450 randomly perturbing.*

**Answer to RC2.44** It will be corrected.

**Comment RC2.45** *l.451 this seems to contradict the preprocessing description in section 3. There you wrote that samples were averaged over a week, which implies that \*all\* samples are used in thre average. What is written here is a random estimator of the weekly average based on two days.*

**Answer to RC2.45** We use this as an augmentation technique only during training. For validation and test, we average all days. We will make this clearer in the revised manuscript.

**Comment RC2.46** *l.460 typo finally.*

**Answer to RC2.46** It will be corrected.

**Comment RC2.47** *l.466 remove comma.*

**Answer to RC2.47** It will be corrected.

**Comment RC2.48** *l.475 I don't understand the reference to radiative transfer models here and suggest it be removed.*

**Answer to RC2.48** We will remove it as suggested.

**Comment RC2.49** *l.484 As shown.*

**Answer to RC2.49** It will be corrected.

**Comment RC2.50** *l.486 shown \*in\* Figs...*

**Answer to RC2.50** It will be corrected.

**Comment RC2.51** *l.493 weakness\*es\*.*

**Answer to RC2.51** It will be corrected.

**Comment RC2.52** *l.505 This discussion on errors could go one step further. While I agree that a full error attribution may be beyond the scope of this paper, you could at least give some indications, which errors come from the TSMP input data, which are from the observations and which may be DL model errors. This would be especially relevant for the TSMP data. For example, by showing the variability in different 2-day "weekly" samples and how this translates into different DL model outputs. Or you could apply some systematic perturbations to the TSMP output, thereby roughly correcting known model biases, and then see how this change the results. (OK, l.537 provides at least some indication already, and you could also refer to section 7 for additional insights).*

**Answer to RC2.52** We will extend the discussion and refer to section 7 as suggested.

**Comment RC2.53** *l.516 typo "for".*

**Answer to RC2.53** It will be corrected.

**Comment RC2.54** *l.540 , who showed.*

**Answer to RC2.54** It will be corrected.

**Comment RC2.55** *l.541 affecting.*

**Answer to RC2.55** It will be corrected.

**Comment RC2.56** *l.723 Klaus Goergen.*

**Answer to RC2.56** It will be corrected.

**References**

[1] C. Furusho-Percot, K. Goergen, C. Hartick, K. Kulkarni, J. Keune, and S. Kollet, "Pan-european groundwater to atmosphere terrestrial systems climatology from a physically consistent simulation," *Scientific Data*, vol. 6, no. 1, p. 320, 2019. [Online]. Available: https://doi.org/10.1038/s41597-019-0328-7

[2] W. Yang, F. Kogan, and W. Guo, "An ongoing blended long-term vegetation health product for monitoring global food security," *Agronomy*, vol. 10, no. 12, 2020. [Online]. Available: https://www.mdpi.com/2073-4395/10/12/1936

[3] J. Zeng, R. Zhang, Y. Qu, V. A. Bento, T. Zhou, Y. Lin, X. Wu, J. Qi, W. Shui, and Q. Wang, "Improving the drought monitoring capability of vhi at the global scale via ensemble indices for various vegetation types from 2001 to 2018," *Weather and Climate Extremes*, vol. 35, p. 100412, 2022. [Online]. Available: https://www.sciencedirect.com/science/article/pii/S2212094722000068

[4] J. Zeng, T. Zhou, Y. Qu, V. Bento, J. Qi, Y. Xu, Y. Li, and Q. Wang, "An improved global vegetation health index dataset in detecting vegetation drought," *Scientific Data*, vol. 10, p. 338, 05 2023.

[5] D. M. Lawrence, R. A. Fisher, C. D. Koven, K. W. Oleson, S. C. Swenson, G. Bonan, N. Collier, B. Ghimire, L. van Kampenhout, D. Kennedy, E. Kluzek, P. J. Lawrence, F. Li, H. Li, D. Lombardozzi, W. J. Riley, W. J. Sacks, M. Shi, M. Vertenstein, W. R. Wieder, C. Xu, A. A. Ali, A. M. Badger, G. Bisht, M. van den Broeke, M. A. Brunke, S. P. Burns, J. Buzan, M. Clark, A. Craig, K. Dahlin, B. Drewniak, J. B. Fisher, M. Flanner, A. M. Fox, P. Gentine, F. Hoffman, G. Keppel-Aleks, R. Knox, S. Kumar, J. Lenaerts, L. R. Leung, W. H. Lipscomb, Y. Lu, A. Pandey, J. D. Pelletier, J. Perket, J. T. Randerson, D. M. Ricciuto, B. M. Sanderson, A. Slater, Z. M. Subin, J. Tang, R. Q. Thomas, M. Val Martin, and X. Zeng, "The community land model version 5: Description of new features, benchmarking, and impact of forcing uncertainty," *Journal of Advances in Modeling Earth Systems*, vol. 11, no. 12, pp. 4245–4287, 2019. [Online]. Available: https://agupubs.onlinelibrary.wiley.com/doi/abs/10.1029/2018MS001583

---

## Author Response (AR2)

We thank the topic editor for coordinating the review process of our work. We also thank the anonymous referees for reviewing the manuscript and for their valuable comments and time. We revised the manuscript according to the suggestions. Below are the referees' comments and our detailed responses.

**Answers to Anonymous Referee #1**

**Major Comment**

**Comment RC1.1** *The introduction lacks rationales on conducting the study. I suggest integrating sections 2.1 and 2.2 into the introduction. In the introduction, I would like to emphasize that the authors need to summarize what previous studies did, not describe what they did one by one (first paragraph in section 2.1 and first paragraph in section 2.2, which are not very readable and not necessary), but summarize and provide rationales and research gaps that this study will fill.*

**Answer to RC1.1** We have revised the introduction (see lines 33-75 in the revised manuscript). We have integrated the discussion (lines 606-616 in the old manuscript) with the introduction (lines 45-54 in the revised manuscript). In addition, we have integrated the related works section (sections 2.1 and 2.2) with the introduction. More specifically, we have shortened the related works on radiative transfer models from section 2.1 and integrated them with the introduction (see lines 76-81 and 104-109 in revised manuscript). The related works from section 2.2 have been also shortened, summarized and integrated with the introduction (lines 83-104 and 109-114 in the revised manuscript).

**Comment RC1.2** *Lines 190-192: what is the rationale on variable selection?*

**Answer to RC1.2** Only variables within the period applicable for analysis were selected. Some variables were not selected due to the reason that for some variables in the model, the full groundwater states equilibrium starts after 1995. In addition, we did not select the variables that do not cover the whole period of the simulation (see lines 146-147 in the revised manuscript).

**Comment RC1.3** *Section 4.1: the authors need to provide rationale and advantages about the proposed architecture at the beginning of this section compared to state of art architectures (information in lines 322 to 333 with adjustments should be provided at the beginning of section 4.1). I also suggest using jargons as less as possible in the methodology section.*

**Answer to RC1.3** We revised section 4.1 accordingly. In particular, we have moved the discussion (lines 322-333 in the old manuscript) into the beginning of section model architectures (see lines 244-258 in the revised manuscript). We have also simplified some terms in the Eq. (11) to improve the readability.

**Comment RC1.4** *I suggest that lines 403-464 should be in the methodology section as baseline approaches and evaluation metrics. I also suggest the italics section in section 5 to be separated section as 5.1, 5.2 and so on, which will be much clear.*

**Answer to RC1.4** We revised the manuscript accordingly (see sections 3.4-3.6 in the revised manuscript). In particular, we have moved lines 422-437 into a new subsection 3.4 (lines 357-571 in the revised manuscript) and modified lines 438-444 in the old manuscript and moved them into the end of the new subsection 3.4 (lines 372-377 in the revised manuscript). We have also moved lines 445-461 in the old manuscript into a new subsection 3.5 (lines 379-396 in the revised manuscript). We have added a new subsection 3.6 where we moved lines 403-421 from the old manuscript into the new subsection 3.6 (lines 398-416 in the revised manuscript).

**Comment RC1.5** *It is not clear what the authors would like to emphasize in lines 438-444.*

**Answer to RC1.5** The main application of the study is to estimate the vegetation condition using a DL-based model for periods where no satellite images are available. As a baselines, we compare the results of DL models to two NDVI/BT climatologies from remote sensing observations (the climatology is computed pixel-wise and on a weekly basis). The climatology has the limitation that the inter-annual variability in NDVI/BT is neglected as average cycles are used. The first climatology (1981-1988) is used to show the limitation of using prescribed satellite phenology for future projection and we argue a DL-based model maybe a better replacement (for example future simulations often prescribe vegetation condition in a satellite phenology-mode neglecting the inter-annual variability [1]). This is still not a fair comparison because the DL-models were trained on periods succeed this climatology. The second climatology (1989-2016) is used to show that the DL-models still outperform the climatology and generalized beyond the mean annual cycles of NDVI/BT. We could compute the second climatology (1989-2016) because we used a climate simulation in an overlapped period with historical remote sensing observations. This comparison is not possible for future climate projections since no satellite observations exist yet. Please note that we excluded the validation/testing (2010-2012) years for the computation of this second climatology. The last part is that we could not compare to a specific physically-based model since such a model has to construct the albedo and emission for visible, near-infrared, and infrared bands under all conditions and for all surface types and then trace the atmospheric correction (NOAA AVHRR depends on temporal compositing for the atmospheric correction) to the top of the atmosphere regardless of the solar and satellite geometry. We have rewritten lines 438-444 (see lines 372-377 in the revised manuscript).

**Comment RC1.6** *It is not clear in line 451. Based on the methodology (lines 285-289), both the input variables and outputs are in weekly time scale, why here claims average two days?*

**Answer to RC1.6** We use this as an augmentation technique only during training. For validation and test, we average all days. We made this clearer in the revised manuscript (see line 386 in the revised manuscript)

**Comment RC1.7** *The results are not well organized and too many sections. I suggest organizing all the results into one section as subsections for each topic*

**Answer to RC1.7** We revised the manuscript accordingly. Instead of having sections 5, 6 and 7 as separated sections, we reorganized them under one section called *Experimental results and analysis* (section 4 in the revised manuscript). More precisely, section 5 is now named subsection 4.1, and section 6 is named subsection 4.2 in the revised manuscript. Section 7 is now named subsection 4.3 in the revised manuscript.

**Comment RC1.8** *Lines 467-471: it is confusing about different climatology. Based on the dataset information in section 3.1, the model simulation is from 1989 to 2019. What does climatology from 1981 to 1988 come from? What is the main point for analyzing different climatology (climate change?)? Table 1 is also confusing for mixing climatology different periods and validation/test different years.*

**Answer to RC1.8** Please see the answer to the major comment RC1.5. We agree that the comparison to climatology in tables may appear confusing. We made it clearer in the suggested new section "baseline approaches" and changed the name of climatology (1981-1988) to climatology I and climatology (1989-2016) to climatology II. For validation/test, we took the last 3 years of AVHRR (2010-2012) and the last 3 years of converted VIIRS (2017-2019).

**Comment RC1.9** *Based on Table 1, the metric differences for different DL models are trivial (mostly around percentage scale), although the DL models have very different building blocks. The proposed DL model is also not consistent better than others particularly for the test dataset. My doubt is whether the trivial differences are caused by stochasticity not due to DL model itself. The authors claimed fixed random seed for reproductivity, but what if the seed is not fixed and what does the results look like for running the model several times? Will the results be consistent with any conclusion got from the table?*

**Answer to RC1.9** Table A reports the mean and standard deviation for training with 3 different random seeds. We have replaced Table 1 with the results from the new experiments. In general, there are some differences for the results on the test set and we adjusted our comments accordingly in the revised manuscript. We would also like to note that all of the described models are our implementation and we found Focal Modulation among them to work overall the best. We also replaced the name 2D CNN with U-Net to be consistent through out the manuscript (see Tables 1 and 2 in the revised manuscript).

**Comment RC1.10** *I suggest combining section 8 and 9 as one section (conclusions and limitations). The current discussion section appears only limitations not discussion.*

**Answer to RC1.10** We deleted section 8 and integrated it with the other sections. More specific, we moved the lines 617-620 into the introduction (lines 42-45 in the revised manuscript). We have integrated the discussion (lines 606-616 in the old manuscript) with the introduction (lines 45-54 in the revised manuscript). Lines 624-628 were integrated into the experimental results and analysis section (subsection 4.1 lines 470-474 in the revised manuscript). Lines 629-637 were also integrated into the experimental results and analysis section (subsection 4.1 lines 463-470 in the revised manuscript). Lines 639-642 were integrated into the experimental results and analysis section (section 4.2 lines 503-506 in the revised manuscript). Furthermore, lines 643-648 were shortened and integrated into the conclusions and outlook section (section 5 lines 591-593 in the revised manuscript). Finally, the discussion in lines 656-663 was integrated into the experimental results and analysis section (subsection 4.3 lines 570-578 in the revised manuscript).

**Minor Comments**

**Comment RC1.1** *At the beginning of abstract, I suggest adding one or two sentences about*

Table A: Comparing the performance of different DL models on the validation and test sets.

| Validation - Years (2010, 2011, 2017) - 156 weeks | | | | | |
|---|---|---|---|---|---|
| NDVI | | | | | |
| Algorithm | MAE($\downarrow$) | RMSE($\downarrow$) | R$^2$($\uparrow$) | R$_p$($\uparrow$) | R$_s$($\uparrow$) |
| Climatology 1981-1988 | 0.0550 | 0.0680 | 0.5763 | 0.8939 | 0.8669 |
| Climatology 1989-2016 | 0.0326 | 0.0416 | 0.8372 | 0.9353 | 0.9113 |
| U-Net | 0.0277 ±0.0001 | 0.0365 ±0.0002 | 0.8743 ±0.0008 | 0.9406 ±0.0005 | 0.9172 ±0.0005 |
| Wave-MLP | 0.0272 ±0.0003 | **0.0358** ±0.0003 | 0.8784 ±0.0018 | 0.9422 ±0.0018 | 0.9183 ±0.0021 |
| Swin Transformer V1 | 0.0273 ±0.0003 | 0.0362 ±0.0003 | 0.8759 ±0.0022 | 0.9411 ±0.0013 | 0.9161 ±0.0023 |
| Swin Transformer V2 | 0.0277 ±0.0003 | 0.0369 ±0.0003 | 0.8703 ±0.0021 | 0.9415 ±0.0010 | 0.9167 ±0.0008 |
| **Focal Modulation** | **0.0269** ±0.0001 | **0.0358** ±0.0002 | **0.8790** ±0.0017 | **0.9432** ±0.0001 | **0.9194** ±0.0009 |
| BT (K) | | | | | |
| Algorithm | MAE($\downarrow$) | RMSE($\downarrow$) | R$^2$($\uparrow$) | R$_p$($\uparrow$) | R$_s$($\uparrow$) |
| Climatology 1981-1988 | 2.9130 | 3.7302 | 0.8454 | 0.9466 | 0.9408 |
| Climatology 1989-2016 | 2.3017 | 3.0020 | 0.8963 | 0.9601 | 0.9539 |
| U-Net | 1.9377 ±0.0093 | 2.6067 ±0.0057 | 0.9243 ±0.0014 | 0.9667 ±0.0004 | 0.9603 ±0.0007 |
| Wave-MLP | 1.9200 ±0.0491 | 2.5834 ±0.0486 | 0.9248 ±0.0035 | 0.9668 ±0.0006 | 0.9603 ±0.0007 |
| Swin Transformer V1 | 1.9642 ±0.0246 | 2.6341 ±0.0303 | 0.9221 ±0.0012 | 0.9661 ±0.0005 | 0.9590 ±0.0006 |
| Swin Transformer V2 | 1.9741 ±0.0191 | 2.6420 ±0.0258 | 0.9225 ±0.0013 | 0.9659 ±0.0011 | 0.9590 ±0.0014 |
| **Focal Modulation** | **1.9010** ±0.0071 | **2.5364** ±0.0073 | **0.9280** ±0.0012 | **0.9679** ±0.0001 | **0.9614** ±0.0007 |
| Test - Years (2012, 2018, 2019) - 139 weeks | | | | | |
| NDVI | | | | | |
| Algorithm | MAE($\downarrow$) | RMSE($\downarrow$) | R$^2$($\uparrow$) | R$_p$($\uparrow$) | R$_s$($\uparrow$) |
| Climatology 1981-1988 | 0.0567 | 0.0697 | 0.5529 | 0.8933 | 0.8704 |
| Climatology 1989-2016 | 0.0314 | 0.0400 | 0.8507 | 0.9433 | 0.9254 |
| U-Net | 0.0274 ±0.0004 | 0.0359 ±0.0005 | 0.8772 ±0.0006 | 0.9435 ±0.0006 | 0.9237 ±0.0009 |
| Wave-MLP | **0.0261** ±0.0006 | **0.0343** ±0.0008 | **0.8861** ±0.0043 | **0.9467** ±0.0024 | 0.9252 ±0.0011 |
| Swin Transformer V1 | 0.0269 ±0.0003 | 0.0355 ±0.0004 | 0.8795 ±0.0029 | 0.9442 ±0.0010 | 0.9239 ±0.0014 |
| Swin Transformer V2 | 0.0270 ±0.0005 | 0.0359 ±0.0005 | 0.8766 ±0.0038 | 0.9447 ±0.0012 | 0.9251 ±0.0020 |
| **Focal Modulation** | 0.0266 ±0.0003 | 0.0350 ±0.0004 | 0.8808 ±0.0014 | 0.9454 ±0.0009 | **0.9253** ±0.0016 |
| BT (K) | | | | | |
| Algorithm | MAE($\downarrow$) | RMSE($\downarrow$) | R$^2$($\uparrow$) | R$_p$($\uparrow$) | R$_s$($\uparrow$) |
| Climatology 1981-1988 | 2.8806 | 3.6864 | 0.8447 | 0.9485 | 0.9470 |
| Climatology 1989-2016 | 2.2024 | 2.8880 | 0.9036 | 0.9623 | 0.9606 |
| U-Net | 1.9920 ±0.0148 | 2.6652 ±0.0262 | 0.9164 ±0.0021 | 0.9644 ±0.0009 | 0.9616 ±0.0005 |
| Wave-MLP | 1.9376 ±0.0184 | 2.6221 ±0.0177 | 0.9172 ±0.0005 | 0.9647 ±0.0005 | 0.9619 ±0.0008 |
| Swin Transformer V1 | 1.9563 ±0.0329 | 2.6381 ±0.0397 | 0.9169 ±0.0038 | 0.9649 ±0.0009 | 0.9627 ±0.0008 |
| Swin Transformer V2 | 1.9516 ±0.0639 | 2.6277 ±0.0874 | 0.9183 ±0.0060 | 0.9641 ±0.0025 | 0.9619 ±0.0020 |
| **Focal Modulation** | **1.9179** ±0.0458 | **2.5745** ±0.0470 | **0.9204** ±0.0030 | **0.9664** ±0.0007 | **0.9636** ±0.0006 |

*research problem the authors would like to solve.*

**Answer to RC1.1** We revised the abstract accordingly (see lines 1-9 in the revised manuscript). The beginning of the abstract is revised as follows: *"Satellite-derived agricultural drought indices*

*can provide a complementary perspective of terrestrial vegetation trends and their integration for drought assessments under future climates is beneficial for providing more comprehensive assessments. However, satellite-derived drought indices are only available for observed periods. In this study, we aim to improve the agricultural drought assessments under future climate change by applying deep learning (DL) to predict satellite-derived vegetation indices from a regional climate simulation. The simulation is produced by the Terrestrial Systems Modelling Platform (TSMP) and performed in a free evolution mode over Europe. TSMP simulations incorporate variables from underground to the top of the atmosphere (Ground to Atmosphere G2A) and are widely used for research studies related to water cycle and climate change. We leverage these simulations for long-term forecasting and DL to map the forecast variables into Normalized Difference Vegetation Index (NDVI) and Brightness Temperature (BT) images that are not part of the simulation model.".*

**Comment RC1.2** *AVHRR in line 44 firstly appear without full name and also check for other short names.*

**Answer to RC1.2** We checked the abbreviations and adjusted the manuscript accordingly (see lines 60-61 in the revised manuscript).

**Comment RC1.3** *Line 50: change synthesis to be 'synthesize'.*

**Answer to RC1.3** Thank you. We corrected it (see line 76 in the revised manuscript).

**Comment RC1.4** *Line 195: What is DA represented? If the term only shows once in the manuscript, it is not necessary to use the short name.*

**Answer to RC1.4** DA refers to data assimilation. We rewrote line 195 in the old manuscript (see line 151 in the revised manuscript).

**Comment RC1.5** *Line 200: theta and lambda represents?*

**Answer to RC1.5** Theta and Lambda refer to the coordinates of the rotated pole. We rewrote the terms as follows: "... The grid specification for TSMP is a standardized rotated coordinate system ($\phi_{(rotated\ pole)}$ = 39.5° N, $\lambda_{(rotated\ pole)}$ = 18° E) with a spatial resolution ..." (see line 156 in the revised manuscript.)

**Comment RC1.6** *Line 253: Is it equation (5) is for NDVI for AVHRR instead of VIIRS as stated in line 253?*

**Answer to RC1.6** Thank you for this notice. Equation (5) computes NDVI for the AVHRR system. Lines 252-253 should be "... re-composite NDVI for AVHRR ...". We corrected it in the revised manuscript (see line 206 in the revised manuscript).

**Comment RC1.7** *Figure 1: It is better to use TSMP instead of TerrSysMP since TSMP is used all over the manuscript.*

**Answer to RC1.7** We changed the figure accordingly and replaced the term *TerrSysMP* with *TSMP* (please see Fig. 1).

**Comment RC1.8** *Line 325: it should have a period between 'efficiently In'.*

**Answer to RC1.8** We corrected it and added a period (see line 247 in the revised manuscript).

**Comment RC1.9** *Line 368-369: it is not clear why the third gate focuses on the water area? How do you know that?*

**Answer to RC1.9** The bright colors for specific regions in Fig. 3 represent higher values, which correspond to higher attentions of the model to that regions. For $G_3^k$, the attention values are high at areas with water. We revised lines 368-369 to make it clear (see the end of caption for Fig. 3 in the revised manuscript).

**Comment RC1.10** *Add references on line 430.*

**Answer to RC1.10** We added the following references [2–5] (see line 364 in the revised manuscript).

**Comment RC1.11** *Line 448-450: the sign N(0, I =0.02) is not clear. It is better to use normal distribution with zero mean and standard deviation.*

**Answer to RC1.11** We replaced the terms as suggested (see lines 382-385 in the revised manuscript).

**Comment RC1.12** *Line 461: reported how long it takes for the focal model, what about other DL models?*

**Answer to RC1.12** In Table B, we report the inference time to generate one sample for NDVI and BT containing 397×409×2 grid points. We included the table (see Table 3 and lines 434-437 in the revised manuscript).

Table B: Inference time in seconds for different DL models.

| Algorithm | GPU[1] | CPU[2] |
|---|---|---|
| U-Net | 0.09 ±0.02 | 5 ±0.2 |
| Wave-MLP | 0.28 ±0.00 | 10 ±0.3 |
| Swin Transformer V1 | 0.18 ±0.00 | 11 ±0.2 |
| Swin Transformer V2 | 0.19 ±0.00 | 11 ±0.2 |
| Focal Modulation | 0.24 ±0.01 | 12 ±0.1 |

[1]NVIDIA GeForce RTX 3090 GPU
[2]AMD Ryzen 9 3900X 12-Core CPU

**Comment RC1.13** *Line 471-472: What is these non-ML baselines? It is not clear for this statement.*

**Answer to RC1.13** The non-ML baselines represent the climatology. We rewrote the sentience to make it clearer in the revised manuscript (see line 425 in the revised manuscript).

**Comment RC1.14** *Line 516: change 'fro' to 'for'.*

**Answer to RC1.14** We corrected it (see line 481 in the revised manuscript).

**Comment RC1.15** *Figure 10: change 'sprint' to 'spring' in the left two plots in the first row.*

**Answer to RC1.15** Thank you for this notice. We corrected it (see Fig. 10 in the revised manuscript).

**Answers to Anonymous Referee #2**

**Major Comment**

**Comment RC2.1** *The abstract lacks some context. Not many readers will know TSMP, so it needs to be pointed out more clearly (and earlier) what the applications of this model system are and, then, why it is important to generate pseudo observations of NDVI and BT from simulation output, when real Earth observations are available for these quantities. There are some hints at the end of the abstract (climate simulations), but this remains vague and doesn't help to understand why this study was performed. A side aspect of this is that this lack of clarity makes it difficult to evaluate the stated model errors. Are MAEs of 0.027 for NDVI and 1.9 K for BT good or better than SOTA? What would be the reference here? Retrieval errors?*

**Answer to RC2.1** We revised the abstract accordingly (see lines 1-9 in the revised manuscript). The beginning of the abstract is revised as follows: *"Satellite-derived agricultural drought indices can provide a complementary perspective of terrestrial vegetation trends and their integration for drought assessments under future climates is beneficial for providing more comprehensive assessments. However, satellite-derived drought indices are only available for observed periods. In this study, we aim to improve the agricultural drought assessments under future climate change by applying deep learning (DL) to predict satellite-derived vegetation indices from a regional climate simulation. The simulation is produced by the Terrestrial Systems Modelling Platform (TSMP) and performed in a free evolution mode over Europe. TSMP simulations incorporate variables from underground to the top of the atmosphere (Ground to Atmosphere G2A) and are widely used for research studies related to water cycle and climate change. We leverage these simulations for long-term forecasting and DL to map the forecast variables into Normalized Difference Vegetation Index (NDVI) and Brightness Temperature (BT) images that are not part of the simulation model."*.
Please note that we do not aim to generate pseudo observations of NDVI/BT when satellite measurements are available but rather to predict NDVI/BT for periods where no satellite observations exist (i.e., future periods). However, we have to train and evaluate the model on a historical period. The real application is then to apply DL to a climate projection and predict NDVI/BT in the future. We agree that the stated model errors are hard to be interpreted without a reference. However, this is only mentioned to give an estimation about the accuracy that the model could achieve. The errors for satellite products highly depend on the spatial/temporal resolution and sensor being used. To the best of our knowledge, there are also no baselines models to compare with.

**Comment RC2.2** *Section 1 takes a couple of short-cuts and doesn't always provide good explanations to motivate this study. Please see detailed comments below. Related to this, in section 2 it is unclear why, for example, radiative transfer models are discussed here, and some of the content of this section would better belong elsewhere.*

**Answer to RC2.2** As suggested, sections 1 and 2 were revised (see answers to the minor comments RC2.4 to RC2.14). We have revised the introduction to make clearer (see lines 33-75 in the revised manuscript). We have integrated the discussion (lines 606-616 in the old manuscript) with the introduction (lines 45-54 in the revised manuscript). In addition, we have integrated the related works section (sections 2.1 and 2.2) with the introduction. More specifically, we have shortened the related works on radiative transfer models from section 2.1 and integrated them with the introduction (see lines 76-81 and 104-109 in revised manuscript). The related works from section 2.2 have been also shortened, summarized and integrated with the introduction (lines 83-104 and 109-114 in the revised manuscript).

**Comment RC2.3** *Sections 3-6 are largely OK, except for minor comments listed below.*

**Answer to RC2.3** We revised sections 3-6 accordingly. Please see the answers to minor comments RC2.15 to RC2.55.

**Comment RC2.4** *Section 7 variable importance: I like this analysis very much. However, I think one could point out that channel importance does not "explain" everything. First: if two variables are correlated, the network may decide to focus on one of them and the other one would seem unimportant, while it could provide almost the same information if it were alone. Second: seemingly "unimportant" variables may play an important role to get the final few percent accuracy out of the models. This could of course be tested by training a model on only the N most relevant variables and compare the results.*

**Answer to RC2.4** We are aware that conclusions from such an analysis should be taken carefully as discussed in lines 654-663. We moved this discussion to section 7 (see subsection 4.3 lines 570-578 in the revised manuscript).

**Comment RC2.5** *Section 8: this discussion comes as a surprise as it goes much deeper into remote sensing and modelling issues than any of the other parts of the paper. As indicated above, there is lack of information in the Introduction and related work sections. I therefore suggest to re-arrange some of the text and use some of the material of the discussion in these earlier sections. The discussion could then be shortened and focus more on the applicability and prospects of the new method.*

**Answer to RC2.5** As suggested, we deleted section 8 and moved parts of section 8 into earlier sections. More precisely, we moved the lines 617-620 into the introduction (lines 42-45 in the revised manuscript). We have integrated the discussion (lines 606-616 in the old manuscript) with the introduction (lines 45-54 in the revised manuscript). Lines 624-628 were integrated into the experimental results and analysis section (subsection 4.1 lines 470-474 in the revised manuscript). Lines 629-637 were also integrated into the experimental results and analysis section (subsection 4.1 lines 463-470 in the revised manuscript). Lines 639-642 were integrated into the experimental results and analysis section (section 4.2 lines 503-506 in the revised manuscript). Furthermore, lines 643-648 were shortened and integrated into the conclusions and outlook section (section 5 lines

591-593 in the revised manuscript). Finally, the discussion in lines 656-663 was integrated into the experimental results and analysis section (subsection 4.3 lines 570-578 in the revised manuscript).

**Minor Comments**

**Comment RC2.1** *Abstract l.3: why "intermediate step"? The image synthesis is the main product of the DL model, not an intermediate step in the modelling itself. The derivation of various indices is post-processing. Suggest to remove "in an intermediate step".*

**Answer to RC2.1** We removed it. Please see lines 5-9 in the revised manuscript.

**Comment RC2.2** *l.7: suggest rewording "… to assess the model's applicability to different seasons and regions…"*

**Answer to RC2.2** We revised it accordingly (see lines 15-16 in the revised manuscript).

**Comment RC2.3** *l.12 the unit of temperature is K, not K°.*

**Answer to RC2.3** Thank you for this notice. We checked the manuscript and replaced K° with K as noted.

**Comment RC2.4** *Introduction l.20: Suggest to remove the first sentence (motherhood statement) and integrate "under a changing climate" in the following sentence, which provides a more concise and precise start of the text. Not all droughts are extreme, and while extreme events are a good motivation, this study does not focus on extreme events, but rather tries to provide information to assess droughts or the risk of droughts.*

**Answer to RC2.4** We revised the introduction accordingly where we deleted the first sentence and integrated *"under a changing climate"* with the following sentence (see line 26 in the revised manuscript.)

**Comment RC2.5** *l.32 delete "in the future"*

**Answer to RC2.5** We agree as it looks redundant (see line 98 in the revised manuscript).

**Comment RC2.6** *l.33 ff. The link made here between climate models and the water cycle is a bit too direct. It is a known weakness of general circulation models (aka "climate models") that convection and rainfall are not well captured over many world regions. This is why there is a need for more specialized hydrology models, which are frequently used for regional instead of global simulations. Also, the introduction of drought indices comes somewhat unmotivated. The rationale behind these is usually to convert information from some instrument (or model) into a meaningful quantity that can be used to assess the state of some ecosystem or the climate system. Why focus on agricultural indices here? If this is intended, then this should be stated in the first motivation sentences for this study.*

**Answer to RC2.6** We revised the introduction accordingly as this appears confusing. The study is primarily related to remote sensing based agricultural drought events (please see lines 33-45

in the revised manuscript). The introduction of drought indices comes now in lines 36-40 in the revised manuscript.

**Comment RC2.7** *l.50-55 The "discussion" about retrievals is good and can be used for the motivation of this study, but it is missing an explicit reference to retrieval errors. The problems with current retrievals (or perhaps even fundamental problems = theoretiscal limitations of physics-based retrievals?) should be described more precisely and with some more detail.*

**Answer to RC2.7** We revised the introduction to make it clearer (see lines 33-75 in the revised manuscript). Please note that the study is not mainly related to retrieval errors. We added a short discussion on radiative transfer models (lines 76-81 in the revised manuscript).

**Comment RC2.8** *l.66 I would avoid the word downstream-application here (even though it is technically correct) and rather formulate "To showcase the value (or potential) or our approach, we calculate (or derive) ..."*

**Answer to RC2.8** We revised it according to the suggestion (see line 115 in the revised manuscript).

**Comment RC2.9** *l.74 This sentence is confusing in the context. Before, you give the impression that you rely on the model (implicitly assuming TSMP is perfect), whereas you now state that you can use the derived products to "examine the predicitve capability" of the model. As stated in the major comment above, the Introduction needs to be rewritten with a clearer explanation what this study is based on, how it is motivated (what doesn't work well at present?) and what are its primary objectives. Certainly, the aspect of model errors and their impact - or the potential of the method to quantify them - are one very relevant aspect that should at least shine through in the Introduction.*

**Answer to RC2.9** Thank you for this comment. We revised the introduction accordingly (see lines 33-75 in the revised manuscript). We have deleted the sentence:
*"Our results indicate that a direct prediction of vegetation products from TSMP with deep learning is an effective way to examine the overall predictive capability of TSMP to forecast agricultural drought events."* (please see lines 122-123 in the revised manuscript)
and we kept the sentence:
*"We believe that our approach could also be useful to combine deep learning with data assimilation, i.e., to simulate remote sensing products from down-scaled simulations and to be used as a supportive evaluation framework to further investigate the predictive capability of the simulation to reproduce drought events and consequently to improve the TSMP model development."* in lines 597-600 in the revised manuscript.

**Comment RC2.10** *Section 2.1: the review of radiative transfer models is OK, but the reader doesn't understand why there is half a page or more on cloud retrievals when this paper is about vegetation indices. It would be helpful to add an introductional sentence or two explaining why section 2 is structured in the way it is and what content is expected. The discussion in 2.1 is perhaps a little too detailed.*

**Answer to RC2.10** We merged the first two sections and shortened the discussion. More precisely, We have shortened the related works on radiative transfer models from section 2.1 and

integrated them with the introduction (see lines 76-81 and 104-109 in revised manuscript).

**Comment RC2.11** *l.118: The paragraph introducing the work of this study does not seem to connect to the general radiative transfer discussion above. The connection appears to be only methodologically (use of AI).*

**Answer to RC2.11** Section 2 was revised and integrated with the introduction. As mentioned, we have shortened the related works on radiative transfer models from section 2.1 and integrated them with the introduction (see lines 76-81 and 104-109 in revised manuscript).

**Comment RC2.12** *l.132 grammar "the interaction ... exhibits ... behavior".*

**Answer to RC2.12** We corrected them (see line 87 in the revised manuscript).

**Comment RC2.13** *l.162 awkward phrasing "a single indicator like NDVI excluding BT" - do you mean "... either NDVI or BT"? Or simply cut after "indicator".*

**Answer to RC2.13** We mean when only relying on NDVI. We rewrote the sentence (see lines 100-102 in the revised manuscript).

**Comment RC2.14** *l.162 ff. After reading section 2, it becomes clearer what this paper aims to do. Some of the text here should be moved to the Introduction, and section 2 should no longer explain what is done in this study, but concentrate on discussing what has been made available so far.*

**Answer to RC2.14** As mentioned in the response to the above comments, we have revised the introduction (see lines 33-75 in the revised manuscript). We have integrated the discussion (lines 606-616 in the old manuscript) with the introduction (lines 45-54 in the revised manuscript). In addition, we have integrated the related works section (sections 2.1 and 2.2) with the introduction. More specifically, we have shortened the related works on radiative transfer models from section 2.1 and integrated them with the introduction (see lines 76-81 and 104-109 in revised manuscript). The related works from section 2.2 have been also shortened, summarized and integrated with the introduction (lines 83-104 and 109-114 in the revised manuscript).

**Comment RC2.15** *l.178 delete "at" before "IBG-3 institute".*

**Answer to RC2.15** We deleted it (see line 133 in the revised manuscript).

**Comment RC2.16** *l.180 "near nature realization" - what do you mean by this? Every model is an abstraction of some sort, and many models aim to produce realistic results. However, this expression is not scientific.*

**Answer to RC2.16** We removed it (see lines 134-135 in the revised manuscript). We mean that the simulation serves as a near-natural reference for global change simulations to recreate a historical period without reanalysis. We mentioned this based on [6].

**Comment RC2.17** *l.182 ff. please harmonize grammar in the bullet list - some bullets have verbs others don't.*

**Answer to RC2.17** We revised the second bullet point *"The Community Land Model (CLM) version 3.5 is used to simulate the bio-geophysical processes on the land surface (Oleson et al., 2004, 2008)."* (please see line 139 in the revised manuscript).

**Comment RC2.18** *l.190 "a dynamic equilibrium" - this is not unambiguous and depends on the choice of start and end date, for example.*

**Answer to RC2.18** The dynamic equilibrium with the atmosphere (1979-1988) was obtained to initialize the subsurface and surface hydrologic and energy variables [6]. We made this clearer in the revised manuscript (see lines 145-146 in the revised manuscript).

**Comment RC2.19** *l.195 comment, related to l.180: a free running model without DA will always be further away from "nature" than a model with DA.*

**Answer to RC2.19** This is true. The simulation did not use any re-initialization or nudging. We removed DA (see line 151 in the revised manuscript).

**Comment RC2.20** *l.198 extending.*

**Answer to RC2.20** We corrected it (see line 154 in the revised manuscript.)

**Comment RC2.21** *l.199 why mention "with various vegetation types and climate conditions"? If you refer to Europe as a region, then this is kind of obvious and doesn't add information to this sentence describing the model set-up. If this refers instead to a property of the model or model output, then it doesn't belong here, but in a section where you describe the data and data distributions.*

**Answer to RC2.21** We wanted to emphasize that the extension of the region we study (Europe) includes different vegetation types and climate conditions. We agree this seems obvious so we deleted it (see lines 153-155 in the revised manuscript).

**Comment RC2.22** *l.201 and \*the\* model set-up.*

**Answer to RC2.22** We corrected it (see line 158 in the revised manuscript).

**Comment RC2.23** *l.204 I think you could add an extra sentence to say that DL has already been applied to TSMP simulations, instead of just referring to the papers.*

**Answer to RC2.23** We deleted this sentence and integrated it with the introduction (see lines 57-59 in the revised manuscript).

**Comment RC2.24** *l.221 what are upper and lower bounds of an ecosystem? Do you mean bounds of NDVI and BT for a specific ecosystem class? Also, the sentence "Consequently, ..." doesn't fit well. Better to write "Hence, ..." or "Thus, ..." or "Therefore, ..."*

**Answer to RC2.24** The upper and lower bounds are the min and max values for NDVI and BT for a specific pixel. We made this clearer and rephrased it (see line 173 in the revised manuscript).

**Comment RC2.25** *l.230 Is alpha a fixed coefficient or does it vary with ecosystem class or other parameters?*

**Answer to RC2.25** In our work, we used a standard value $\alpha = 0.5$ [7]. Please note that this does not affect NDVI and BT predictions. As mentioned in lines 267-269 and 637-639, it is better to calibrate $\alpha$ with respect to the location (see [8,9]). However, this is beyond the scope of our work to improve the weighting (see lines 220-222 in the revised manuscript).

**Comment RC2.26** *l.234 delete "Moreover,".*

**Answer to RC2.26** We deleted it (see line 187 in the revised manuscript).

**Comment RC2.27** *l.248-254 grammar (OK, but clearly non-native English).*

**Answer to RC2.27** We proofread these lines (see lines 201-206 in the revised manuscript).

**Comment RC2.28** *l.271 Please describe the data cube dimensions. Is there one datacube with (time, lat, lon) for each variable? Remove "observed".*

**Answer to RC2.28** We stored a datacube for each week (variable, lat, lon). We clarified this. "observed" is removed (see line 224 in the revised manuscript).

**Comment RC2.29** *l.275 zero*

**Answer to RC2.29** We corrected it (see line 228 in the revised manuscript).

**Comment RC2.30** *l.282 the theta doesn't belong in eq 7, which describes the mapping objective. It only comes in when you in fact use a model, i.e. when you describe the U-net.*

**Answer to RC2.30** We reordered the text and inserted Eq 7 after we mention the model (see lines 234-237 in the revised manuscript).

**Comment RC2.31** *l.300 number of channel\*s\*.*

**Answer to RC2.31** We corrected it (see line 265 in the revised manuscript).

**Comment RC2.32** *l.325 period missing. And: pixel representations.*

**Answer to RC2.32** We corrected it (see line 247 in the revised manuscript).

**Comment RC2.33** *l.328 Please be more specific: you refer to the quadratic scaling, which primarily limits the attention span, but not "applications" per se.*

**Answer to RC2.33** You are right. We refer to the quadratic computation complexity of the self-attention. We made this clearer (see lines 249-250 in the revised manuscript).

**Comment RC2.34** *l.337 input channel\*s\*.*

**Answer to RC2.34** We corrected it (see line 290 in the revised manuscript).

**Comment RC2.35** *l.347 remove "a" before "one".*

**Answer to RC2.35** We corrected it (see line 300 in the revised manuscript).

**Comment RC2.36** *l.350 reduce the number of model parameters.*

**Answer to RC2.36** We revised it as suggested (see line 302 in the revised manuscript).

**Comment RC2.37** *eq.11 I suggest to replace the somewhat clumsy expressions FocalModulation-Block etc. by shorter variable names which then need to be defined in the text, of course. This would improve readability of the equation.*

**Answer to RC2.37** We simplified the terms. More specifically, we replaced the terms Feed-ForwardLayer with emphFFL and FocalModulation with *FM* (see Eq. (11) lines 305-306 in the revised manuscript).

**Comment RC2.38** *l.379 "less" compared to what? I assume you mean MSE loss.*

**Answer to RC2.38** Correct. We mean here less than MSE. We made this explicit in the revised manuscript (see lines 332-333 in the revised manuscript).

**Comment RC2.39** *l.392 play *a* more important role.*

**Answer to RC2.39** We corrected it (see line 346 in the revised manuscript).

**Comment RC2.40** *l.425 I think it would be easier to describe the U-net baseline model by simply stating what it consists of instead of "reverse engineering" it by abstracting the focal attention blocks away.*

**Answer to RC2.40** We revised the description (see lines 360-361 in the revised manuscript).

**Comment RC2.41** *l.428 ff Please provide a few more details on the competitor models, such as number of layers, size of attention matrix etc. This could also be summarized in an Appendix, which should then be referenced here.*

**Answer to RC2.41** All baseline models share the number of layers but differ in the type of layers. The U-Net model does not use an attention mechanism, but follows the original U-Net design using 2D convolutions. Swin Transformer uses self-attention inside local windows. The implementation details about the models are mentioned in section 5 "Implementation details" (see subsection 3.5 in the revised manuscript).

**Comment RC2.42** *l.438 Do you mean "Apart from"?*

**Answer to RC2.42** Yes. We corrected it (see line 372 in the revised manuscript.)

**Comment RC2.43** *l.442 Why is the second climatology computing the future? 2016 is in the past. Also, grammar: future should be singular.*

**Answer to RC2.43** The main application of the study is to estimate the vegetation condition using a DL-based model for future periods where no satellite images are available. We compare the results of the DL models to two NDVI/BT climatologies from remote sensing observations. The first climatology (1981-1988) is used to show the limitation of using prescribed satellite phenology for future projection and we argue that a DL-based model maybe a better replacement (for example future simulations often prescribe vegetation condition in a satellite phenology-mode neglecting the inter-annual variability [1]). This is still not a fair comparison because the DL-models were trained on periods succeeding this climatology. The second climatology (1989-2016) is used to show that the DL-models still outperform the climatology. We could compute the second climatology (1989-2016) because we used a climate simulation in an overlapping period with historical remote sensing observations. This comparison is not possible for future climate projections since no satellite observations exist yet. We rewrote lines 438-444 to make it clear (see lines 372-377 in the revised manuscript).

**Comment RC2.44** *l.450 randomly perturbing.*

**Answer to RC2.44** We corrected it (see line 384 in the revised manuscript).

**Comment RC2.45** *l.451 this seems to contradict the preprocessing description in section 3. There you wrote that samples were averaged over a week, which implies that \*all\* samples are used in thre average. What is written here is a random estimator of the weekly average based on two days.*

**Answer to RC2.45** We use this as an augmentation technique only during training. For validation and test, we average all days. We made this clearer in the revised manuscript (see line 386 in the revised manuscript).

**Comment RC2.46** *l.460 typo finally.*

**Answer to RC2.46** We corrected it (see line 395 in the revised manuscript).

**Comment RC2.47** *l.466 remove comma.*

**Answer to RC2.47** We removed it (see line 420 in the revised manuscript).

**Comment RC2.48** *l.475 I don't understand the reference to radiative transfer models here and suggest it be removed.*

**Answer to RC2.48** We removed it as suggested (see lines 429-430 in the revised manuscript).

**Comment RC2.49** *l.484 As shown.*

**Answer to RC2.49** We corrected it (see line 441 in the revised manuscript).

**Comment RC2.50** *l.486 shown \*in\* Figs...*

**Answer to RC2.50** We corrected it (see line 444 in the revised manuscript).

**Comment RC2.51** *l.493 weakness\*es\*.*

**Answer to RC2.51** We corrected it (see line 450 in the revised manuscript).

**Comment RC2.52** *l.505 This discussion on errors could go one step further. While I agree that a full error attribution may be beyond the scope of this paper, you could at least give some indications, which errors come from the TSMP input data, which are from the observations and which may be DL model errors. This would be especially relevant for the TSMP data. For example, by showing the variability in different 2-day "weekly" samples and how this translates into different DL model outputs. Or you could apply some systematic perturbations to the TSMP output, thereby roughly correcting known model biases, and then see how this change the results. (OK, l.537 provides at least some indication already, and you could also refer to section 7 for additional insights).*

**Answer to RC2.52** We extended the discussion and referred to section 7 (subsection 4.2 in the revised manuscript) as suggested (see lines 460-477 in the revised manuscript). We have revised and moved the discussion in lines 629-636 into the lines 463-470 in the revised manuscript. The discussion in lines 624-628 was revised and moved into the lines 470-474 in the revised manuscript.

**Comment RC2.53** *l.516 typo "for".*

**Answer to RC2.53** We corrected it (see line 481 in the revised manuscript).

**Comment RC2.54** *l.540 , who showed.*

**Answer to RC2.54** We corrected it (see line 509 in the revised manuscript).

**Comment RC2.55** *l.541 affecting.*

**Answer to RC2.55** We corrected it (see line 509 in the revised manuscript).

**Comment RC2.56** *l.723 Klaus Goergen.*

**Answer to RC2.56** We corrected it (see line 639 in the revised manuscript).

**References**

[1] D. M. Lawrence, R. A. Fisher, C. D. Koven, K. W. Oleson, S. C. Swenson, G. Bonan, N. Collier, B. Ghimire, L. van Kampenhout, D. Kennedy, E. Kluzek, P. J. Lawrence, F. Li, H. Li, D. Lombardozzi, W. J. Riley, W. J. Sacks, M. Shi, M. Vertenstein, W. R. Wieder, C. Xu, A. A. Ali, A. M. Badger, G. Bisht, M. van den Broeke, M. A. Brunke, S. P. Burns, J. Buzan, M. Clark, A. Craig, K. Dahlin, B. Drewniak, J. B. Fisher, M. Flanner, A. M. Fox, P. Gentine, F. Hoffman, G. Keppel-Aleks, R. Knox, S. Kumar, J. Lenaerts, L. R. Leung, W. H. Lipscomb, Y. Lu, A. Pandey, J. D. Pelletier, J. Perket, J. T. Randerson, D. M. Ricciuto, B. M. Sanderson, A. Slater, Z. M. Subin, J. Tang, R. Q. Thomas, M. Val Martin, and X. Zeng, "The community land model version 5: Description of new features, benchmarking, and impact of forcing uncertainty," *Journal of Advances in Modeling Earth Systems*, vol. 11, no. 12, pp. 4245–4287, 2019. [Online]. Available: https://agupubs.onlinelibrary.wiley.com/doi/abs/10.1029/2018MS001583

[2] L. Wang, R. Li, C. Duan, C. Zhang, X. Meng, and S. Fang, "A novel transformer based semantic segmentation scheme for fine-resolution remote sensing images," *IEEE Geoscience and Remote Sensing Letters*, vol. 19, pp. 1–5, 2022.

[3] L. Gao, H. Liu, M. Yang, L. Chen, Y. Wan, Z. Xiao, and Y. Qian, "Stransfuse: Fusing swin transformer and convolutional neural network for remote sensing image semantic segmentation," *IEEE Journal of Selected Topics in Applied Earth Observations and Remote Sensing*, vol. 14, pp. 10 990–11 003, 2021.

[4] L. Wang, R. Li, C. Zhang, S. Fang, C. Duan, X. Meng, and P. M. Atkinson, "Unetformer: A unet-like transformer for efficient semantic segmentation of remote sensing urban scene imagery," *ISPRS Journal of Photogrammetry and Remote Sensing*, vol. 190, pp. 196–214, 2022. [Online]. Available: https://www.sciencedirect.com/science/article/pii/S0924271622001654

[5] A. A. Aleissaee, A. Kumar, R. M. Anwer, S. Khan, H. Cholakkal, G.-S. Xia, and F. S. Khan, "Transformers in remote sensing: A survey," *Remote Sensing*, vol. 15, no. 7, 2023. [Online]. Available: https://www.mdpi.com/2072-4292/15/7/1860

[6] C. Furusho-Percot, K. Goergen, C. Hartick, K. Kulkarni, J. Keune, and S. Kollet, "Pan-european groundwater to atmosphere terrestrial systems climatology from a physically consistent simulation," *Scientific Data*, vol. 6, no. 1, p. 320, 2019. [Online]. Available: https://doi.org/10.1038/s41597-019-0328-7

[7] W. Yang, F. Kogan, and W. Guo, "An ongoing blended long-term vegetation health product for monitoring global food security," *Agronomy*, vol. 10, no. 12, 2020. [Online]. Available: https://www.mdpi.com/2073-4395/10/12/1936

[8] J. Zeng, R. Zhang, Y. Qu, V. A. Bento, T. Zhou, Y. Lin, X. Wu, J. Qi, W. Shui, and Q. Wang, "Improving the drought monitoring capability of vhi at the global scale via ensemble indices for various vegetation types from 2001 to 2018," *Weather and Climate Extremes*, vol. 35, p. 100412, 2022. [Online]. Available: https://www.sciencedirect.com/science/article/pii/S2212094722000068

[9] J. Zeng, T. Zhou, Y. Qu, V. Bento, J. Qi, Y. Xu, Y. Li, and Q. Wang, "An improved global vegetation health index dataset in detecting vegetation drought," *Scientific Data*, vol. 10, p. 338, 05 2023.

---

## Author Response (AR3)

Dear Editor,

Thank you for coordinating the review process and accepting our manuscript "Focal-TSMP: Deep learning for vegetation health prediction and agricultural drought assessment from a regional climate simulation" for publication in GMD. We would also like to thank the anonymous referees for reviewing and improving the manuscript.

We corrected the revised manuscript according to the technical issues raised by the reviewers and fixed a minor issue in the Figures:

- We added notes for Tables 1 and 2 about the bold and underlined numbers as well as the standard deviation for the three runs.

- We split the first sentence in the abstract into two sentences and replaced the wording "for observed periods" by "for the Earth observation era".

- In Figures 5, 6, 7, F1, and F2, we replaced K° with K.

- In the captions of Figures 8 and 9, we corrected the reference from Section 3.4 to 3.

- In the caption of Figure F2, we corrected the reference from Figure 8 to Figure 9.

- In the captions of Figures F1 and F2, we corrected the reference from Section 5 to 3 and removed the reference to Table 1.

Kind regards,

Mohamad Hakam Shams Eddin